_Article_

# Tension-sensitive LINC-RhoA signaling prevents chromatin bridge breakage in cytokinesis

Sofia Balafouti ⬤, Maria Kabouraki ⬤, George Zachos ⬤ ✉ & Eleni Petsalaki ⬤ ✉

## Abstract

In the presence of chromatin bridges in cytokinesis, human cells retain actin-rich structures (actin patches) at the base of the intercellular canal to prevent chromosome breakage. Here, we show that daughter nuclei connected by chromatin bridges are under mechanical tension that requires interaction of the nuclear membrane Sun1/2-Nesprin-2 Linker of Nucleoskeleton and Cytoskeleton (LINC) complex with the actin cytoskeleton, and an intact nuclear lamina. This nuclear tension promotes accumulation of Sun1/2-Nesprin-2 proteins at the base of chromatin bridges and local enrichment of the RhoA-activator PDZ RhoGEF through PDZ-binding to cytoplasmic Nesprin-2 spectrin repeats. In turn, PDZ RhoGEF activates the small GTPase RhoA and downstream ROCK-LIMK-Cofilin and mDia1 signaling to generate actin patches and prevent chromatin bridge breakage in cytokinesis. These findings identify a novel mechanosensing mechanism by which chromatin bridges promote remodeling of the actin cytoskeleton, through tension-induced activation of LINC-PDZ RhoGEF-RhoA signaling, to generate actin patches to preserve genome integrity.

**Keywords** Actin Patches; Chromatin Bridges; LINC; Nesprin-2; RhoA
**Subject Categories** Cell Adhesion, Polarity & Cytoskeleton; Cell Cycle

## Introduction

Chromatin bridges are strings of mis-segregated chromatin connecting the anaphase poles or daughter nuclei in mitotic cell division and can arise from incomplete DNA replication or decatenation, or from dicentric chromosomes generated by end-to-end chromosome fusions (Finardi et al, 2020). The role of chromatin bridge breakage-fusion-bridge cycles in generating genomic instability has been recognized for decades (McClintock, 1939); furthermore, chromosomes trapped inside chromatin bridges are susceptible to chromothripsis, which can cause burst-like accumulation of genomic alterations within a short window relative to the cell life-history that can drive carcinogenesis (Cortes-Ciriano et al, 2020; Maciejowski et al, 2020; Mazzagatti et al, 2024; Umbreit et al, 2020; Voronina et al, 2020). As a result, preventing chromatin bridges from breaking is essential for cells to maintain genome stability.

In response to chromatin bridges in cytokinesis, eukaryotic cells impose an abscission-delay, called "the abscission checkpoint" in human cells, to prevent chromatin breakage or tetraploidization by regression of the cleavage furrow (Carlton et al, 2012; Petsalaki and Zachos, 2016; Petsalaki and Zachos, 2021a, Steigemann et al, 2009; Thoresen et al, 2014). It was recently shown that the DNA topoisomerase IIα (Top2α) enzyme binds to "knots" of catenated DNA on chromatin bridges and forms abortive Top2-cleavage complexes next to the midbody (Petsalaki et al, 2023). In turn, the DNA damage sensor protein Rad17 promotes recruitment of the Mre11-Rad50-Nbs1 protein complex on Top2α-induced double-strand DNA ends and activates downstream ATM-Chk2-INCENP signaling, to ensure optimal localization of the Aurora B kinase at the midbody (Petsalaki et al, 2023; Petsalaki and Zachos, 2021b, Petsalaki and Zachos, 2024). There, Aurora B phosphorylates the endosomal sorting complex required for transport-III (ESCRT-III) subunit Chmp4c (Capalbo et al, 2012; Carlton et al, 2012; Petsalaki and Zachos, 2016), leading to inhibition of the ATPase Vps4 on ESCRT-III filaments at the abscission site to delay the final cut (Caballe et al, 2015; Morita et al, 2007; Thoresen et al, 2014). Furthermore, control cells with chromatin bridges prevent the depolymerization of actin filaments inside the intercellular canal by recruiting the human methionine sulfoxide reductase B2 (MsrB2) to the midbody, to delay recruitment of ESCRT-III proteins to the abscission site and to delay abscission (Bai et al, 2020).

In addition, human cells with chromatin bridges generate structures of polymerized actin (called "actin patches") at the base of the cytoplasmic canal that connects the two daughter cells, to stabilize DNA bridges (Dandoulaki et al, 2018; Steigemann et al, 2009). Control cells retain the actin patches until the chromatin bridges are resolved; furthermore, impaired actin patches correlate with chromatin bridge breakage that is not caused by premature abscission (Dandoulaki et al, 2018; Steigemann et al, 2009). The non-receptor protein-tyrosine kinase Src that regulates actin dynamics and its downstream target focal adhesion kinase (FAK) are required for optimal accumulation of actin patches in cytokinesis with chromatin bridges (Dandoulaki et al, 2018). However, how cells "sense" chromatin bridges to generate actin patches and the molecular mechanisms involved are incompletely understood.

The LINC (Linker of Nucleoskeleton and Cytoskeleton) complex spans the nuclear envelope and directly connects major cytoskeletal networks to the nuclear interior (Bouzid et al, 2019; Crisp et al, 2006). This offers the cell a grid for transmitting mechanical forces from plasma membrane receptors to the nucleus

Department of Biology, University of Crete, Vassilika Vouton, Heraklion 70013, Greece. ✉E-mail: gzachos@uoc.gr; grad600@edu.biology.uoc.gr

to induce molecular pathways that allow cells to adapt to new conditions, a process known as mechanotransduction that is relevant to several human diseases and cancer (Broders-Bondon et al, 2018; Di et al, 2023; DuFort et al, 2011; Janin et al, 2017). LINC complexes comprise an outer nuclear membrane Nesprin (nuclear envelope spectrin repeat-enriched protein) and an inner nuclear membrane SUN (Sad1/UNC-84)-domain protein (Sun1 and Sun2 in vertebrate somatic cells) (Starr and Fridolfsson, 2010; Tapley and Starr, 2013). Nesprin-2 is a giant protein (~800 kDa) that contains N-terminal tandem actin-binding calponin homology (CH) domains, a spectrin repeat (SR) rod domain comprising 56 SRs, and a C-terminal KASH (Klarsicht/ANC-1/SYNE homology) domain which spans the outer nuclear membrane and extends into the space between the outer and inner nuclear membrane (Zhang et al, 2001; Zhen et al, 2002). There, it interacts with the SUN-domain of Sun1/2, which spans the inner nuclear membrane, to connect the LINC complex to the nuclear lamina and the chromatin, to transduce forces from dynamic cytoskeletal networks to the nuclear interior (Haque et al, 2006; McGee et al, 2006; Padmakumar et al, 2005; Sosa et al, 2012).

RhoA is a member of the Rho GTPase family of proteins that regulate a diverse range of cellular functions including cell polarity, cell movement and cell division through their abilities to modulate the actin cytoskeleton and regulate microtubule dynamics (Jaffe and Hall, 2005; Wojnacki et al, 2014). They cycle between an active GTP-bound and an inactive GDP-bound state and are regulated by GTPase-activating proteins (GAPs), which enhance the intrinsic GTPase activity of Rho proteins leading to their inactivation, and guanine nucleotide-exchange factors (GEFs) that catalyze the exchange of GDP for GTP thus promoting Rho activity (Mosaddeghzadeh and Ahmadian, 2021). RhoA activates mDia1, a formin molecule that catalyzes actin nucleation and polymerization to produce long, straight actin filaments (Watanabe et al, 1997). Furthermore, RhoA activates ROCK (Rho-associated protein kinase), a serine/ threonine kinase that can phosphorylate LIMK (LIN-11/Isl-1/MEC-3 kinase) to activate it; in turn, active LIMK phosphorylates and inactivates the actin-depolymerizing and severing factor Cofilin, resulting in stabilization of existing actin filaments and increasing their content (Arber et al, 1998; Matsui et al, 1996; Moriyama et al, 1996; Mosaddeghzadeh and Ahmadian, 2021; Ohashi et al, 2000). However, a role for the LINC complex or RhoA in cytokinesis with chromatin bridges has not been previously reported.

Here, we show that RhoA and its effectors ROCK, LIMK, phospho-Cofilin and mDia1 localize to actin patches; furthermore, inhibition of the above proteins impairs actin patches and correlates with chromatin bridge breakage in cytokinesis in human cells. The RhoA regulator PDZ (PSD-95/DGL/ZO-1) RhoGEF also localizes to actin patches and is required for actin patch formation. We also show that, in control cells with intact chromatin bridges, the daughter nuclei exhibit oval-shaped nuclear chromatin and nuclear envelope distortions, indicating that they are under mechanical ("nuclear") tension. This nuclear tension correlates with accumulation of the Sun1/2-Nesprin-2 LINC complex at the "front" of the nucleus, near the base of the chromatin bridge. We show that impaired Sun1/2-Nesprin-2 complex correlates with reduced nuclear tension and with impaired actin patches. By using mutant mini-Nesprin-2 proteins and fusion protein strategies in depletion-replacement experiments, we show that nuclear tension

requires Nesprin-2-binding to the actin cytoskeleton and an intact nuclear lamina. Furthermore, we demonstrate that the cytoplasmic Nesprin-2 spectrin repeats (SRs) bind to PDZ RhoGEF in cell extracts and in vitro and are required for actin patches and stable chromatin bridges in the presence of nuclear tension. We propose that, in response to nuclear tension in cytokinesis with chromatin bridges, the Sun1/2-Nesprin-2 LINC complex localises asymmetrically proximal to the chromatin bridge. There, the Nesprin-2 SRs bind to and recruit PDZ RhoGEF to locally activate RhoA, to generate actin patches through RhoA-ROCK-LIMK-Cofilin and RhoA-mDia1 signaling and prevent chromatin bridge breakage in cytokinesis.

# Results

## RhoA promotes actin patches and prevents chromatin bridge breakage in cytokinesis

RhoA is an actin modulator, we therefore investigated a potential role for RhoA at actin patches. Confocal microscopy analysis of fixed human colon carcinoma BE cells in telophase showed that depletion of RhoA by siRNA (siRhoA) or treatment of cells with Y16, an inhibitor of G-protein-coupled RhoA GEFs (RhoAi), diminished actin patches compared with untransfected or vehicle (DMSO) controls, and this was not due to chromatin bridge breakage or to reduced total levels of actin by western blotting (Fig. 1A–D; Appendix Fig. S1A–D). BE cells in late cytokinesis without chromatin bridges (i.e., connected by midbodies exhibiting midbody thickness of 400–700 nm) were devoid of actin patches (Fig. 1D; Appendix Fig. S1E) (Petsalaki and Zachos, 2016; Petsalaki and Zachos, 2021b). RhoA-deficient cells also exhibited increased frequency of broken chromatin bridges in cytokinesis (Fig. 1E). Furthermore, after depletion of the endogenous protein, expression of RhoA$^{Res}$, resistant to degradation by siRhoA, fused to GFP (GFP:RhoA$^{Res}$) rescued the actin patches and reduced chromatin bridge breakage compared with controls expressing GFP-alone (Appendix Fig. S1F–K).

To further investigate the role of RhoA at actin patches, HeLa Lap2b:RFP/Lifeact:GFP cells expressing the inner nuclear envelope marker Lamina-associated polypeptide 2b fused to RFP (Lap2b:RFP), which correlates with chromatin bridges (Steigemann et al, 2009), and the F-actin-binding peptide Lifeact fused to GFP (Lifeact:GFP) were synchronized in prometaphase by nocodazole-treatment (Fig. 1F). After release, cells displaying Lap2b:RFP bridges in cytokinesis were monitored for up to 120 min by live-cell imaging, in the presence of vehicle control (DMSO) or RhoAi (Fig. 1F). We found that 30/30 control (DMSO) HeLa cells with Lap2b:RFP bridges exhibited actin patches and sustained the Lap2b:RFP bridges and intercellular canals for the duration of the experiment (Fig. 1G; Movies EV1–3). In contrast, 18/33 (54%) HeLa cells treated with RhoAi exhibited breakage of the Lap2b:RFP bridges and intercellular canals after 50 ± 36 min, and this correlated with impaired actin patches in RhoA-deficient cells compared with controls (Fig. 1H–J; Movies EV4–6). DNA bridge breakage after RhoA inhibition was also examined in BE cells stained with the DNA dye Biotracker, by live-cell imaging. We found that 22/22 vehicle control (DMSO) BE cells with DNA bridges sustained the DNA bridges and intercellular canals for the

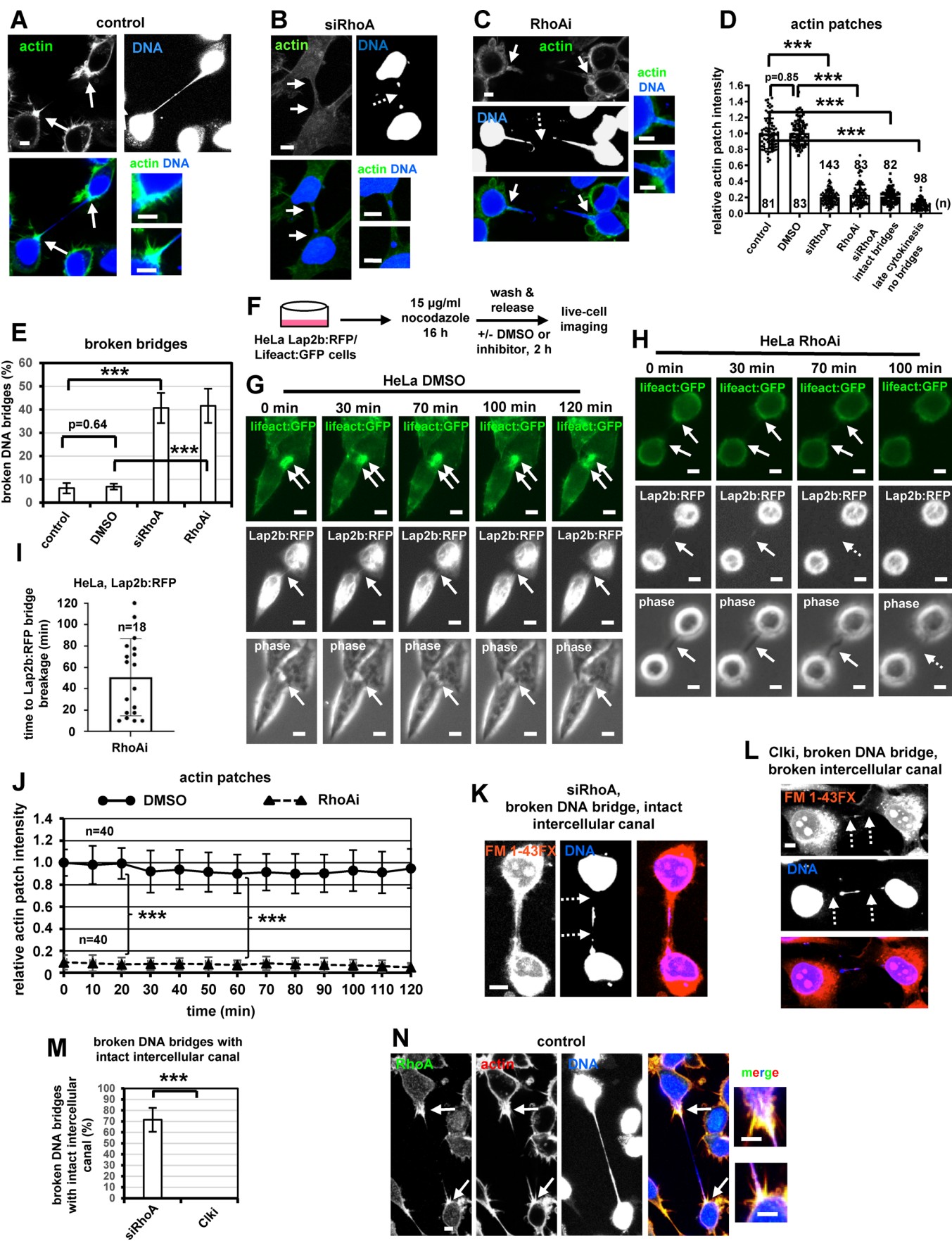

**Figure 1. Inhibition of RhoA impairs actin patch formation and correlates with chromatin bridge breakage in cytokinesis.**

(A–C) Actin patches and DNA bridges in untransfected control BE cells with chromatin bridges, cells transfected with RhoA siRNA (siRhoA), or treated with 50 μM Y16 (RhoAi). (D) Actin patches intensity. Cells were treated as in A-C, or with 50 μM DMSO for 30 min. Mean ± SD from n cells from two independent experiments. Values in control were set to 1. ***$P = 1.18E-109$ (control vs siRhoA), 6.78426E-76 (control vs siRhoA intact bridges), 2.80E-97 (control vs late cytokinesis no bridges), 1.20E-77 (DMSO vs RhoAi) by ANOVA and Student's t test. (E) Percentage of broken DNA bridges. Mean ± SD from three independent experiments ($n = 95, 56, 191, 146$). ***$P = 6.00E-07$ (control vs siRhoA), 0.0001 (DMSO vs RhoAi) by ANOVA and Student's t test. (F) Protocol for cell-synchronization in prometaphase followed by live-cell imaging. (G, H) HeLa cells expressing Lap2b:RFP and Lifeact:GFP were analyzed by live-cell imaging. Time is from the detection of Lap2b:RFP bridges. (I) Time to Lap2b:RFP bridge breakage. Mean ± SD from n cells. (J) Intensity of actin patches in cells from (G, H). Mean ± SD from n patches from three independent experiments. ***$P = 1.08E-43$ (DMSO vs RhoAi, 20 min), 2.22E-43 (DMSO vs RhoAi, 60 min) by Student's t test. (K, L) Examples of cells with DNA bridges exhibiting broken or intact intercellular canals after labeling with the FM 1-43FX lipophilic dye, in the presence of siRhoA or after treatment with 1 μM TG003 (Clki) for 5 h. (M) Frequency of cells with broken DNA bridges exhibiting intact intercellular canals. Mean ± SD from three independent experiments ($n = 64, 58$) ***$P = 0.00034$ (siRhoA vs Clki) by Student's t test. (N) Localization of RhoA in cells with chromatin bridges. Intact arrows indicate actin patches, intact Lap2b:RFP bridges or intercellular canals. Broken arrows indicate broken Lap2b:RFP bridges, DNA bridges or intercellular canals. Insets show high magnifications of the canal bases. Numbers below/next to each bar indicate n. Bars, 5 μm. Source data are available online for this figure.

duration of the experiment, whereas 17/37 (46%) cells treated with RhoAi exhibited breakage of the DNA bridges and intercellular canals after 59 ± 25 min (Appendix Fig. S1L–N; Movies EV7–10). Collectively, these results show that RhoA promotes actin patch formation and prevents chromatin bridge breakage in cytokinesis.

## Chromatin breakage in RhoA-deficient cells is not caused by abscission

Approximately 70% of RhoA-depleted cells with broken DNA bridges exhibited intact intercellular canals as evidenced by staining with the fluorescence lipophilic dye FM 1-43FX that labels membranes (Fig. 1K,M; Appendix Fig. S2A,B) (Dandoulaki et al, 2018), suggesting that chromatin breakage in RhoA-deficient cells does not correlate with abscission. For comparison, 45/45 cells with broken DNA bridges treated with Clki exhibited broken intercellular canals, indicating that Clki-treated cells had completed abscission (Fig. 1L,M). These results suggest that chromatin breakage in RhoA-deficient cells is not caused by abscission, but correlates with impaired formation of actin patches in cytokinesis.

## RhoA and RhoA-signaling proteins localize to actin patches

Proteins that regulate actin remodeling can localize to actin structures. We found that both the endogenous RhoA and transfected GFP:RhoA localized to actin patches in control cells with chromatin bridges (Fig. 1N; Appendix Fig. S1I). The RhoA signaling proteins ROCK and phosphorylated (active) LIMK-threonine 508 (T508), which is a phosphorylation target of ROCK, also localized to actin patches in control cells; furthermore, inhibition of RhoA diminished localization of ROCK and phosphorylated LIMK-T508 to the base of the intercellular canal, regardless of whether the chromatin bridges were broken or intact, compared with controls (Fig. 2A–F; Appendix Fig. S2C,D). These results suggest a role for RhoA-signaling proteins in actin patch formation.

## RhoA promotes actin patches through ROCK-LIMK-Cofilin signaling

Inhibition of ROCK by treatment of cells with the ROCK inhibitor Y-27632 (ROCKi) also diminished localization of phosphorylated

LIMK-T508 to the base of the intercellular canal compared with controls in cytokinesis with chromatin bridges (Fig. 2D,F; Appendix Fig. S2E). Treatment of cells with ROCKi or with the LIMK inhibitor TH-257 (LIMKi) impaired actin patches and increased the frequency of broken chromatin bridges in cytokinesis compared with controls (Fig. 2G–K; Appendix Fig. S2F,G). Also, after release of HeLa Lap2b:RFP/Lifeact:GFP cells from nocodazole (Fig. 1F), 28/82 (34%) cells with Lap2b:RFP bridges treated with the ROCKi and 31/74 (42%) cells treated with the LIMKi exhibited breakage of Lap2b:RFP bridges and intercellular canals, and this correlated with impaired actin patches in ROCK-deficient or LIMK-deficient cells compared with vehicle controls (DMSO) by live-cell microscopy (Appendix Fig. S2H–M; Movies EV11–16). Similarly, 14/30 (47%) BE cells stained with the DNA dye Biotracker and treated with the ROCKi exhibited breakage of DNA bridges and intercellular canals after 62 ± 23 min (Appendix Fig. S3A,B; Movies EV17 and 18). These results show that ROCK and LIMK catalytic activities are required for actin patch formation and stable chromatin bridges in cytokinesis.

Expression of myc-tagged truncated ROCK-Δ2 (amino acids 1–727) that is constitutively active (myc-ROCK Δ2) (Priya et al, 2017), or expression of a phosphomimetic mutant (constitutively active) LIMK1, in which T508 is replaced by two glutamic acid residues, fused to GFP (GFP:LIMK-T508EE) (Ohashi et al, 2000), but not the wild-type (WT) proteins, rescued actin patches and prevented chromatin bridge breakage after RhoA-depletion compared with WT controls (Figs. 2L–Q and 3A–F; Appendix Fig. S3C,D), showing that RhoA acts upstream of ROCK and LIMK in actin patch formation. Phosphorylated Cofilin-serine 3 (S3), which is a target of LIMK, also localized to actin patches in control cells; furthermore, inhibition of RhoA or LIMK impaired localization of phospho-Cofilin-S3 at the base of the intercellular canal compared with controls in cytokinesis with chromatin bridges (Fig. 3G–J; Appendix Fig. S3E,F). These results suggest that the RhoA-ROCK-LIMK-Cofilin signaling pathway promotes actin patches in cytokinesis with chromatin bridges.

## mDia1 cooperates with ROCK-LIMK signaling to prevent chromatin bridge breakage in cytokinesis

The RhoA-effector protein mDia1 also localized to actin patches in control cells; furthermore, RhoA-depletion diminished localization of mDia1 to the base of the intercellular canal compared with

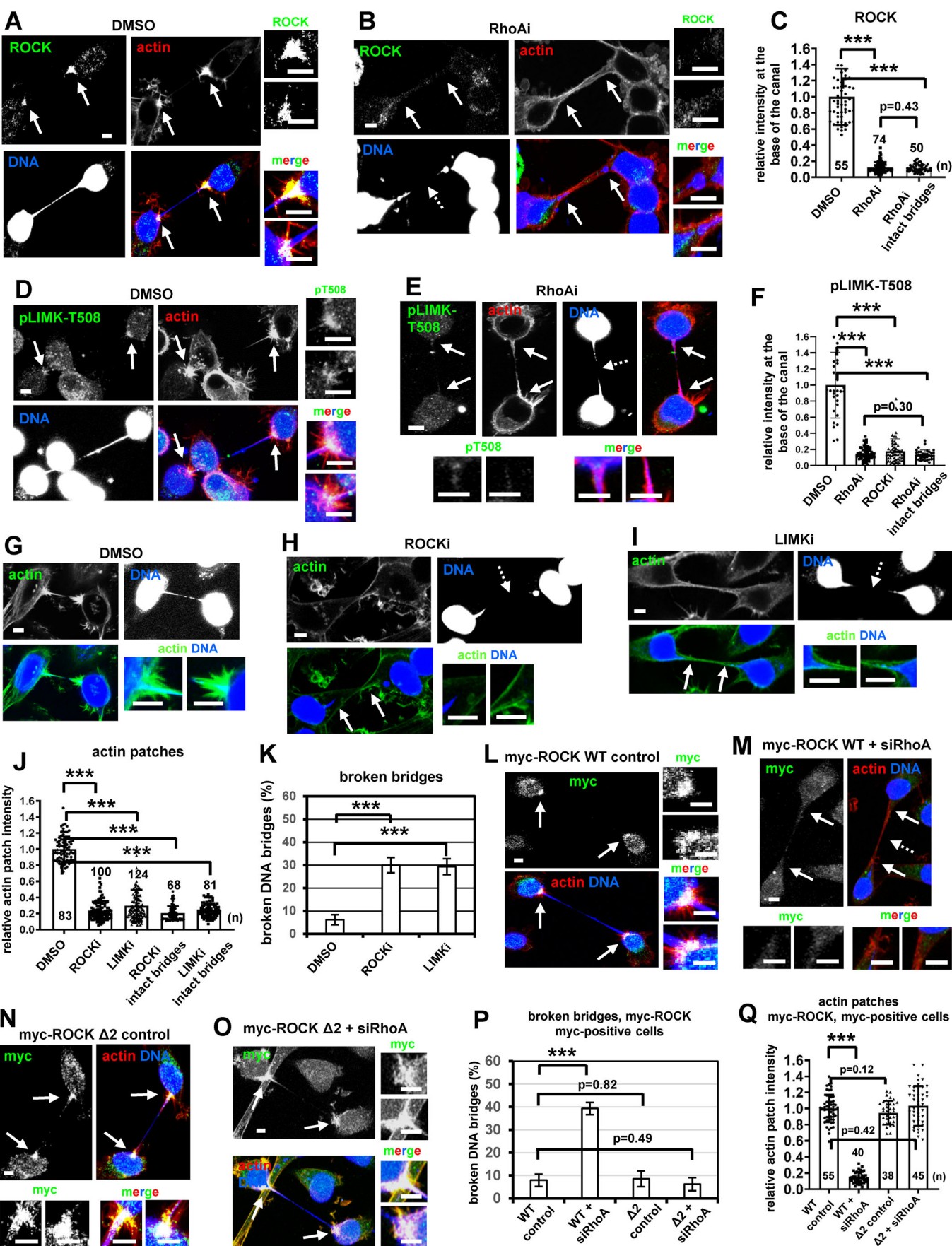

**Figure 2. ROCK inhibition diminishes actin patches and correlates with chromatin bridge breakage in cytokinesis.**

(A, B) Localization of ROCK in BE cells with chromatin bridges treated with 50 µM DMSO, or with 50 µM Y16 (RhoAi) for 30 min. (C) ROCK fluorescence intensity. Mean ± SD from n cells from two independent experiments. Values in DMSO were set to 1. ***P = 2.68E-43 (DMSO vs RhoAi), 3.90E-33 (DMSO vs RhoAi intact bridges) by Student's t test. (D, E) Localization of phosphorylated LIMK-T508 (pT508). (F) Phospho-LIMK-T508 intensity. Mean ± SD from n cells from two independent experiments. Values in DMSO were set to 1. ***P = 5.93E-28 (DMSO vs RhoAi), 1.70E-22 (DMSO vs ROCKi), 3.02E-18 (DMSO vs RhoAi intact bridges) by ANOVA and Student's t test. (G–I) Actin patches and DNA bridges in cells treated with DMSO, 10 µM Y27632 (ROCKi) for 1 h, or with 10 µM TH-257 (LIMKi) for 5 h. (J) Actin patches intensity. Mean ± SD from n cells from two independent experiments. Values in DMSO were set to 1. ***P = 5.05E-88 (DMSO vs ROCKi), 1.85E-69 (DMSO vs LIMKi), 7.43E-78 (DMSO vs ROCKi intact bridges), 2.13E-83 (DMSO vs LIMKi intact bridges) by ANOVA and Student's t test. (K) Percentage of broken DNA bridges. Mean ± SD from three independent experiments (n = 56, 59, 79). ***P = 0.00034 (DMSO vs ROCKi), 0.00047 (DMSO vs LIMKi) by ANOVA and Student's t test. (L–O) Actin patches in cells expressing wild-type (WT) or truncated myc-ROCK Δ2 in the absence (control) or presence of RhoA siRNA (siRhoA). Intact arrows indicate actin patches and/or canal bases. Broken arrows indicate broken DNA bridges. Insets show high magnifications of the canal bases. (P) Percentage of broken DNA bridges in myc-positive cells. Mean ± SD from three independent experiments (n = 64, 66, 70, 64). ***P = 0.00014 (WT control vs WT+siRhoA) by ANOVA and Student's t test. (Q) Actin patches intensity in myc-positive cells. Mean ± SD from n cells from two independent experiments. Values in WT control were set to 1. ***P = 1.01E-49 (WT control vs WT+siRhoA) by ANOVA and Student's t test. Numbers below/next to each bar indicate n. Scale bars, 5 µm. Source data are available online for this figure.

controls in cytokinesis with chromatin bridges (Fig. 3K–M; Appendix Fig. S3G). Inhibition of mDia1 by the formin inhibitor SMIFH2 (mDia1i) reduced actin patches and increased chromatin bridge breakage in cytokinesis compared with controls (Fig. 3N–P; Appendix Fig. S3H). Simultaneous treatment of cells with RhoAi and mDia1i (RhoAi+mDia1i), or with RhoAi+ROCKi, did not exacerbate broken chromatin bridges compared with cells treated with RhoAi-only (Fig. 3P), suggesting that RhoA operates upstream of mDia1 and ROCK to prevent chromatin breakage. However, treatment of cells with mDia1i+ROCKi or mDia1i+LIMKi increased the frequency of broken chromatin bridges compared with cells treated with mDia1i-, ROCKi- or LIMKi-only, indicating that mDia1 functions in a separate signaling branch to ROCK and LIMK to prevent chromatin breakage (Fig. 3P,Q). These results suggest that mDia1 cooperates with the ROCK-LIMK signaling axis downstream of RhoA to prevent chromatin bridge breakage in cytokinesis (Fig. 3R).

## PDZ-RhoGEF activates RhoA to promote actin patch formation

Because Y16 (RhoAi) selectively inhibits the regulator of G-protein signaling (RGS) domain-containing family of RhoGEFs comprising ArhGEF1/p115-RhoGEF, ArhGEF11/PDZ RhoGEF/GTRAP48 and ArhGEF12/LARG, we investigated a potential role of the above RhoGEFs for stable chromatin bridges in cytokinesis (Shang et al, 2013). Depletion of PDZ, but not p115 or LARG, RhoGEF by siRNA induced chromatin bridge breakage compared with control cells, indicating that PDZ is required for stable chromatin bridges (Fig. 4A; Appendix Fig. S3I–K). PDZ localized to actin patches in control cells (Fig. 4B). Depletion of PDZ by siRNA (siPDZ) diminished actin patches and impaired localization of RhoA to the base of the intercellular canal compared with controls (Fig. 4B–E). Furthermore, expression of Citrine:PDZ^Res resistant to degradation by PDZ siRNA-2 (siPDZ-2), but not of GFP-only, rescued actin patches and prevented chromatin bridge breakage in cytokinesis after PDZ-depletion compared with GFP controls (Appendix Fig. S3L–R). To examine Rho activity, we used the dimericTomato-2xrGBD (dT-2xrGBD) Rho biosensor consisting of a dimericTomato fluorescence protein and a double rhotekin G protein-binding domain (Mahlandt et al, 2021). dT-2xrGBD localized to actin patches at the base of the intercellular canal in control cells with chromatin bridges and this localization was

impaired in PDZ-deficient cells, indicating that localization of active Rho to actin patches is PDZ-dependent (Appendix Fig. S4A–C). Also, expression of a constitutively active mutant GFP:RhoA-G14V harboring mutation of RhoA-glycine 14 to valine that makes RhoA resistant to GTP-hydrolysis (Morin et al, 2009), but not GFP-only, rescued actin patches and prevented broken chromatin bridges after PDZ-depletion compared with GFP controls (Fig. 4F–J; Appendix Fig. S4D). These results suggest that PDZ RhoGEF activates RhoA to promote actin patch formation and prevent chromatin bridge breakage in cytokinesis.

## RhoA functions upstream of Src-FAK in actin patch formation

The actin regulators Src and FAK are also required for actin patches (Dandoulaki et al, 2018). Simultaneous depletion of RhoA and Src by siRNAs (siRhoA+siSrc) did not exacerbate chromatin bridge breakage in cytokinesis compared with RhoA- or Src-single depleted cells, suggesting that RhoA and Src-FAK operate in the same signaling pathway (Fig. 4K,L; Appendix Fig. S4E,F). Expression of GFP-fused constitutively active Src-S51D, in which Src-serine 51 was mutated to aspartic acid (Dandoulaki et al, 2018), or constitutively active FAK-Y397E, in which FAK-tyrosine 397 was changed to glutamic acid (Schaller et al, 1994), but not GFP-only, rescued actin patches and prevented breakage of chromatin bridges after RhoA-depletion compared with GFP controls (Fig. 4M,N; Appendix Fig. S4G–N), indicating that RhoA functions upstream of Src and FAK in actin patch formation. In contrast, after depletion of Src by siRNA, expression of constitutively active GFP:RhoA-G14V did not rescue actin patches and stable chromatin bridges compared with controls expressing GFP (Appendix Figs. S4O and S5A,B), which is consistent with Src functioning downstream of RhoA. These results show that RhoA functions upstream of Src-FAK signaling during actin patch formation in cytokinesis with chromatin bridges.

## Integrin signaling is not required for actin patch formation

Focal adhesion proteins, such as integrins, can modulate the actin cytoskeleton by controlling the activity of Rho-family GTPases and by regulating Src-FAK signaling (Barry et al, 1997; Gimond et al, 1999; Mitra and Schlaepfer, 2006). Confocal microscopy analysis of

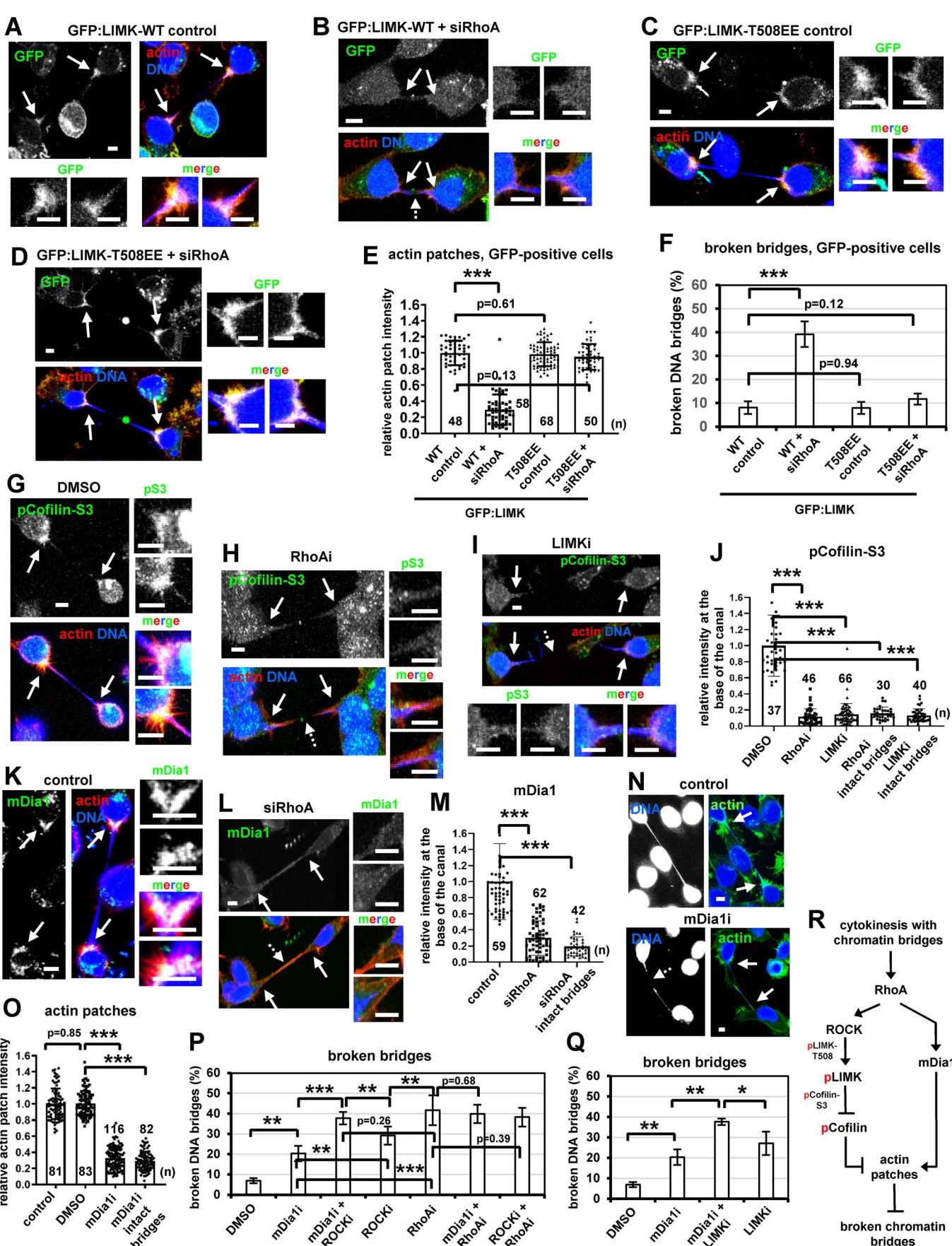

**Figure 3.   Expression of phosphomimetic T508EE GFP:LIMK rescues actin patches in RhoA-deficient cells.**

(A–D) Actin patches in BE cells expressing wild-type (WT) or GFP:LIMK-T508EE in the absence or presence of RhoA siRNA (siRhoA). (E) Actin patches intensity in GFP-positive cells. Mean ± SD from n cells from two independent experiments. Values in GFP:LIMK WT control were set to 1. ***P = 7.35E-39 (WT control vs WT+siRhoA) by ANOVA and Student's t test. (F) Percentage of broken DNA bridges in GFP-positive cells. Mean ± SD from three independent experiments (n = 62, 55, 89, 78). ***P = 0.00028 (WT control vs WT+siRhoA) by ANOVA and Student's t test. (G–I) Localization of phosphorylated Cofilin-serine 3 (pS3) in cells treated with 50 μM DMSO, 50 μM Y16 (RhoAi) for 30 min, or with 10 μM TH-257 (LIMKi) for 5 h. (J) Phospho-Cofilin-S3 fluorescence intensity. Mean ± SD from n cells from two independent experiments. Values in DMSO were set to 1. ***P = 2.50E-25 (DMSO vs RhoAi), 1.77E-30 (DMSO vs LIMKi), 5.88E-18 (DMSO vs RhoAi intact bridges), 1.06E-22 (DMSO vs LIMKi intact bridges) by ANOVA and Student's t test. (K, L) Localization of mDia1 in the absence (control) or presence of siRhoA. (M) mDia1 fluorescence intensity. Mean ± SD from n cells from three independent experiments. Values in control were set to 1. ***P = 6.36E-19 (control vs siRhoA), 2.00E-18 (control vs siRhoA intact bridges) by ANOVA and Student's t test. (N) Actin patches and chromatin bridges in control cells or cells treated with 25 μM SMIFH2 (mDia1i) for 5 h. Intact arrows indicate actin patches and/or canal bases. Broken arrows indicate broken DNA bridges. Insets show high magnifications of the canal bases. Scale bars, 5 μm. (O) Actin patches intensity. Mean ± SD from n cells from two independent experiments. Values in control were set to 1. Numbers below/next to each bar indicate n. ***P = 8.66E-82 (DMSO vs mDia1i), 1.06E-74 (DMSO vs mDia1i intact bridges) by ANOVA and Student's t test. (P) Percentage of broken DNA bridges. Mean ± SD from three independent experiments (n = 56, 92, 116, 144, 146, 65, 65). ***P = 4.28E-05 (mDia1i vs mDia1i+ROCKi), 0.00075 (mDia1i vs RhoAi); **P = 0.0022 (DMSO vs mDiai1), 0.0078 (mDia1i vs ROCKi), 0.0043 (mDia1+ROCKi vs ROCKi), 0.0068 (ROCKi vs RhoAi) by ANOVA and Student's t test. (Q) Percentage of broken DNA bridges. Mean ± SD from three independent experiments (n = 56, 92, 60, 142). **P = 0.0022 (DMSO vs mDiai1), 0.0010 (mDia1 vs mDia1+LIMKi); *P = 0.028 (mDia1+LIMKi vs LIMKi) by ANOVA and Student's t test. (R) Proposed RhoA signaling pathways that promote actin patch formation in cytokinesis with chromatin bridges. p phosphorylation. Source data are available online for this figure.

individual image planes showed that actin patches do not contain the focal adhesion proteins integrin or paxillin (Fig. 4O–Q; Appendix Fig. S5C,E,F). Instead, focal adhesions were detected near the bottom of the cell (i.e., close to the substrate) at an average distance of ~6.0 ± 1.0 μm below the main image plane containing the actin patches (Appendix Fig. S5D), and the average hight of BE cells with chromatin bridges (distance from top to bottom image planes) was approximately 12 ± 1.5 μm (n = 30). These results show that actin patches are formed near the middle of the cell and do not contain focal adhesions; furthermore, a 3D reconstruction of image stacks showed that actin patches are all around the base of the chromatin bridge (Movie EV19). Also, inhibition of integrin signaling by the small molecule inhibitor SB273005 (integrin i) at a concentration that diminished focal adhesions by staining for paxillin, did not reduce actin patches and did not promote chromatin bridge breakage in cytokinesis compared with control cells (Fig. 5A,B; Appendix Fig. S5G–I). These results show that integrin signaling is not required for actin patch formation.

## Cell nuclei with chromatin bridges are under mechanical tension

Control cells with intact chromatin bridges exhibited deformed (oval-shaped) nuclei with a longer (x) axis extending parallel to the DNA bridge and a shorter (y) axis perpendicular to the bridge (x/y > 1) compared with interphase cells or telophase cells in late cytokinesis without chromatin bridges that were not elongated in any specific direction (x/y ≈ 1), by high resolution confocal microscopy analysis of nuclear chromatin (DNA) or nuclear lamina (Lamin A staining; Fig. 5C,D,F,G,I; Appendix Fig. S5J–L,N,O). Control cells with chromatin bridges also exhibited pronounced linear arrays ("nuclear lines") of Sun1, Sun2 and Lamin A proteins, indicating nuclear membrane deformations caused by mechanical forces (Fig. 5J–N; Appendix Figs. S5N,P and S6A–C) (Luxton et al, 2010; Smith et al, 2022; Versaevel et al, 2014). Furthermore, relatively faint actin nuclear lines that colocalized with the Lap2b lines were observed in fixed HeLa Lap2b:RFP/Lifeact:GFP cells with chromatin bridges (Appendix Fig. S6D) (Luxton et al, 2010; Smith et al, 2022; Versaevel et al, 2014). Cells with broken chromatin bridges after depletion of RhoA

by siRNA exhibited reduced nuclear chromatin shape-deformation compared with controls or RhoA-depleted cells with intact DNA bridges (Fig. 5O), showing that relatively strong nuclear deformations require intact DNA bridges. Control cells with chromatin bridges also exhibited accumulation of Sun1, Sun2 and Nesprin-2 nuclear envelope proteins at the base of chromatin bridges (nuclear "front"), compared with interphase or late cytokinesis cells without chromatin bridges showing even distribution of the above proteins at the periphery of the nucleus; furthermore, the accumulation of Nesprin-2 was confirmed using two different antibodies against amino acids 4200–4350 or amino acids 1–300 (Ab2; Fig. 5L,P–T; Appendix Figs. S5J and S6A,B,E–I). Depletion of Sun1/2 by siRNA, or transfection of cells with a dominant-negative KASH domain (dnKASH) fused to mCherry impaired Nesprin-2 accumulation at the front of the nucleus compared with controls in cytokinesis with intact chromatin bridges (Fig. 5R,S,U,V; Appendix Fig. S6F–I), indicating that Nesprin-2 forms a complex with Sun1/2 at the front of the nucleus. These results suggest that pulling forces exerted on daughter nuclei by intact chromatin bridges cause mechanical stress ("nuclear tension") and accumulation of the Sun1/2-Nesprin-2 LINC complex at the base of chromatin bridges.

## The Sun1/2-Nesprin-2 LINC complex is required for nuclear tension

Depletion of Sun1/2 or Nesprin-2, or transfection of cells with dnKASH fused to mCherry reduced nuclear chromatin shape-deformation and Sun2 nuclear lines compared with controls in cytokinesis with intact chromatin bridges (Fig. 5I,K–N). Furthermore, LINC-deficient cells exhibited increased frequency of intact "loose" DNA bridges, which is consistent with weakened chromatin bridge-pulling forces, compared with controls exhibiting stretched DNA bridges (Fig. 5C,E,F,H,U–X; Appendix Fig. S6J–M). Perhaps of note, inhibition of LINC did not completely abolish nuclear chromatin shape-deformation compared with cells without DNA bridges, showing that LINC-deficient cells with intact chromatin bridges can exhibit relatively low nuclear tension in our experimental conditions (Fig. 5I). These results show that the Sun1/2-Nesprin-2 LINC complex is required for optimal nuclear tension in cytokinesis with chromatin bridges. In contrast, RhoA-deficient

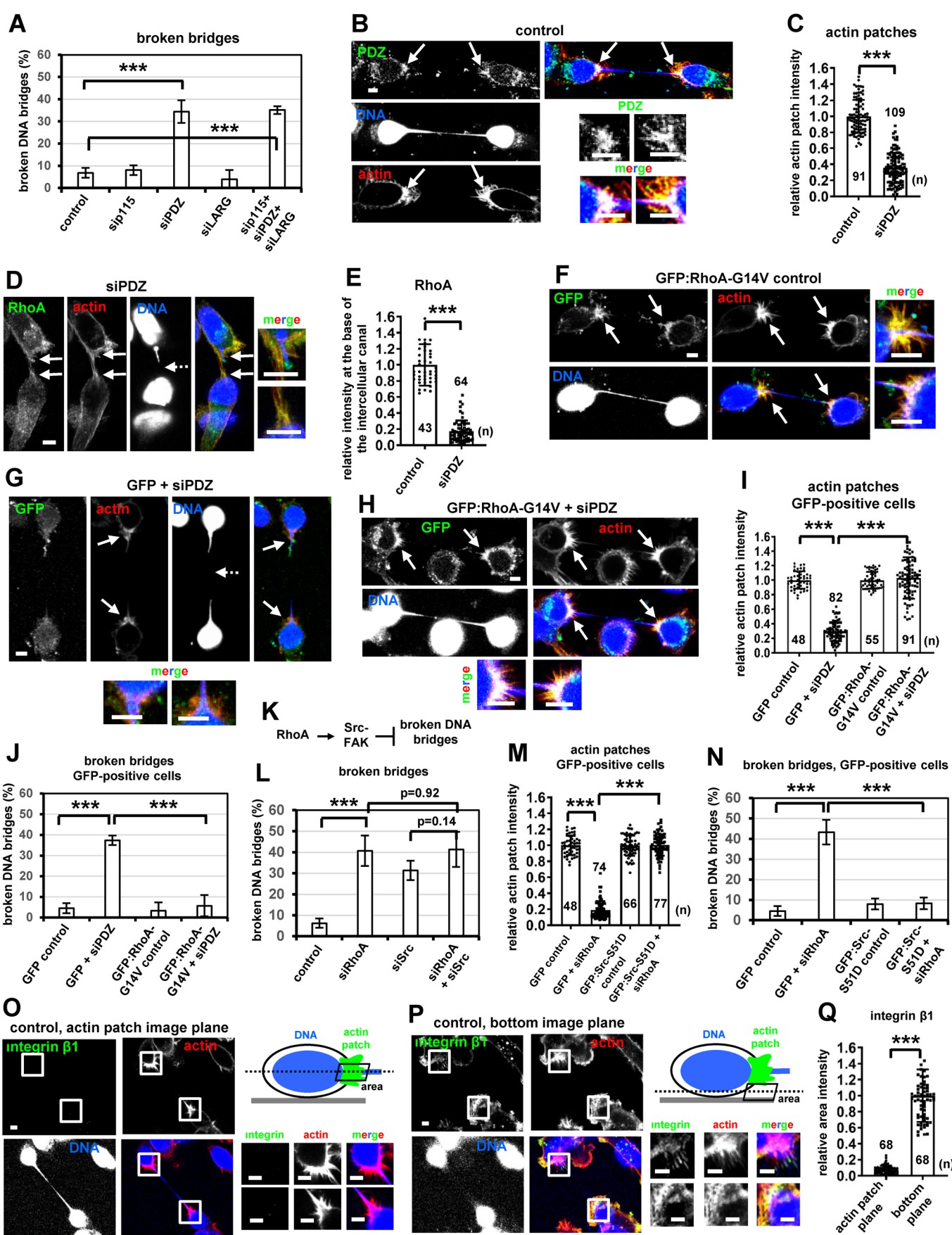

◀ **Figure 4. Inhibition of PDZ RhoGEF correlates with chromatin bridge breakage in cytokinesis.**

(A) Percentage of broken DNA bridges in control BE cells, or cells transfected with p115 siRNA (sip115), PDZ RhoGEF siRNA (siPDZ) or LARG siRNA (siLARG). Mean ± SD from three independent experiments ($n = 72, 53, 57, 53, 66$). ***$P = 0.0010$ (control vs siPDZ), 6.51E-05 (control vs sip115+siPDZ+siLARG) by ANOVA and Student's t test. (B) Localization of PDZ RhoGEF. (C) Actin patches intensity. Mean ± SD from $n$ cells from two independent experiments. Values in control were set to 1. ***$P = 4.21$E-58 by Student's t test. (D) Localization of RhoA. (E) RhoA fluorescence intensity. Mean ± SD from $n$ cells from three independent experiments. Values in control were set to 1. ***$P = 1.88$E-40 by Student's t test. (F–H) Actin patches and chromatin bridges in cells expressing mutant GFP:RhoA-G14V or GFP-only, in the absence or presence of siPDZ. Intact arrows indicate actin patches and/or canal bases. Broken arrows indicate broken DNA bridges. Insets show high magnifications of the canal bases. (I) Actin patches intensity in GFP-positive cells from F-H. Mean ± SD from $n$ cells from two independent experiments. Values in GFP were set to 1. ***$P = 1.28$E-61 (GFP control vs GFP+siPDZ), 2.43E-50 (GFP+siPDZ vs GFP:RhoA-G14V+siPDZ) by ANOVA and Student's t test. (J) Percentage of broken DNA bridges in GFP-positive cells. Mean ± SD from three independent experiments ($n = 125, 46, 83, 50$). ***$P = 2.42$E-07 (GFP control vs GFP+siPDZ), 0.00062 (GFP+siPDZ vs GFP:RhoA-G14V+siPDZ) by ANOVA and Student's t test. (K) Schematic diagram of RhoA-Src-FAK signaling preventing chromatin bridge breakage. (L) Percentage of broken DNA bridges in control cells, or cells transfected with RhoA siRNA (siRhoA) or Src siRNA (siSrc). Mean ± SD from three independent experiments ($n = 95, 63, 67, 75$). ***$P = 0.00025$ (control vs siRhoA) by ANOVA and Student's t test. (M) Actin patches intensity in GFP-positive cells expressing GFP:Src-S51D or GFP-only in the absence or presence of siRhoA. Mean ± SD from n cells from two independent experiments. Values in GFP control were set to 1. ***$P = 1.84$E-69 (GFP control vs GFP+siRhoA), 7.27E-83 (GFP+siRhoA vs GFP:Src-S51D+siRhoA) by ANOVA and Student's t test. (N) Percentage of broken DNA bridges in GFP-positive cells. Mean ± SD from three independent experiments ($n = 125, 89, 62, 60$). ***$P = 5.29$E-07 (GFP control vs GFP+siRhoA), 0.00026 (GFP+siRhoA vs GFP:Src-S51D+siRhoA) by ANOVA and Student's t test. (O, P) Integrin β1 localization at the actin patch image plane or at the cell bottom image plane of the same cells, grown on fibronectin-coated slides. Dotted lines indicate the depicted image planes; rectangles indicate areas magnified in insets. (Q) Integrin β1 intensity. Mean ± SD from $n$ cells from two independent experiments. Values in bottom plane were set to 1. Numbers below/next to each bar indicate $n$. ***$P = 2.011$E-48 by Student's t test. Scale bars, 5 μm. Source data are available online for this figure.

cells with intact chromatin bridges exhibited Sun2 nuclear lines, chromatin shape-deformation and relatively low frequency of loose DNA bridges similar to controls (Fig. 5N,O,X), showing that RhoA is not required for nuclear tension.

## The Sun1/2-Nesprin-2 LINC complex is required for actin patch formation

Cells depleted of Sun1/2 or Nesprin-2 by siRNA, or transfected with dnKASH, but not cells expressing KASH-ΔL lacking the luminal SUN protein-binding KASH domain (Luxton et al, 2010), exhibited reduced actin patches at the base of the intercellular canal and increased frequency of broken chromatin bridges in cytokinesis compared with controls (Fig. 5R,U–W,Y,Z; Appendix Fig. S6J,N–T). Furthermore, 19/48 (40%) BE cells depleted of Nesprin-2 exhibited breakage of DNA bridges and intercellular canals after 44 ± 26 min (Appendix Fig. S7A,B; Movie EV20), by live-cell microscopy. Depletion of Sun1/2 also diminished localization of RhoA at the base of the intercellular canal, but not total RhoA protein levels, compared with controls (Fig. 6A–D; Appendix Fig. S7C). Furthermore, expression of constitutively active GFP:RhoA-G14V, but not GFP-only, rescued actin patches and prevented chromatin bridge breakage in cells depleted of Sun1/2 compared with GFP controls, showing that RhoA acts downstream of Sun1/2 in actin patch formation (Fig. 6E,F; Appendix Fig. S7D,E). However, expression of GFP:RhoA-G14V did not prevent loose chromatin bridges in cytokinesis in Sun1/2-depleted cells compared with GFP controls (Appendix Fig. S7F), further supporting that RhoA activity is not required for nuclear tension. These results suggest that the Sun1/2-Nesprin-2 LINC complex promotes actin patches by activating RhoA signaling in cytokinesis with chromatin bridges.

## The Nesprin-2 SRs are required for actin patch formation

To dissect the roles of the LINC complex in nuclear tension and actin patch formation, we used a mouse mini-Nesprin-2 protein (CB) that lacks SRs 3-54, but can still bind to the actin cytoskeleton through its CH domain (Fig. 6G) (Déjardin et al, 2020; Luxton et al, 2010; Ostlund et al, 2009). Mouse mini-Nesprin-2 is resistant to

human Nesprin-2 siRNA because the mouse sequence is sufficiently diverse from the human sequence. After depletion of the endogenous protein, cells with intact chromatin bridges expressing CB exhibited nuclear chromatin shape-deformation, relatively high frequency of Sun2 nuclear lines and low frequency of loose DNA bridges, similar to GFP controls (Fig. 6H–J; Appendix Fig. S7G–I), suggesting that SRs 3-54 are dispensable for nuclear tension. However, cells expressing CB exhibited impaired actin patches and increased frequency of broken chromatin bridges in cytokinesis compared with GFP controls (Fig. 6K–M; Appendix Fig. S7J,K). These results show that Nesprin-2 SRs 3-54 are required for actin patch formation and stable chromatin bridges, but not for nuclear tension in cytokinesis.

## Nesprin-2 interaction with the actin cytoskeleton is required for nuclear tension

To further investigate how nuclear tension is generated, cells were transfected with an siRNA-resistant mini-Nesprin-2 variant (CH*) bearing Ile128, 131Ala mutations in the calponin homology domain that impair binding to actin (Fig. 6G) (Déjardin et al, 2020; Luxton et al, 2010). After depletion of the endogenous Nesprin-2, in cells with intact chromatin bridges, expression of CH* reduced (but did not completely abolish) nuclear chromatin shape-deformation, reduced Sun2 nuclear lines, and increased the frequency of loose chromatin bridges compared with GFP controls, showing that Nesprin-2 binding to the actin cytoskeleton is required for optimal nuclear tension (Fig. 6H–J,N; Appendix Fig. S7L). Furhermore, cells expressing CH* exhibited impaired actin patches and increased chromatin bridge breakage in cytokinesis compared with GFP controls (Fig. 6L,M,O). Perhaps of note, after Nesprin-2 depletion, cells expressing CB, i.e., with relatively high nuclear tension, exhibited increased frequency of broken chromatin bridges compared with cells expressing CH* or GFP (Fig. 6M), which is consistent with high nuclear tension causing chromatin bridges to break more often (Umbreit et al, 2020). These results show that interaction of Nesprin-2 with the actin cytoskeleton is required for optimal nuclear tension and actin patch formation in cytokinesis with chromatin bridges.

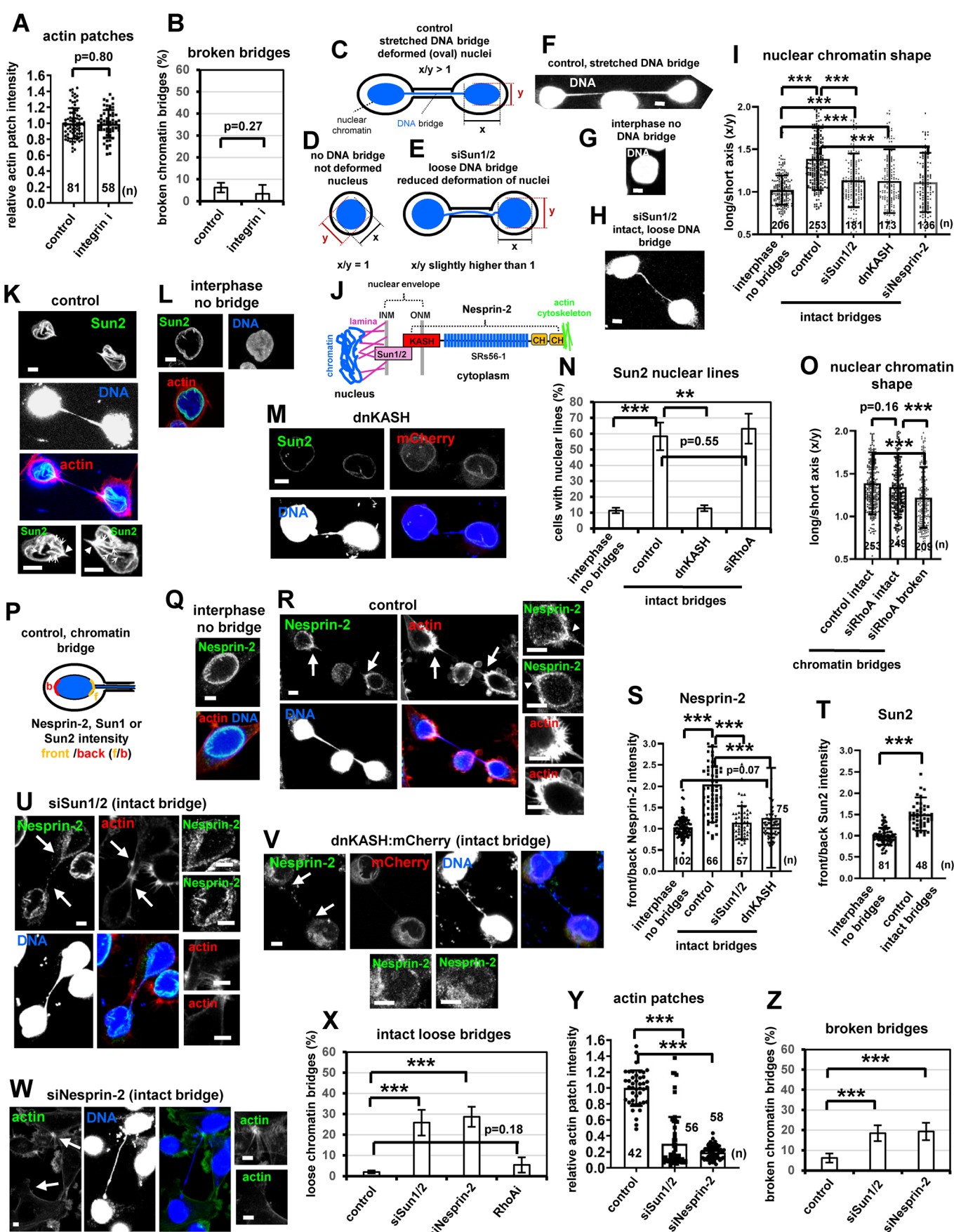

**Figure 5. Depletion of Sun1/2 or Nesprin-2 reduces nuclear chromatin shape-deformation, and correlates with impaired actin patches and broken chromatin bridges in cytokinesis.**

(A) Actin patches intensity in control BE cells or cells treated with 1 µM SB273005 (integrin i) for 15 min. Mean ± SD from $n$ cells from two independent experiments. Values in control were set to 1. $P = 0.80$ by Student's $t$ test. (B) Percentage of broken DNA bridges. Mean ± SD from four independent experiments ($n = 95, 53$). $P = 0.27$ by Student's $t$ test. (C–E) Cartoons of cells exhibiting deformed nuclei. (F–H) DNA staining in cells with stretched or loose chromatin bridges, or without DNA bridges. (I) Nuclear chromatin shape in interphase cells without chromatin bridges, untransfected (control) cells with intact chromatin bridges, or cells transfected with Sun1/2 siRNA (siSun1/2), dominant-negative KASH (dnKASH) or Nesprin-2 siRNA (siNesprin-2) with intact chromatin bridges. Mean ± SD from $n$ cells from two independent experiments. ***$P = 4.04E-34$ (interphase no bridges vs control), 7.89E-06 (interphase no bridges vs siSun1/2), 0.00037 (interphase no bridges vs dnKASH), 4.14E-13 (control vs siSun1/2), 2.46E-12 (control vs siNesprin-2) by ANOVA and Student's $t$ test. (J) Cartoon of the Sun1/2-Nesprin-2 LINC complex. SRs, spectrin repeats; INM/ONM, inner/outer nuclear membrane; CH, calponin homology. (K–M) Sun2 localization. (N) Percentage of cells exhibiting Sun2 nuclear lines. Cells were transfected with RhoA siRNA (siRhoA) or treated as in (I). Mean ± SD from three independent experiments ($n = 213, 79, 69, 72$). ***$P = 0.00098$ (interphase no bridges vs control); **$P = 0.0011$ (control vs dnKASH) by Student's $t$ test. (O) Nuclear chromatin shape in cells with intact or broken chromatin bridges. Mean ± SD from n cells from two independent experiments. ***$P = 1.15E-06$ (control intact vs siRhoA broken), 0.00032 (siRhoA intact vs siRhoA broken) by ANOVA and Student's $t$ test. (P) Cartoon showing the front and back of the nucleus where Nesprin-2, Sun1 or Sun2 intensity was measured. (Q, R) Nesprin-2 localization. (S) Front/back Nesprin-2 intensity. Mean ± SD from $n$ cells from two independent experiments. ***$P = 3.71E-21$ (interphase no bridges vs control), 1.25E-10 (control vs siSun1/2), 1.91E-05 (control vs dnKASH) by ANOVA and Student's $t$ test. (T) Front/back Sun2 intensity. Mean ± SD from $n$ cells from two independent experiments. ***$P = 6.29E-18$ by Student's $t$ test. (U, V) Nesprin-2 localization. Arrowheads show Sun2 nuclear lines or Nesprin-2 accumulation. (W) Actin patches. Intact arrows indicate actin patches and/or canal bases. Insets show high magnifications of the canal bases. Scale bars, 5 µm. (X) Percentage of cells exhibiting intact loose chromatin bridges. Mean ± SD from three independent experiments ($n = 188, 138, 113, 74$). ***$P = 7.29E-05$ (control vs siSun1/2), 0.00025 (control vs siNesprin-2) by ANOVA and Student's $t$ test. (Y) Actin patches intensity. Mean ± SD from $n$ cells from two independent experiments. Values in control were set to 1. Numbers below/next to each bar indicate n. ***$P = 5.48E-20$ (control vs siSun1/2), 1.004E-44 (control vs siNesprin-2) by ANOVA and Student's $t$ test. (Z) Percentage of cells exhibiting broken chromatin bridges. Mean ± SD from four independent experiments ($n = 95, 138, 138$). ***$P = 0.00013$ (control vs siSun1/2), 0.00086 (control vs siNesprin-2) by ANOVA and Student's $t$ test. Source data are available online for this figure.

## An intact nuclear lamina is required for nuclear tension and actin patch formation

Depletion of Lamins A and C, but not RhoA, by siRNA (siLamin A/C) reduced nuclear chromatin shape-deformation, increased the frequency of loose DNA bridges and impaired accumulation of Nesprin-2 at the base of chromatin bridges compared with controls in cytokinesis with intact chromatin bridges (Fig. 6P–R; Appendix Fig. S7M–P). Furthermore, Lamin A/C-deficient cells exhibited impaired actin patches compared with controls (Fig. 6S). Expression of constitutively active GFP:RhoA-G14V, but not GFP-only, rescued actin patches and prevented chromatin bridges from breaking in cells depleted of Lamin A/C compared with GFP controls (Fig. 7A–F), indicating that RhoA functions downstream of Lamins A/C in actin patch formation. However, expression of GFP:RhoA-G14V did not restore nuclear chromatin shape-deformation and did not prevent intact loose chromatin bridges after Lamin A/C-depletion compared with GFP controls (Appendix Fig. S8A,B). Treatment of cells with the myosin II inhibitor blebbistatin impaired nuclear shape deformation, Nesprin-2 accumulation and actin patches in cytokinesis compared with controls, suggesting that actomyosin forces are required for nuclear tension and actin patch formation (Appendix Fig. S8C–F) (Kovacs et al, 2004; Ramamurthy et al, 2004). Because treatment with the mDia1i SMIFH2 did not attenuate nuclear shape deformation (Appendix Fig. S8F), these results also argue against mDia1i reducing actin patches by inhibiting myosin II (Nishimura et al, 2021) and further support a role for mDia1 in actin patch formation. In conclusion, these results suggest that an intact nuclear lamina and actomyosin forces are required for optimal nuclear tension, Nesprin-2 accumulation, and RhoA-dependent actin patch formation in cytokinesis with chromatin bridges.

## The Nesprin-2 SRs bind to PDZ RhoGEF

To map the Nesprin-2 SRs required for actin patch formation, SRs 3-11, 12-20, 21-30, 31-37, 38-47 and 48-54 of human Nesprin-2 were isolated from BE cells by RT PCR, cloned into a GST-coding vector and the corresponding GST-tagged proteins were purified from bacteria. We found that GST-SRs 12-20, 21-30 and 31-37, but not GST-only, exhibited relatively strong binding to transfected PDZ RhoGEF fused to Citrine (Citrine:PDZ) or to endogenous PDZ RhoGEF from cell extracts, whereas GST-SRs 3-11, 38-47 and 48-54 associated with Citrine:PDZ relatively more weakly by GST pull-downs (Fig. 7G; Appendix Fig. S8G). Recombinant PDZ RhoGEF generated by in vitro transcription-translation associated with GST-SRs 31-37, but not with GST-only (Fig. 7H); furthermore, endogenous Nesprin-2 associated with PDZ RhoGEF by coimmunoprecipitation experiments in cell extracts (Appendix Fig. S8H). Also, GFP-tagged proteins containing the RGS-homology (RH) or the Dbl homology/pleckstrin homology (DH/PH) domains of PDZ RhoGEF, but not the PDZ domain (PDZ$^{dom}$) or the C-terminal (C-term) region, associated with GST-SRs 31-37 by GST pull-downs (Appendix Fig. S8I–K). These results suggest that the Nesprin-2 SRs bind to PDZ RhoGEF at the RH and DH/PH domains. Perhaps interestingly, the mammalian formin FHOD1 (Formin Homology 2 Domain Containing 1), which is a target of ROCK and can interact with Nesprin-2 under specific conditions (Luxton et al, 2010; Takeya et al, 2008), is also found at actin patches in control cells (Appendix Fig. S8L).

## Targeting RhoA to Nesprin-2 cytoplasmic region promotes actin patches in the presence of nuclear tension

To investigate the significance of Nesprin-2 SRs-PDZ interaction, the Nesprin-2 SRs 31-37 that bind to PDZ RhoGEF, the wild-type DH/PH domain of PDZ RhoGEF that interacts with RhoA, or a mutant DH/PH* harboring a point mutation (E15A) (Lenoir et al, 2014) that diminishes binding of PDZ RhoGEF DH/PH to active RhoA were inserted in the cytoplasmic region of siRNA-resistant mini-Nesprin-2 CB or CH* proteins to generate CB:SRs31-37, CB:DHPH, CB:DHPH* and CH*:DHPH fusion proteins

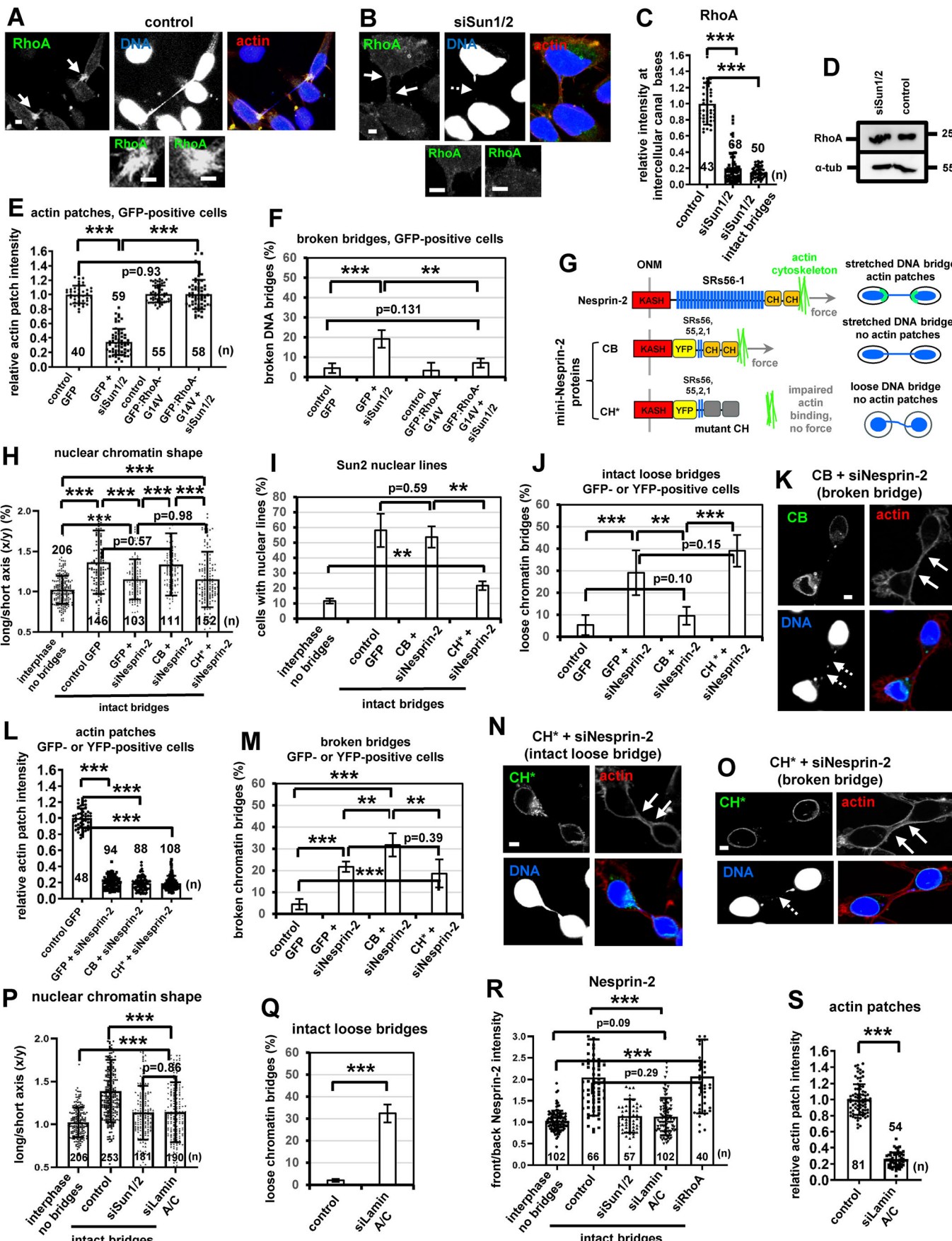

**Figure 6.** Expression of mini-Nesprin-2 CB protein impairs actin patches and correlates with chromatin bridge breakage in cytokinesis.

(A, B) RhoA localization in control BE cells or cells transfected with Sun1/2 siRNA (siSun1/2). Intact arrows indicate actin patches and/or canal bases. Broken arrows indicate broken DNA bridges. Scale bars, 5 μm. (C) RhoA intensity. Mean ± SD from n cells from two independent experiments. Values in control were set to 1. ***$P$ = 1.01E-35 (control vs siSun1/2), 1.45E-38 (control vs siSun1/2 intact bridges) by ANOVA and Student's $t$ test. (D) Western blot analysis of total RhoA or α-tubulin (α-tub). (E) Actin patches intensity in GFP-positive cells. Cells were transfected with GFP:RhoA-G14V or GFP in the absence (control) or presence of siSun1/2. Mean ± SD from $n$ cells from two independent experiments. Values in control GFP were set to 1. ***$P$ = 2.83E-36 (control GFP vs GFP+siSun1/2), 1.79E-36 (GFP+siSun1/2 vs GFP:RhoA-G14V +siSun1/2) by ANOVA and Student's $t$ test. (F) Percentage of broken DNA bridges in GFP-positive cells. Mean ± SD from three independent experiments ($n$ = 125, 73, 83, 85). ***$P$ = 0.00029; **$P$ = 0.0051 by ANOVA and Student's $t$ test. (G) Cartoons of Nesprin-2, mini-Nesprin-2 CB and CH* proteins. SRs spectrin repeats, ONM outer nuclear membrane, CH calponin homology. (H) Nuclear chromatin shape in cells expressing mini-Nesprin-2 CB or CH* proteins in the absence or presence of Nesprin-2 siRNA (siNesprin-2). Mean ± SD from n cells from two independent experiments. ***$P$ = 1.01E-24 (interphase no bridges vs control GFP), 1.33E-07 (interphase no bridges vs GFP+siNesprin-2), 3.46E-06 (interphase no bridges vs CH*+siNesprin-2), 2.63E-06 (control GFP vs GFP+siNesprin-2), 5.98E-05 (GFP+siNesprin-2 vs CB+siNesprin-2), 5.92E-05 (CB+siNesprin-2 vs CH*+siNesprin-2) by ANOVA and Student's $t$ test. (I) Percentage of cells exhibiting Sun2 nuclear lines. Mean ± SD from three independent experiments ($n$ = 215, 64, 52, 65). **$P$ = 0.0062 (interphase no bridges vs CH*+siNesprin-2), 0.0018 (CB+siNesprin-2 vs CH*+siNesprin-2) by ANOVA and Student's $t$ test. (J) Percentage of intact loose DNA bridges in GFP- or YFP-positive cells. Mean ± SD from three independent experiments ($n$ = 143, 70, 102, 107). ***$P$ = 0.00016 (control GFP vs GFP+siNesprin-2), 1.27E-05 (CB+siNesprin-2 vs CH*+siNesprin-2); **$P$ = 0.0035 by ANOVA and Student's $t$ test. (K) Chromatin bridges and actin patches in cells expressing mini-Nesprin-2 CB. (L) Actin patches intensity in GFP- or YFP-positive cells. Values in control GFP were set to 1. Mean ± SD from n cells from two independent experiments. ***$P$ = 4.97E-88 (control GFP vs GFP+siNesprin-2), 1.05E-83 (control GFP vs CB+siNesprin-2), 1.63E-93 (control GFP vs CH*+siNesprin-2) by ANOVA and Student's $t$ test. (M) Percentage of broken DNA bridges in GFP- or YFP-positive cells. Mean ± SD from four independent experiments ($n$ = 125, 92, 84, 107). ***$P$ = 3.83E-06 (control GFP vs GFP+siNesprin-2), 1.34E-06 (control GFP vs CB+siNesprin-2), 0.00072 (control GFP vs CH*+siNesprin-2); **$P$ = 0.009 (GFP+siNesprin-2 vs CB+siNesprin-2), 0.0079 (CB+siNesprin-2vs CH*+siNesprin-2) by ANOVA and Student's $t$ test. (N, O) Chromatin bridges and actin patches in cells expressing mini-Nesprin-2 CH* protein. Intact arrows indicate actin patches and/or canal bases. Broken arrows indicate broken DNA bridges. Insets show high magnifications of the canal bases. Scale bars, 5 μm. (P) Nuclear chromatin shape in cells transfected with Lamin A/C siRNA (siLamin A/C), or as in (A, B). Mean ± SD from n cells from two independent experiments. ***$P$ = 1.28E-05 (interphase no bridges vs siLaminA/C), 4.44E-12 (control vs siLaminA/C) by ANOVA and Student's $t$ test. (Q) Percentage of intact loose DNA bridges. Mean ± SD from three independent experiments ($n$ = 188, 77). ***$P$ = 0.00021 by ANOVA and Student's $t$ test. (R) Front/back Nesprin-2 intensity. Mean ± SD from n cells from two independent experiments. ***$P$ = 4.34E-21 (interphase no bridges vs siRhoA), 1.24E-15 (control vs siLaminA/C) by ANOVA and Student's $t$ test. (S) Actin patches intensity. Mean ± SD from $n$ cells from two independent experiments. Values in control were set to 1. Numbers below/next to each bar indicate n. ***$P$ = 8.75E-56 by Student's $t$ test. Source data are available online for this figure.

(Appendix Fig. S8M). After depletion of the endogenous Nesprin-2, expression of CB:SRs31-37 or CB:DHPH, but not mutant CB:DHPH* or mCherry-tagged DHPH-only, rescued actin patches and prevented chromatin bridge breakage in cytokinesis compared with GFP controls (Fig. 7I,J,L,M,O; Appendix Fig. S8N–P). Furthermore, cells with intact chromatin bridges expressing CB:SRs31-37, CB:DHPH or CB:DHPH* exhibited nuclear chromatin-shape deformation and relatively low frequency of loose chromatin bridges similar to cells expressing mini-Nesprin-2 CB or to GFP controls, indicating that these cells were under nuclear tension (Fig. 7K; Appendix Fig. S9A). In contrast, cells expressing CH*:DHPH exhibited impaired actin patches and increased chromatin bridge breakage; furthermore, cells with intact chromatin bridges expressing CH*:DHPH showed reduced nuclear chromatin shape-deformation and increased frequency of loose DNA bridges compared with GFP controls, indicating reduced nuclear tension (Fig. 7I–K,N; Appendix Fig. S9A). These results suggest that targeting RhoA to the cytoplasmic region of Nesprin-2 promotes actin patch formation and prevents chromatin bridge breakage in cytokinesis in the presence of nuclear tension.

## Nuclear tension correlates with Nesprin-2 accumulation at the base of chromatin bridges

After depletion of the endogenous protein, cells with intact chromatin bridges expressing mini-Nesprin-2 CB or CB:DHPH, which are under relatively high nuclear tension, exhibited increased accumulation of mini-Nesprin-2 proteins (YFP) at the "front" of the nucleus (i.e., at the base of chromatin bridges) compared with cells expressing CH* or CH*:DHPH, which are under reduced nuclear tension (Fig. 7M,N; Appendix Fig. S9B–D). These results suggest that nuclear tension correlates with Nesprin-2 accumulation at the base of chromatin bridges in cytokinesis.

## Accumulation of Nesprin-2 at the base of chromatin bridges promotes actin patch formation

To investigate a potential role for Nesprin-2 accumulation in actin patch formation, we interfered with the localization of Nesprin-2 by using two approaches: First, cells were transfected with Lamin A fused to RFP (Lamin A:RFP). Cells overexpressing Lamin A:RFP (with or without chromatin bridges) exhibited Lamin A:RFP granules at the nuclear periphery and inside the nucleus (Fig. 8A,D,E,L) (Libotte et al, 2005), compared with untransfected controls or cells expressing relatively low levels of Lamin A:RFP that exhibit a Lamin A nuclear ring (Appendix Figs. S7N and S9E,F). The mislocalization of Lamin A:RFP could be due to overexpression and/or improper processing of the protein under high levels of expression (Wang et al, 2012). In cells with intact chromatin bridges, overexpression of Lamin A:RFP did not reduce nuclear tension, as indicated by nuclear chromatin shape-deformation and relatively low frequency of loose chromatin bridges, compared with controls expressing GFP- or mCherry-only (Fig. 8B–D,F,G). However, cells overexpressing Lamin A:RFP and exhibiting intact DNA bridges failed to accumulate Nesprin-2 at the base of chromatin bridges compared with mCherry controls; instead, Nesprin-2 co-localized with Lamin A:RFP at the nuclear granules (Fig. 8B,D,H) (Libotte et al, 2005). Furthermore, cells overexpressing Lamin A:RFP exhibited impaired actin patches and increased frequency of broken chromatin bridges in cytokinesis compared with mCherry controls (Fig. 8C,E,I,J), suggesting that Nesprin-2 accumulation at the base of chromatin bridges is required for tension-induced actin patch formation. Remarkably, cells overexpressing Lamin A:RFP also exhibited actin foci inside the Nesprin-2/ Lamin A:RFP nuclear granules, in the absence or presence of chromatin bridges (Fig. 8D,E,L). Furthermore, in cells overexpressing Lamin A:RFP without DNA bridges, depletion of

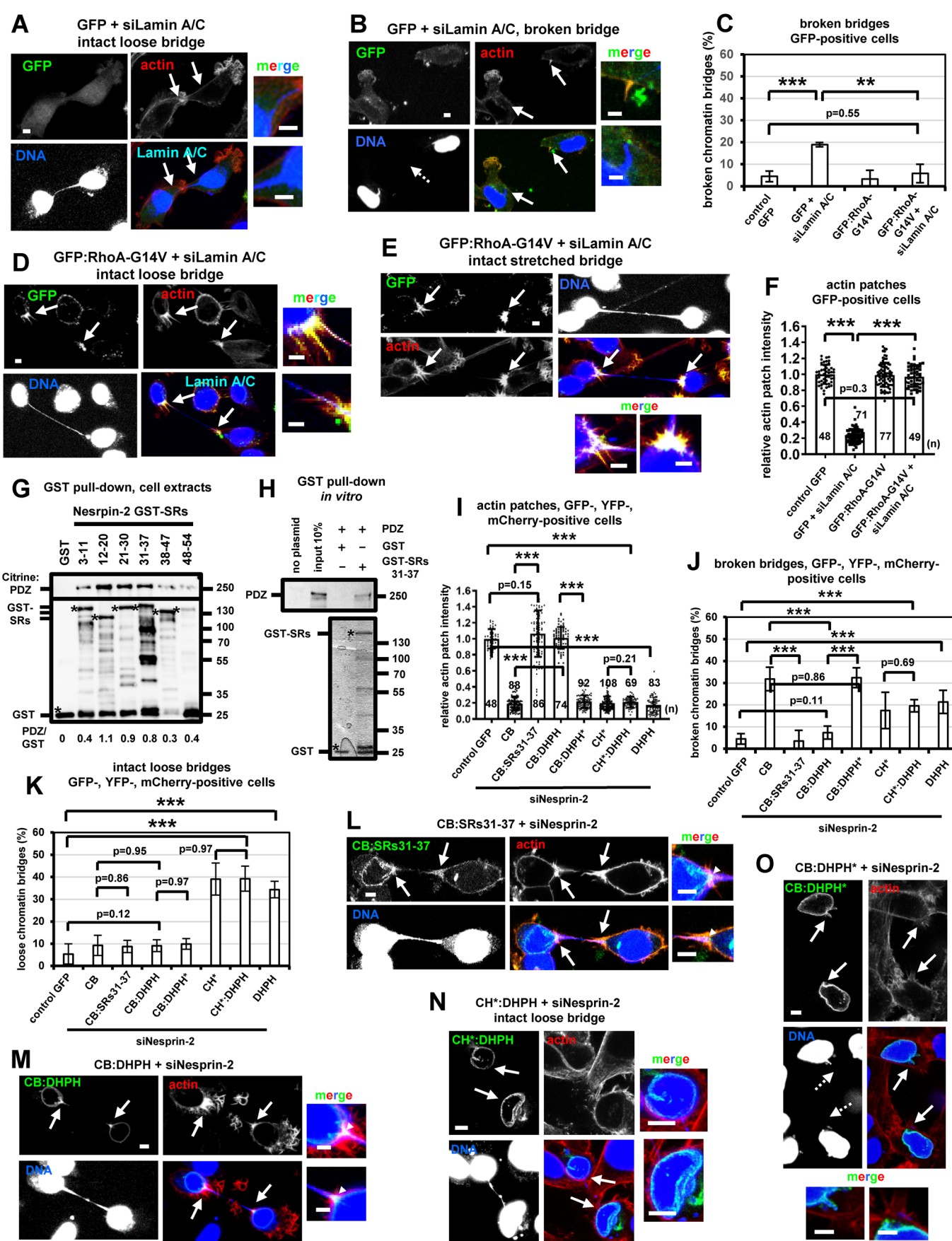

◄

**Figure 7. Expression of mini-Nesprin-2 CB protein fused to spectrin repeats (SRs) 31-37 rescues actin patches and prevents chromatin bridge breakage in cytokinesis.**

(A, B) Actin patches and chromatin bridges in BE cells transfected with Lamin A/C siRNA (siLamin A/C) and expressing GFP Intact arrows indicate actin patches and/or canal bases. Broken arrows indicate broken DNA bridges. (C) Percentage of broken DNA bridges in GFP-positive cells. Mean ± SD from three independent experiments ($n = 125, 58$, 83, 63). ***$P = 2.65E-05$; **$P = 0.0034$ by ANOVA and Student's $t$ test. (D, E) Actin patches and chromatin bridges in cells transfected with siLamin A/C and expressing GFP:RhoA-G14V. Intact arrows indicate actin patches. Scale bars, 5 μm. (F) Actin patches intensity. Mean ± SD from $n$ cells from two independent experiments. Values in control GFP were set to 1. ***$P = 7.64E-71$ (control GFP vs GFP+siLaminA/C), 5.45E-67 (GFP+siLaminA/C vs GFP:RhoA-G14V+siLaminA/C) by ANOVA and Student's $t$ test. (G) Cell lysates from BE cells expressing Citrine:PDZ were incubated with GST-SRs or GST. Associated proteins were detected by Western blotting and the ratios of PDZ/GST bands are shown. (H) Biotin labeled PDZ RhoGEF from Transcend reactions was incubated with GST-SRs 31-37 or with GST. PDZ was detected by alkaline phosphatase and colorimetric detection of the bands (top). Lower gel: Coomassie staining of the GST proteins used in the GST pull-down (bottom). Asterisks, predicted molecular weights. (I) Actin patches intensity in cells expressing GFP, mini-Nesprin-2 CB or CH*, or DHPH proteins in the absence (control) or presence of Nesprin-2 siRNA (siNesprin-2). Mean ± SD from $n$ cells from two independent experiments. Values in control GFP were set to 1. Numbers below/next to each bar indicate $n$. ***$P = 5.29E-75$ (control GFP vs CH*:DHPH +siNesprin-2), 7.80E-79 (control GFP vs DHPH+siNesprin-2), 5.59E-64 (CB+siNesprin-2 vs CB:SRs31-37+siNesprin-2), 1.72E-95 (CB+siNesprin-2 vs CB:DHPH+siNesprin-2), 4.76E-97 (CB:DHPH+siNesprin-2 vs CB:DHPH*+siNesprin-2) by ANOVA and Student's $t$ test. (J) Percentage of broken DNA bridges in GFP- or YFP-positive cells. Mean ± SD from three independent experiments ($n = 125, 84, 72, 95, 70, 107, 56, 50$). ***$P = 6.61E-05$ (control GFP vs CH*:DHPH+siNesprin-2), 0.00023 (control GFP vs DHPH+siNesprin-2), 2.40E-05 (CB+siNesprin-2 vs CB:DHPH+siNesprin-2), 1.99E-05 (CB+siNesprin-2 vs CB:DHPH+siNesprin-2), 2.00E-05 (CB:DHPH+siNesprin-2 vs CB:DHPH*+siNesprin-2) by ANOVA and Student's $t$ test. (K) Percentage of intact loose DNA bridges. Mean ± SD from three independent experiments ($n = 143, 84, 67, 95, 70$, 107, 56, 50). ***$P = 8.72E-07$ (control GFP vs CH*:DHPH+siNesprin-2), 1.70E-06 (control GFP vs DHPH+siNesprin-2) by ANOVA and Student's $t$ test. (L–O) Actin patches and chromatin bridges in cells treated as in (I). Intact arrows indicate actin patches and/or canal bases. Broken arrows indicate broken DNA bridges. Insets show high magnifications of the canal bases. Scale bars, 5 μm. Source data are available online for this figure.

Nesprin-2 or RhoA impaired generation of actin foci inside the Lamin A:RFP nuclear granules compared with controls (Fig. 8K–N), showing that ectopic Nesprin-2 accumulation can promote local actin polymerization through RhoA. Second, cells were transfected with progerin fused to GFP (progerin:GFP), which increases nuclear stiffness (Booth et al, 2015; Verstraeten et al, 2008). In cells with intact chromatin bridges, expression of progerin:GFP correlated with reduced nuclear chromatin shape-deformation, impaired Nesprin-2 accumulation at the base of chromatin bridges and impaired actin patches compared with mCherry- or GFP-only controls (Appendix Fig. S9G–K). Collectively, these results suggest that Nesprin-2 accumulation at the base of chromatin bridges promotes actin patch formation.

## Discussion

On the basis of the above findings we propose the following model (Fig. 8O): In cytokinesis with intact chromatin bridges, the nuclear envelope Sun1/2-Nesprin-2 LINC complex transmits mechanical tension on daughter nuclei by interacting with the actin cytoskeleton and the nuclear lamina. This nuclear tension leads to accumulation of the Sun1/2-Nesprin-2 LINC complex at the front of the nucleus, near the base of chromatin bridges, and to local enrichment of the RhoA activator PDZ RhoGEF at the cytoplasm, through PDZ-binding to Nesprin-2 spectrin repeats. In turn, PDZ RhoGEF activates the RhoA-ROCK-LIMK-Cofilin and RhoA-mDia1 signaling pathways to promote actin patch formation and prevent chromatin bridge breakage in cytokinesis.

These findings are the first to describe a mechanism for detecting chromatin bridges to generate actin patches in cytokinesis, by converting mechanical stress exerted on the nucleus by the chromatin bridge-pulling forces to biochemical signaling in the cytoplasm through the LINC complex. This mechanism directly couples chromatin bridge forces with actin patch formation and is consistent with previous findings that actin patches remain stable during interphase but disappear after chromatin bridge resolution or cleavage furrow regression, which would likely diminish nuclear tension

(Steigemann et al, 2009). In contrast to other known mechanotrans-duction pathways that transmit mechanical forces from plasma membrane receptors to the nucleus to induce chromatin remodeling and/or gene expression (Bouzid et al, 2019; DuFort et al, 2011), this mechanism operates "inside out" by reacting to intracellular stress to modulate the actin cytoskeleton, raising the possibility that other mechanical cues may use LINC complexes to trigger similar responses. Our data is also consistent with previous findings that LINC complexes can promote RhoA activation and the assembly of an actomyosin network around the nucleus through transcription-independent mechanisms under specific conditions (Booth et al, 2019; Thakar et al, 2017). Furthermore, whereas other mechanosensing pathways promote tumor formation or progression by activating biochemical pathways that enhance cell proliferation and motility or DNA damage (Broders-Bondon et al, 2018), the above mechanism preserves genome stability by preventing chromatin bridges from breaking in cytokinesis and could protect against cancer (Mazzagatti et al, 2024; Umbreit et al, 2020).

It was previously reported that inhibition of myosin light chain kinase, which prevents myosin II activation but did not apparently block chromatin bridge extension, delayed chromatin bridge breakage in cytokinesis (Umbreit et al, 2020). Because RhoA inhibition can prevent myosin II activation, other members of the Rho protein family could activate myosin II in RhoA-deficient cells with chromatin bridges (Hirano and Hirano, 2022). Alternatively, longitudinal tension rather than contractile actomyosin forces may constitute the main cause of chromatin bridge breakage in RhoA-deficient cells in our experimental system.

Why impaired actin polymerization/stabilization leads to chromatin bridge breakage in some cases (Dandoulaki et al, 2018) but furrow regression in others (Bai et al, 2020) remains unclear and could depend on the experimental conditions (Petsalaki and Zachos, 2021a). Also, whether actin patch formation is somehow linked to the activation of the abscission checkpoint is a matter of active investigation. For example, it will be interesting to examine whether cells with dicentric bridges that do not activate the abscission checkpoint can generate actin patches through the present mechanism (Petsalaki et al, 2023).

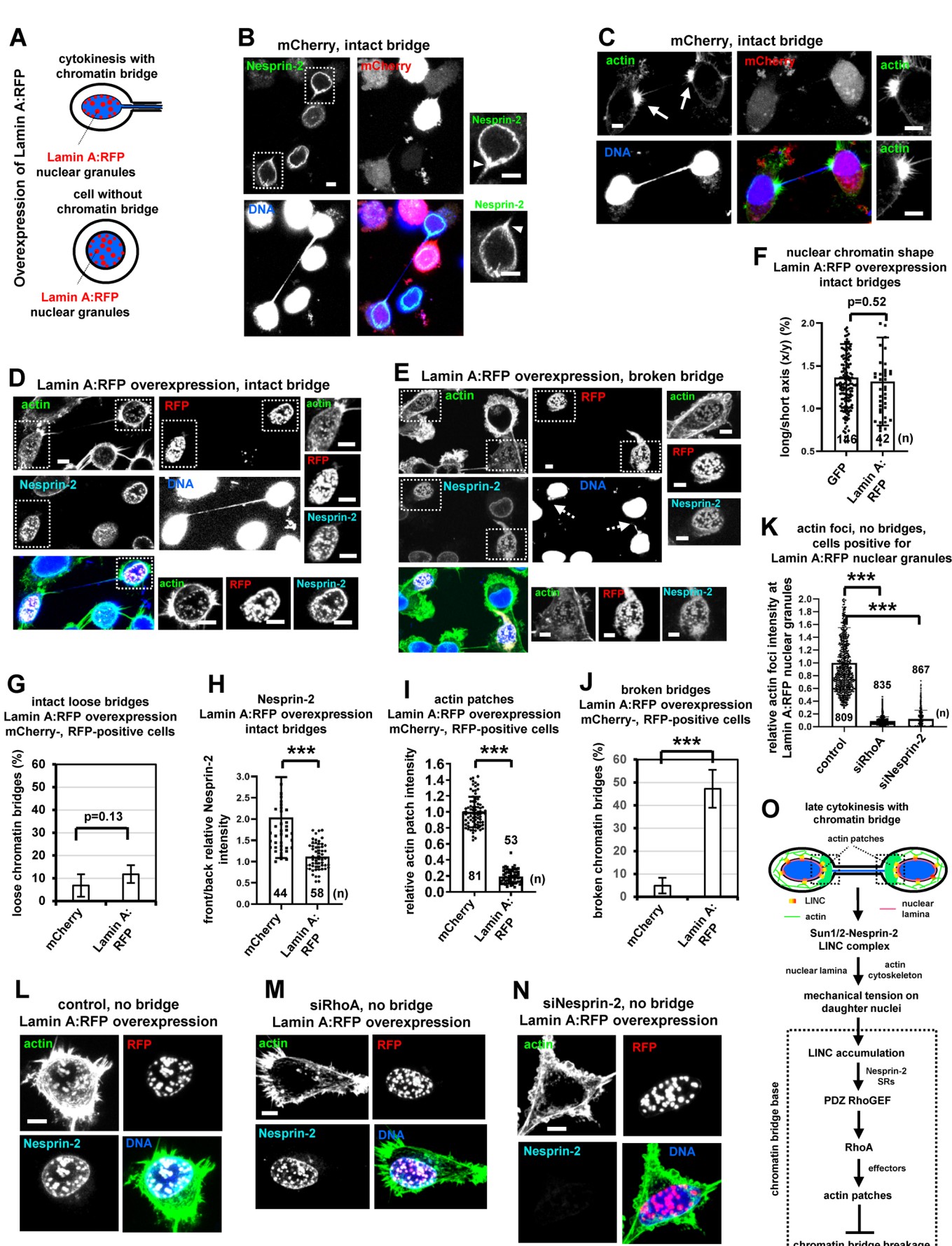

**Figure 8.    Ectopic accumulation of Nesprin-2 in nuclear granules promotes local actin polymerization.**

(A) Cartoon of Lamin:A RFP nuclear granules in BE cells overexpressing Lamin A:RFP. (B–E) Actin patches, chromatin bridges and Nesprin-2 localization in mCherry-positive cells or cells overexpressing Lamin A:RFP. Intact arrows indicate actin patches. Broken arrows indicate broken DNA bridges. Arrowheads show Nesprin-2 accumulation at the base of the DNA bridge. Insets show high magnifications of actin patches or boxed nuclei. Scale bars, 5 μm. (F) Nuclear chromatin shape in GFP-positive cells or cells overexpressing Lamin A:RFP. Mean ± SD from *n* cells from two independent experiments. $P = 0.52$ by Student's *t* test. (G) Percentage of intact loose DNA bridges in mCherry- or RFP-positive cells. Mean ± SD from three independent experiments ($n = 107, 60$). $P = 0.13$ by Student's *t* test. (H) Front/back Nesprin-2 intensity. Mean ± SD from *n* cells from two independent experiments. ***$P = 4.41E-10$ by Student's *t* test. (I) Actin patches intensity in mCherry- or RFP-positive cells. Values in mCherry were set to 1. Mean ± SD from n cells from two independent experiments. ***$P = 4.10E-59$ by Student's *t* test. (J) Percentage of broken DNA bridges. Mean ± SD from four independent experiments ($n = 57, 60$). ***$P = 8.17E-05$ by Student's *t* test. (K) Intensity of actin foci at Lamin A:RFP nuclear granules in the absence (control) or presence of RhoA siRNA (siRhoA) or Nesprin-2 siRNA (siNesprin-2). Values in control were set to 1. Mean ± SD from *n* cells from two independent experiments. Numbers below/next to each bar indicate n. ***$P = 0$ (control vs siRhoA), $1.09E-297$ (control vs siNesprin-2) by ANOVA and Student's *t* test. (L–N) Nesprin-2 and actin localization in cells exhibiting Lamin A:RFP nuclear granules. Scale bars, 5 μm. (O) Proposed mechanism by which human cells sense nuclear tension to generate actin patches in cytokinesis with chromatin bridges. SRs spectrin repeats. Source data are available online for this figure.

Our model proposes that nuclear tension promotes binding of Nesprin-2 SRs to PDZ RhoGEF, to activate RhoA signaling. Although a role for mechanical forces inducing a change in conformation of the Nesprin-2 cytoplasmic domain that makes SRs more accessible for binding to PDZ cannot be formally excluded (Grum et al, 1999), our results are more consistent with nuclear tension activating RhoA by promoting accumulation of Nesprin-2 and associated PDZ RhoGEF at the base of chromatin bridges for two main reasons: First, targeting RhoA to the cytoplasmic region of Nesprin-2 by inserting the DH/PH domain of PDZ in mini-Nesprin-2 protein CH* did not promote actin patch formation in the absence of Nesprin-2 accumulation. Secondly, ectopic accumulation of Nesprin-2 in nuclear granules promoted local actin polymerization by RhoA in cells without chromatin bridges, therefore in the absence of nuclear tension. Because depletion of RhoA does not impair nuclear tension in cells with intact chromatin bridges, one possibility is that, in RhoA-depleted cells, the endogenous RhoA that remains, although it is not sufficient to support actin patch formation, can still target the constitutively active forms of ROCK and LIMK to the base of the intercellular canal to promote actin patches. Also, because depletion of Sun1/2 or Lamin A/C reduces, but does not completely abolish, nuclear tension, the transfected constitutively active RhoA could associate, through PDZ RhoGEF, with the remaining Nesprin-2 at the base of the intercellular canal in Nesprin-2-depleted cells to promote actin patches, whereas the endogenous RhoA may not be sufficient to support actin patch formation in the absence of a relatively pronounced Nesprin-2 accumulation. Our model could explain why/how actin patches are spatially confined to the bases of the DNA bridges and has analogies with protein kinase clustering at kinetochores or intercellular compartments stimulating their catalytic activity and generating spatial restrictions to their substrates (Gormal et al, 2024; Kelly et al, 2007).

Although the mechanism by which nuclear tension promotes Nesprin-2 accumulation at the base of chromatin bridges is incompletely understood, it is reminiscent of Nesprin accumulation toward the front of the nucleus during migration of cells through confined spaces, which requires Nesprin-2 binding to the actin cytoskeleton (Davidson et al, 2020). One possibility is that perturbations in nuclear membrane thickness near the base of the chromatin bridge, where membrane deformation by bridge forces is presumably stronger, can promote assembly of Nesprin-2 aggregates through local protein interactions to generate actin filaments (Goulian, 1996). In turn, these actin filaments could facilitate

transport or retention of Nesprin-2 to the "front" of the nucleus to catalyze actin patch formation (Davidson et al, 2020).

The mechanism by which actin patches protect against chromosome breakage is unclear. One possibility is that actin patches buffer the forces exerted on the nuclei by chromatin bridges thus preventing breakage. However, RhoA-deficient cells with intact chromatin bridges (but without actin patches) exhibit nuclear shape deformations and Sun2 nuclear lines similar with controls (Fig. 5N,O), suggesting that nuclei with chromatin bridges do not exhibit additional tension in the absence of actin patches. Alternatively, the dense structure of actin patches could increase cell stiffness and elasticity near the base of chromatin bridges, providing local mechanical support and preventing bridges from breaking (Fletcher and Mullins, 2010).

How does the LINC complex promote nuclear tension in cytokinesis with chromatin bridges? One possibility is that LINC complexes distal to chromatin bridges couple the nuclei to the actomyosin network and the cell cortex through their actin binding domains and that, as the dividing cell migrates or spreads, cortical forces are transmitted to the nuclei while chromatin bridges are pulling the daughter nuclei towards the opposite direction thus generating nuclear tension.

How do Nesprin-2 SRs activate PDZ RhoGEF? Spectrin repeats consist of three helices, of which A and C are parallel and B is antiparallel (Grum et al, 1999; Pascual et al, 1997). Because all Nesprin-2 spectrin repeat-fragments that we tested associated with PDZ RhoGEF in cell extracts, PDZ could recognize the structural fold of SRs rather than a specific amino acid sequence. Furthermore, Nesprin-2 SRs bind to the RH and the catalytic DH/PH domains, but not to the PDZ domain or to the C-terminal oligomerization region (Aittaleb et al, 2010; Chikumi et al, 2004; Longenecker et al, 2001). Interaction of proteins, such as activated G proteins, with the RH or DH/PH subunits help target RhoGEFs to specific locations near the plasma membrane to activate local Rho GTPase signaling (Aittaleb et al, 2010; Longenecker et al, 2001). Because spectrin repeats are found in several cytoskeletal proteins where they act as platforms for the assembly of protein complexes (Djinovic-Carugo et al, 2002; Jayo et al, 2016), interactions between spectrin repeats and RhoGEFs could provide an alternative mechanism to promote actin polymerization in response to specific signals, independently of integrin-based extracellular signaling.

Our study identifies RhoA signaling pathways that generate actin patches in cytokinesis with chromatin bridges. We have

previously reported that the actin modulators Src and FAK are required for actin patch formation (Dandoulaki et al, 2018). Here, we show that RhoA functions upstream of Src and FAK in actin patch formation, perhaps serving to coordinate actin polymerization with reorganization of the actin cytoskeleton by Src/FAK and/or to enhance actin nucleation through Src-binding to formins (Mitra and Schlaepfer, 2006; Tominaga et al, 2000). Furthermore, these data are consistent with previous reports that RhoA/ROCK can activate Src/FAK in response to mechanical stress or to specific levels of RhoA activity (Ju et al, 2022; Torsoni et al, 2005).

Our findings also show novel functions for PDZ RhoGEF and RhoA that can protect against tumorigenesis, by preventing chromatin bridge breakage in cytokinesis. This is perhaps unexpected because Rho GEFs and RhoA are commonly believed to be pro-proliferative and pro-metastatic, and are therefore considered to act as cancer oncogenes (Vigil et al, 2010). Because RhoA is a therapeutic target for tumor progression, understanding the full range of RhoA functions in cancer cells could be important for designing optimal treatment protocols (Vigil et al, 2010). In conclusion, these findings describe a novel mechanosensing mechanism by which human cells with chromatin bridges generate actin patches to maintain genome integrity.

# Methods

### Reagents and tools table

| Reagent/resource | Reference or source | Identifier or catalog number |
|---|---|---|
| **Experimental models** | | |
| Human colon carcinoma BE cells | Chris Marshall's lab (Petsalaki et al, 2021b) | |
| Human cervical carcinoma HeLa Lap2b:RFP cells | Daniel Gerlich's lab (Steigemann et al, 2009) | |
| Human cervical carcinoma HeLa Lap2b:RFP/Lifeact:GFP cells | This study | Section: "Generation of cell lines" |
| **Recombinant DNA** | | |
| pEGFP-N1 | Takara Bio | |
| pEGFP-RhoA | Addgene | 23224 |
| mCherry2-N1 | Addgene | 54517 |
| dTomato-2xrGBD | Addgene | 129625 |
| pBABE-puro-GFP-progerin | Addgene | 17663 |
| mCherry-DN KASH | Addgene | 125553 |
| RHGEF11(DHPH)-CRY2-mCherry | Addgene | 89481 |
| mRFP-Lamin A | Addgene | 124268 |
| GFP-LIMK(WT) | Gareth Thomas' lab (George et al, 2015) | |
| Myc-ROCK(WT) | Mike Olson's lab (Priya et al, 2017) | |
| Myc-ROCK Δ2 | Mike Olson's lab (Priya et al, 2017) | |
| YFP-CB | Nicolas Borghi's lab (Déjardin et al, 2020) | |

| Reagent/resource | Reference or source | Identifier or catalog number |
|---|---|---|
| YFP-CH* | Nicolas Borghi's lab (Déjardin et al, 2020) | |
| mCitrine-YFP-ARHGEF11 | Oliver Rocks' lab (Muller et al, 2020) | |
| 3FLAG-ARHGEF12 | Oliver Rocks' lab (Muller et al, 2020) | |
| pEGFP/FAK | Paris Skouridis' lab | |
| pcDNA3.1/ARCHGEF11 | This study | Section: Plasmid construction |
| YFP-CB/DHPH | This study | Section: Plasmid construction |
| YFP-CH*/DHPH | This study | Section: Plasmid construction |
| YFP-CB/SRs31-37 | This study | Section: Plasmid construction |
| pGEX4T1/SRs3-11 | This study | Section: Plasmid construction |
| pGEX4T1/SRs21-30 | This study | Section: Plasmid construction |
| pGEX4T1/SRs31-37 | This study | Section: Plasmid construction |
| pGEX4T1/SRs38-47 | This study | Section: Plasmid construction |
| pGEX4T1/SRs48-54 | This study | Section: Plasmid construction |
| GFP:PDZ$^{dom}$ | This study | Section: Plasmid construction |
| GFP:PDZ(RH) | This study | Section: Plasmid construction |
| GFP:PDZ(DH/PH) | This study | Section: Plasmid construction |
| GFP:PDZ(C-term) | This study | Section: Plasmid construction |
| pGEX4T1/SRs12-20 | This study | Section: Plasmid construction |
| EGFP/Src-S51D | Zachos' lab (Dandoulaki et al, 2018) | |
| EGFP/RhoA$^{Res}$ | This study | Section: Mutagenesis |
| GFP/LIMK(T408EE) | This study | Section: Mutagenesis |
| YFP-CB/DHPH(E15A) | This study | Section: Mutagenesis |
| Citrine-PDZ$^{Res}$ | This study | Section: Mutagenesis |
| mCherry-KASH-ΔL | This study | Section: Mutagenesis |
| EGFP/FAK(Y397E) | This study | Section: Mutagenesis |
| **Antibodies** | | |
| Mouse monoclonal anti-RhoA (26C4) | Santa Cruz Biotechnology | sc-418 |
| Mouse monoclonal anti-mDia1 (E4) | Santa Cruz Biotechnology | sc-373807 |

| Reagent/resource | Reference or source | Identifier or catalog number |
|---|---|---|
| Mouse monoclonal anti-ROCK1 (G6) | Santa Cruz Biotechnology | sc-17794 |
| Mouse monoclonal anti-paxillin (B-2) | Santa Cruz Biotechnology | sc-365379 |
| Mouse monoclonal anti-PDZ RhoGEF (C-9) | Santa Cruz Biotechnology | sc-74565 |
| Mouse monoclonal anti-GST (B-14) | Santa Cruz Biotechnology | sc-138 |
| Mouse monoclonal anti-c-Myc (N-19) | Santa Cruz Biotechnology | sc-867 |
| Mouse monoclonal anti-LIMK1 (H-12) | Santa Cruz Biotechnology | sc-515585 |
| Mouse monoclonal anti-FAK (D-1) | Santa Cruz Biotechnology | sc-271123 |
| Rabbit polyclonal anti-GFP (FL) | Santa Cruz Biotechnology | sc-8334 |
| Rabbit polyclonal anti-OctA (FLAG, D-8) | Santa Cruz Biotechnology | sc-807 |
| Mouse monoclonal anti-integrin b1 | Abcam | ab24693 |
| Rabbit polyclonal anti-phospho Cofilin-serine 3 | Abcam | ab100836 |
| Rabbit polyclonal anti-Nesprin-2 | Abcam | ab204308 |
| Rabbit polyclonal anti-Nesprin-2 (Ab2) | Abcam | ab233034 |
| Rabbit polyclonal anti-phospho LIMK1-threonine 508 | Abcam | ab194798 |
| Rabbit polyclonal anti-Lamin A | Abcam | ab26300 |
| Rabbit polyclonal anti-mCherry | Abcam | ab167453 |
| Rabbit polyclonal anti-FHOD1 | Abcam | ab224461 |
| Rabbit polyclonal anti-Src | Abcam | ab109381 |
| Mouse monoclonal anti-a-tubulin | Sigma | DM1A |
| Mouse monoclonal anti-actin | Sigma | AC-40 |
| Mouse monoclonal anti-Sun2 | Millipore | clone 3.1E |
| Rabbit polyclonal anti-Sun1 | Novus | NBP1-87395 |
| goat anti-rabbit IgG Alexa Fluor633-conjugated (for IF) | Invitrogen, Thermo Fisher Scientific | A21071 |
| goat anti-mouse IgG FITC-conjugated (for IF) | Jackson ImmunoResearch | 115-096-072 |
| goat anti-rabbit IgG TRITC-conjugated (for IF) | Jackson ImmunoResearch | 111-025-046 |
| sheep anti-mouse IgG-TRITC conjugated (for IF) | Jackson ImmunoResearch | 515-025-072 |
| goat anti-rabbit IgG FITC-conjugated (for IF) | Jackson ImmunoResearch | 111-096-047 |
| horse anti-mouse IgG HRP-linked | Cell Signaling | 7076 |
| goat anti-rabbit IgG HRP-linked | Cell Signaling | 7074 |
| **Oligonucleotides and other sequence-based reagents** | | |
| siRNAs | This study | Section: siRNA sequences |

| Reagent/resource | Reference or source | Identifier or catalog number |
|---|---|---|
| **Chemicals, enzymes and other reagents** | | |
| Y16 (RhoAi)) | Sigma-Aldrich | 504043 |
| Y27632 (ROCKi), | EMD Millipore | 688002 |
| SMIFH2 (mDia1i), | Merck Millipore | 344092 |
| TH-257 (LIMKi), | Sigma-Aldrich | SML2275 |
| SB273005 (integrin i) | Selleckchem | S7540, |
| TG003 (Clki) | Sigma-Aldrich | T5575 |
| blebbistatin | Sigma-Aldrich | 203391 |
| Biotracker 488 Green Nuclear Dye | Sigma | SCT-120 |
| Fluorescein Phalloidin | Invitrogen | F432; |
| Phalloidin Rhodamine | Abcam | ab235138 |
| FM 1-43FX | Invitrogen | F35355 |
| TnT Quick Coupled Transcription/Translation System | Promega | L1170 |
| Vectashield medium | Vector Laboratories | H-1000 |
| low-fluorescence immersion oil | Leica Microsystems | 11513859 |
| fibronectin | EMD Millipore | FC010 |
| Q5 site-directed mutagenesis kit | New England Biolabs | E0554S |
| **Software** | | |
| LASX | Leica Microsystems | |
| ImageJ | Open source | |
| **Other** | | |
| Sequence of human Nesprin-2 Spectrin Repeats 3-54 | This study | GenBank PQ453063 |

## Plasmid construction

Plasmid pEGFP-N1 coding for GFP under cytomegalovirus promoter was obtained from Takara Bio. Plasmids pEGFP-RhoA (#23224) coding for RhoA protein fused to EGFP inside the pEGFP-C3 vector, mCherry2-N1 (#54517) coding for mCherry under cytomegalovirus promoter, dTomato-2xrGBD (#129625) coding for dimericTomato fused to a double rhotekin G protein-binding domain inside the pEGFP-C1 backbone, and pBABE-puro-GFP-progerin (#17663) coding for D50 lamin A fused to GFP inside the pBABE-puro vector were from Addgene. Plasmids mCherry-DN KASH (#125553) coding for the KASH domain of human Nesprin-1 fused to mCherry inside the pmcherry N1 vector, ARHGEF11(DHPH)-CRY2-mCherry (#89481) coding for the DH/PH domain of human PDZ RhoGEF fused to mCherry inside the pmcherry N1 vector, and mRFP-Lamin A (#124268) coding for human Lamin A/C protein fused to RFP inside the mRFP-N2 vector were also from Addgene. Plasmid GFP-LIMK(WT) coding for wild-type human LIMK1 fused to GFP inside the eGFP-N2 vector (Clontech) was a gift from Gareth Thomas (University of Southampton, UK) (George et al, 2015). Plasmids Myc-ROCK(WT) coding for myc-tagged full-length human ROCK1, and Myc-ROCK Δ2 coding for myc-tagged human ROCK1 amino acids 1–727 inside the pCAG vector were a gift from Mike Olson (Ryerson

University, Toronto, Canada) (Priya et al, 2017). Plasmids YFP-CB and YFP-CH* coding for the CH, KASH domains and spectrins 1, 2, 55, 56 of mouse Nesprin-2G fused to YFP inside the pcDNA3.1 Hygro (−) vector (Thermo Fisher Scientific) were a gift from Nicolas Borghi (Institut Jacques Monod, Paris, France) (Déjardin et al, 2020). The CH* mutant exhibits the Isoleucines 128, 131 to alanine point mutations that impair Nesprin-2 binding to F-actin (Luxton et al, 2010). Plasmids mCitrine-YFP-ARHGEF11 coding for human PDZ-RhoGEF fused to Citrine YFP and 3FLAG-ARHGEF12 coding for human LARG fused to 3xFLAG peptides were from Oliver Rocks (Max-Delbrück-Center for Molecular Medicine, Berlin, Germany) (Muller et al, 2020). Plasmid pEGFP/FAK coding for human FAK fused to EGFP inside the pCS105 vector (Addgene) was a gift from Paris Skouridis (University of Cyprus, Nicosia, Cyprus) (Dandoulaki et al, 2018). To generate plasmid pcDNA3.1/ ARCHGEF11 coding for human PDZ RhoGEF for in vitro transcription-translation, PDZ RhoGEF was amplified by PCR by using plasmid mCitrine-YFP-ARHGEF11 as a template and inserted into the pcDNA3.1 zeo (+) vector (Invitrogen) in two steps: first, the nucleotides 1–1943 were inserted as a KpnI-EcoRI fragment and then the nucleotides 1944–4566 as an EcoRI-XhoI fragment. To generate YFP-CB/DHPH and YFP-CH*/ DHPH plasmids coding for mini-Nesprin-2 CB or CH* proteins fused to DHPH, the DHPH domain of human PDZ RhoGEF was amplified by PCR by using ARHGEF11(DHPH)-CRY2-mCherry as substrate and cloned into the YFP-CB or YFP-CH* vectors, respectively, between the YFP and spectrin repeat 56 as a PacI-NarI fragment. To generate YFP-CB/SRs31-37 plasmid coding for CB protein fused to spectrin repeats 31–37 of Nesprin-2, human SRs 31–37 were amplified by PCR by using pGEX4T1/SRs31–37 plasmid as substrate and cloned into the YFP-CB vector between the YFP and SR56 as a PacI-NarI fragment. Plasmids pGEX4T1/SRs3-11, pGEX4T1/SRs21-30, pGEX4T1/SRs31-37, pGEX4T1/SRs38-47 and pGEX4T1/SRs48-54 coding, respectively, for spectrin repeats 3–11, 21–30, 31–37, 38–47 and 48–54 of human Nesprin-2 fused to GST were generated by PCR-amplifying the respective sequences from cDNA derived from human BE cells and inserting them into the pGEX4T1 vector (GE Healthcare) as XhoI-NotI fragments. Plasmids GFP:PDZ^dom, GFP:PDZ(RH), GFP:PDZ(DH/PH) and GFP:PDZ(C-term) coding, respectively, for amino acid sequences 1–219, 220–701, 702–1152 or 1168–1522 of human PDZ RhoGEF fused to GFP were generated by PCR-amplifying the respective sequences from mCitrine-YFP-ARHGEF11 plasmid and inserting them into the pEGFP-C1 vector (Clontech) as EcoRI-XhoI fragments. Plasmid pGEX4T1/SRs12-20 coding for spectrin repeats 12–20 of human Nesprin-2 fused to GST was generated by amplifying the respective sequence by PCR from cDNA derived from human BE cells and inserting it into the pGEX4T1 vector (GE Healthcare) as a SalI-NotI fragment. Plasmid EGFP/Src-S51D coding for human c-Src in which serine 51 is changed to aspartic acid fused to EGFP inside the pcDNA4/TO vector (Invitrogen) was previously described (Dandoulaki et al, 2018).

## Generation of cell lines

To generate the HeLa Lap2b:RFP/Lifeact:GFP cell line, HeLa cells stably expressing Lap2b fused to RFP (a gift from Daniel Gerlich, Institute of Molecular Biotechnology, Vienna, Austria) (Steigemann et al, 2009) were transfected with the pEGFP-C1/Lifeact-EGFP plasmid (#58470, Addgene) coding for Lifeact peptide fused to EGFP inside the pEGFP-C1 vector, conferring resistance to neomycin (Invitrogen). Neomycin-resistant clones that were

positive for Lap2b:RFP and Lifeact:EGFP were identified by fluorescence microscopy, isolated and expanded.

## Isolation of total RNA and first strand cDNA synthesis

Human colon carcinoma BE cells were lysed in Trizol Reagent (ThermoFischer Scientific) and centrifuged at $12,000 \times g$ for 5 min at 4 °C. Samples were then extracted with chloroform, centrifuged at $12,000 \times g$ for 15 min at 4 °C and total RNA was precipitated from the aqueus phase with isopropanol followed by centrifugation at $12,000 \times g$ for 10 min at 4 °C, and resuspended in RNAse free water (Merck).

For first strand cDNA synthesis, 1 µg total RNA was incubated with SuperScript II RT reverse transcriptase (Invitrogen) using gene-specific primers at 42 °C for 50 min.

## Mutagenesis

Point mutations were generated by using the Q5 site-directed mutagenesis kit (E0554S, New England Biolabs). To generate the EGFP/RhoA^Res plasmid encoding an siRNA-resistant form of RhoA fused to GFP, the pEGFP-RhoA plasmid was used to introduce C432G, A435G and T438C point mutations giving resistance to the RhoA siRNA. To generate the GFP/RhoA(G14V) plasmid, the pEGFP/RhoA plasmid was used to introduce a G41T point mutation. To generate the GFP/LIMK(T408EE) plasmid, the GFP-LIMK(WT) plasmid was used to introduce A1522GA, C1523AG and C1524AA point mutations. To generate YFP-CB/ DHPH(E15A) plasmid (DHPH*) (Lenoir et al, 2014), the YFP-CB/ DHPH plasmid was used to introduce a A44C point mutation inside the DH/PH domain. To generate siRNA resistant Citrine-PDZ^Res, the G2346A, T2349C and A2352G point mutations conferring resistance to siPDZ-2 were introduced into the mCitrine-YFP-ARHGEF11 plasmid. To generate mCherry-KASH-ΔL lacking the luminal SUN protein-binding KASH domain (Luxton et al, 2010), the mCherry-DN KASH plasmid was used as template to introduce the G844T, C845G and C846A point mutations creating a stop codon. To generate EGFP/FAK(Y397E) plasmid, the EGFP/FAK plasmid was used to introduce T1189G and T1191A point mutations.

## siRNA sequences

Human RhoA (sc-29471; 5'-GGCAGAGAUAUGGCAAACA-3'), p115 RhoGEF (sc-41734; a pool of three individual siRNAs: 5'-CUGGAG GAGAUGCAACAUA-3', 5'-CCAAGAGUGGAGACAAGAA-3', 5'-CC GAUCACAAAGCCUUCUA-3'), PDZ RhoGEF (sc-45823; a pool of three individual siRNAs: 5'-GAACCUGCCUGAACUCAUA-3', 5'-CA AGAGCCUGGAUCUUACA-3', 5'-CCUCAGACAUGCAAGUGAA-3'), LARG RhoGEF (sc-41800; a pool of three individual siRNAs: 5'-GGAUGGAGCUGUAGUUACA-3', 5'-CCAGAGCAUUGAAUUAC UA-3', 5'-CGAAGGAGAUAAUGAUGAA-3'), Sun1 (sc-106672; a pool of three individual siRNAs: 5'-GGAUGCCGUACAAGAAAGA-3', 5'-GUAACUGCUGGGCAUUUAA-3', 5'-CAAGGCACUUAAAGUGU UA-3'), Sun2 (sc-76612; a pool of three individual siRNAs: 5'-GG AAAUCCAGCAACAUGAA-3', 5'-GACGUAUGGUGCUUGGUAU-3', 5'-GCAUCAGCAAGACUCAGAA-3'), Nesprin-2 (Syne-2; sc-61630; a pool of three individual siRNAs: 5'-GGUAGAACGUCAACCUCAA-3', 5'-CAAACAGCCUUCUCAUUAA-3', 5'-GACUUCUGUUGUACUGA

AA-3'), Lamin A/C (sc-35776, 5'-CUGGACUUCCAGAAGAACA-3'), PDZ-2 (5'-GAACCUGCCUGAACUCAUA-3') and c-Src (sc-29228, 5'-GCAGUUGUAUGCUGUGGUU-3') siRNAs were from Santa Cruz Biotechnology. Only the sense sequences of the siRNA duplexes are shown.

## Cell culture and treatments

Human colon carcinoma BE cells (diploid cells that contain an oncogenic Kras-G13D mutation as well as the BRAF-G463V oncogenic mutation; a gift from Simon Wilkinson and Christopher Marshall, Institute of Cancer Research, London, UK) (Petsalaki and Zachos, 2021b), cervical carcinoma HeLa Lap2b:RFP (Steigemann et al, 2009) and HeLa Lap2b:RFP/Lifeact:GFP cells were grown in DMEM (Biosera) containing 10% FBS (Biosera), 100 U/ml penicillin, and 100 µg/ml streptomycin at 37 °C in 5% $CO_2$. Cells were treated with 50 µM Y16 (RhoAi; 504043, Sigma-Aldrich), 10 µM Y27632 (ROCKi; 688002, EMD Millipore), 25 µM SMIFH2 (mDia1i; 344092, Merck Millipore), 10 µM TH-257 (LIMKi; SML2275, Sigma-Aldrich), 1 µM SB273005 (integrin i; S7540, Selleckchem), 1 µM TG003 (Clki; T5575, Sigma-Aldrich), 50 µM or 10 µM DMSO (vehicle controls), 50 µM blebbistatin (203391, Sigma-Aldrich), or 15 µg/ml nocodazole (Sigma-Aldrich) as appropriate. Drugs or DMSO were added to the medium 15 min–5 h before fixation for analysis by indirect immuno-fluorescence microscopy, or immediately before filming for time-lapse imaging. siRNA duplexes were transfected into BE cells 48 h before analysis using Lipofectamine 2000 (Invitrogen), unless otherwise stated. For expression of exogenous proteins, plasmids were transfected into cells in the absence or presence of appropriate siRNA duplexes 48 h before analysis or further drug treatment using Lipofectamine 2000 (Invitrogen). To visualize integrin or paxillin by fluorescence microscopy, cells were grown on fibronectin-coated slides. All cell lines used exhibited consistent morphology and growth properties and were negative for mycoplasma contamination.

## Coating with fibronectin

Slides were incubated with 1 µg/ml fibronectin (FC010, EMD Millipore) for 1 h at 37 °C before use.

## Time-lapse imaging and Biotracker staining

HeLa Lap2b:RFP, HeLa Lap2b:RFP/Lifeact:GFP or BE cells were seeded onto Petri dishes with a 30-mm glass base (Greiner) and an inverted fluorescence microscope (Observer D1; Zeiss) was used. Fluorescence or phase-contrast images were taken by using a ×20 Plan Neofluor 0.4 NA Ph2 dry objective (Zeiss). Imaging was performed at 37 °C in 5% $CO_2$ by using a Zeiss AxioCam MRm camera and Zeiss ZEN 2 acquisition software. Drugs or DMSO were added to the medium immediately before filming, as appropriate. For DNA staining in BE cells using the Biotracker 488 Green Nuclear Dye (Sigma, Cat# SCT-120), 0.5 µl/ml Biotracker and 100 µM of the efflux pump inhibitor Verapamil were added to the medium 1 h before filming.

## Indirect immunofluorescence microscopy and microscope image acquisition

Cells were fixed in 4% paraformaldehyde in cytoskeleton buffer (1.1 M $Na_2HPO_4$, 0.4 M $KH_2PO_4$, 137 mM NaCl, 5 mM KCl, 2 mM

$MgCl_2$, 2 mM EGTA, 5 mM Pipes, and 5 mM glucose, pH 6.1) for 5 min at 37 °C, permeabilized in 0.5% Triton X-100 in cytoskeleton buffer, washed twice with PBS at room temperature, and immunostained. FITC- or rhodamine-TRITC-conjugated (Jackson ImmunoResearch) or Alexa Fluor 633-conjugated (Thermo Fisher Scientific) secondary antibodies were used as appropriate. DNA was stained with 10 µg/ml DAPI (Biotium). Actin patches were stained with Fluorescein Phalloidin (F432; Invitrogen, Thermo Fisher Scientific) or with Phalloidin Rhodamine (ab235138, Abcam) and cells were mounted in Vectashield medium (H-1000, Vector Laboratories).

For staining of intercellular canals, cells in culture dishes were incubated with the fluorescence lipophilic dye FM 1-43FX (F35355; Invitrogen, Thermo Fisher Scientific) in cold (4 °C) OptiMEM medium (Gibco) for 5 min at 4 °C. Cells were then fixed with 4% cold (4 °C) paraformaldehyde in cytoskeleton buffer for 10 min at 4 °C, permeabilized in 0.5% Triton X-100 in warm (37 °C) cytoskeleton buffer at room temperature, washed twice with PBS at room temperature, and immunostained.

Images were collected by using a super-resolution SP8 LIGHT-NING laser-scanning spectral confocal microscope (Leica Micro-systems), LASX software (Leica Microsystems), and a 63 × Apochromat 1.40 NA oil objective. The low-fluorescence immersion oil (11513859; Leica Microsystems) was used and imaging was performed at room temperature. Mean projections of image stacks were generated by using the LASX software. 3D reconstructions were generated from image stacks by using the LCS 3D tool.

## Quantification of fluorescence signals

Fluorescence intensity signals at actin patches or at the base of DNA bridges were quantified using the LCS Lite polygon tool by analyzing an image area of 40 µm² around the base of the DNA bridge, and intensity values were normalized versus background values obtained by analyzing a nearby identical area within the cell, by subtracting the background-signal value from the actin-patch or the DNA bridge-base value (Dandoulaki et al, 2018). After background subtraction, the average values from control or mutant samples were calculated and control or mutant samples values were divided with the average control value to obtain the relative intensity of the values plotted (i.e., relative to control = 1). Nesprin-2, Sun1 or Sun2 intensity signals at the nuclear membrane (front/back) were quantified using the LCS draw line tool by analyzing a line of 4.5 µm at the nuclear membrane "front" (i.e., towards the base of the DNA bridge) and a diametrically opposed line of equal length at the "back" of the nuclear membrane, and dividing the corresponding fluorescence intensity values (front/back). To measure the length ($x$) and width ($y$) of nuclei the LASX line tool was used. In nuclei with DNA bridges the longer ($x$) axis was extending parallel to the DNA bridge and the shorter ($y$) axis was perpendicular to the bridge, whereas in interphase nuclei without DNA bridges the $x$ and $y$ axes ($y$ perpendicular to $x$) were randomly selected. To analyze midbody thickness, the diameter of each microtubule bundle at the midbody was measured using the LASX line tool and the average value calculated. To measure the distance between actin-patches and focal adhesions, cell image stacks were obtained by confocal microscopy using a fixed 1 µm step and the distance between the

image plane containing the paxillin or integrin foci and the one containing the actin patches was calculated by counting the number of image planes between them. Because actin patches were usually detected in three image planes, the middle plane was used. To quantify paxillin or integrin intensity at the actin patch or bottom image plane, the LCS Lite polygon tool was used to analyze an image area of $40\,\mu m^2$ around each actin patch or directly below it. To measure paxillin intensity at focal adhesions in cells without DNA bridges, the LCS Lite polygon tool was used to analyze an image area of $\sim 10\,\mu m^2$ at focal adhesions. To measure actin foci intensity at Lamin A:RFP granules, an image area was drawn around each RFP-positive nuclear granule using the LCS Lite polygon tool.

## Broken bridges

Cells were fixed and stained for immunofluorescence microscopy as appropriate. The percentage of broken chromatin (DAPI) bridges was calculated as follows: (number of bridges that appear broken/ total number of bridges) *100%. The entire bridge was photographed and a bridge was marked "broken" if a clear gap in DAPI staining of $>2\,\mu m$ appeared in all image contrasts.

## Purification of GST proteins

GST-tagged proteins were expressed in BL21 (DE3) cells (Agilent Technologies) and purified by using glutathione-agarose beads (Santa Cruz Biotechnology).

## In vitro transcription-translation and colorimetric detection of proteins

For in vitro transcription-translation, the TnT Quick Coupled Transcription/Translation System (Promega) was used. Briefly, $0.8\,\mu g$ plasmid DNA template was incubated with $40\,\mu l$ TnT T7 Quick Master Mix, $20\,\mu M$ Methionine and $2\,\mu l$ Transcend Biotin-Lysyl-tRNA (L1170, Promega) per $50\,\mu l$ reaction for 60 min at $30\,°C$. Biotin-labelled proteins from Transcend reactions or after GST pull-downs were analyzed by SDS PAGE, followed by transfer onto nitrocellulose membrane (Amersham Protran Premium 0.45 NC, Cat #10600003; GE Healthcare), incubation with streptavidine-alkaline phosphatase (Promega) for 2 h at room temperature, and colorimetric detection of the bands using Western Blue (Promega). Bands were documented with the Sapphire Biomolecular Imager (Azure Biosystems).

## Preparation of cell lysates

Cells were sonicated three times for 10 s in ice-cold immunoprecipitation/kinase buffer (50 mM Hepes, pH 7.5, 150 mM NaCl, 1 mM EDTA, 2.5 mM EGTA, 10% glycerol, 0.1% Tween 20, 0.1 mM PMSF, $10\,\mu g/ml$ leupeptin, $10\,\mu g/ml$ aprotinin, 1 mM sodium fluoride, 10 mM sodium β-glycerophosphate, and 0.1 mM sodium vanadate) and incubated for another 30 min on ice.

## GST pull-down and coimmunoprecipitation

For GST pull-downs from cell lysates, 1 mg cell lysate was incubated with $1\,\mu g$ GST protein on glutathione-agarose beads for 4 h at $4\,°C$. Samples were spun down, washed three times with immunoprecipitation/kinase buffer, and immunoprecipitated proteins on agarose beads were analyzed by SDS PAGE and Western blotting. For GST pull-downs from in vitro transcribed-translated proteins, $20\,\mu l$ Transcend reaction was incubated with $1\,\mu g$ GST protein on glutathione-agarose beads for 4 h at $4\,°C$. Samples were spun down, washed three times with immunoprecipitation/kinase buffer, and immunoprecipitated proteins on agarose beads were analyzed by SDS PAGE and colorimetric detection.

For coimmunoprecipitation, 2 mg cell lysate was incubated with $2\,\mu g$ anti-Nesprin-2 Ab2 for 16 h followed by the addition of $10\,\mu l$ protein A/ G PLUS–agarose beads (Santa Cruz Biotechnology) for 1 h at $4\,°C$.

## Western blotting

Cells were lysed in ice-cold, whole-cell extract buffer (20 mM Hepes, 5 mM EDTA, 10 mM EGTA, 0.4 M KCl, 0.4% Triton X-100, 10% glycerol, 5 mM NaF, 1 mM DTT, $5\,\mu g/ml$ leupeptin, $50\,\mu g/ml$ PMSF, 1 mM benzamidine, $5\,\mu g/ml$ aprotinin, and 1 mM $Na_3VO_4$) for 30 min on ice. Lysates were cleared by centrifugation at $15,000 \times g$ for 10 min at $4\,°C$. Samples were then analyzed by SDS-PAGE, transferred onto nitrocellulose membrane (Amersham Protran Premium 0.45 NC, Cat #10600003; GE Healthcare), and incubated with the appropriate antibodies. Secondary antibodies were detected by chemiluminescence (Clarity Western ECL Substrate, Cat #1705061; Biorad) and documented with the Sapphire Biomolecular Imager (Azure Biosystems).

## Densitometry

Densitometric analysis of bands was performed using ImageJ.

## Statistical analysis and repetitions

For fluorescence intensities at actin patches or at the bases of the intercellular canals, a minimum of 15 cells per experiment from two independent experiments were analyzed per treatment ($n > 30$) and SD was calculated. For the frequency of broken or intact loose chromatin (DAPI) bridges, typically 15–20 cells with bridges per experiment from three independent experiments were scored blindly and SD was calculated. For Nesprin-2, Sun1 or Sun2 front/back fluorescence intensity at the nuclear membrane, a minimum of 20 cells per experiment from two independent experiments were analyzed per treatment ($n > 40$) and SD was calculated. For distances from actin patches to focal adhesions, 15–20 cells per experiment from two independent experiments were analyzed per treatment ($n \geq 38$) and SD was calculated. For nuclear chromatin shape calculations, at least 30 cells (typically more than 50 cells) per experiment from two independent experiments were analyzed per treatment and SD was calculated. For Lamin A, Sun1 or Sun2 nuclear lines, at least 20 cells per experiment from three independent experiments were analyzed per treatment and SD was calculated. For paxillin or integrin intensity at the actin patch or bottom image plane, at least 30 cells per experiment from two independent experiments were analyzed and the SD was calculated. For paxillin intensity at focal adhesions in cells without DNA bridges, approximately 15 focal adhesions per cell from at least 30 cells per experiment from two independent experiments were analyzed per treatment ($n > 450$) and the SD was

calculated. For actin foci intensity at Lamin A:RFP granules, ~20 granules per cell from at least 20 cells per experiment from two independent experiments were analyzed per treatment ($n > 800$) and the SD was calculated. Statistically significant differences among three or more groups were determined by one-way ANOVA followed by two-tailed Students' $t$ test between two groups. No statistical method was used to predetermine the sample size. Blots were done twice and representative gels are shown.

## Data availability

Newly created materials and reagents are available upon request. The cDNA sequence of human Nesprin-2 Spectrin Repeats 3-54 from this publication has been deposited to GenBank database and assigned the accession number PQ453063.

The source data of this paper are collected in the following database record: biostudies:S-SCDT-10_1038-S44318-025-00565-3.

## Peer review information

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

## Acknowledgements

We thank Panayiotis Theodoropoulos and George Garinis for helpful discussions. We also thank Nicolas Borghi, Daniel Gerlich, Mike Olson, Oliver Rocks, Paris Skouridis and Gareth Thomas for generously sharing reagents. This work was supported by Worldwide Cancer Research (Project 25-0103) and by Fondation Santé. G Zachos was supported by the H.F.R.I. under the "2nd Call for H.F.R.I. Research Projects to support Faculty Members and Researchers" (Project Number: 2486). S. Balafouti and E. Petsalaki were supported by the Hellenic Foundation for Research and Innovation (H.F.R.I.) under the "2nd Call for H.F.R.I. Research Projects to support Post-Doctoral Researchers" (Project Number: 629).

## Author contributions

**Sofia Balafouti:** Investigation; Methodology; Writing—review and editing. **Maria Kabouraki:** Investigation; Performed GST pull-downs mapping the PDZ RhoGEF protein domain that interacts with GST-spectrin repeats. **George Zachos:** Conceptualization; Supervision; Funding acquisition; Validation; Visualization; Writing—original draft; Project administration; Writing—review and editing. **Eleni Petsalaki:** Conceptualization; Supervision; Funding acquisition; Investigation; Methodology; Writing—original draft; Project administration; Writing—review and editing.

Source data underlying figure panels in this paper may have individual authorship assigned. Where available, figure panel/source data authorship is listed in the following database record: biostudies:S-SCDT-10_1038-S44318-025-00565-3.

## Disclosure and competing interests statement

The authors declare no competing interests.

