## [Peer Review File · The EMBO Journal]

Tension-sensitive LINC-RhoA signaling prevents chromatin bridge breakage in cytokinesis

Sofia Balafouti, Maria Kabouraki, George Zachos, and Eleni Petsalaki

Corresponding author(s): George Zachos (gzachos@uoc.gr) , Eleni Petsalaki (grad600@edu.biology.uoc.gr)

Review Timeline:

Submission Date:	9th Dec 24
Editorial Decision:	7th Feb 25
Revision Received:	27th Jun 25
Editorial Decision:	29th Jul 25
Revision Received:	8th Aug 25
Accepted:	29th Aug 25

Editor: Hartmut Vodermaier

Transaction Report:

Dear Dr. Zachos,

Thank you for submitting your manuscript on nucleus-linked mechanosensing of chromatin bridges during cytokinesis to The EMBO Journal. I apologize for the delay in getting back to you with a decision, as it had been very difficult to assign appropriate reviewers in the busy period around the end of the year. I am contacting you now with two very thorough reports from experts in cytoskeletal/abscission processes and LINC complexes, which you will find copied below. As you will see, both reviewers acknowledge the potential interest of your findings, but remain unconvinced that the main conclusions of the study are already sufficiently supported by the presented experiments. Among the key concerns are absence of crucial controls (including for RNAi knockdowns), issues with literature presentation and experimental rationales, inconclusive data related to Nesprin-2, or the importance of live-cell imaging to complement the fixed-cell stainings.

Since we normally only consider a single round of major revision, I would at this point invite you to carefully consider the referee reports with your coworkers and to prepare a tentative point-by-point response, detailing how you might envision addressing the key concerns of the referees in case you should be given the opportunity to revise this work for The EMBO Journal. Based on this response (which I may share and discuss with some of the referees) and possible follow-up discussion, we could then determine whether it would be warranted and promising to invite a major revision for The EMBO Journal. It would be great if you could get back to me with such a revision plan by early next week.

Looking forward to hearing from you,

Best regards,

Hartmut Vodermaier

Referee #1 (Report for Author)

In this manuscript, Balafouti et al. address the role of the LINC complex in the formation of actin patches at the base of chromatin bridges that can be abnormally present during

cytokinesis. Previous work from the Zachos lab and others indicates that actin polymerization/ actin patches prevent chromatin breakage and presumably promote genome stability. However, the mechanisms that induce the formation of actin patches close to the nucleus, at the base of chromatin bridges are unknown. Here, the authors propose a mechanotransduction-based model. They show that actin patches are generated by local RhoA-ROCK-LIMK-mDia1-pCofilin-1 signaling. They found that chromatin bridges are associated with nuclear envelop stretching, which is associated with the asymmetric localization of the LINC complex proximal to the chromatin bridge. There, the SR domain of Nesprin-2 recruits (through direct interaction) the PDZ RhoA GEF to locally activate RhoA and generate actin patches. Perturbing any of these components leads to defective patch formation and is associated with chromatin breakage. Using elegant fusion protein strategies, it is concluded that local activation of RhoA by this molecular cascade generates actin patches that prevent chromosome breakage.

The paper is very well written, and all the steps are generally well demonstrated. It is arguably a bit overwhelming (> 150 panels...), but overall, the conclusions are well supported by the data. In my opinion, the model is very interesting, and the manuscript deserves to be published in the EMBO Journal. However, there are several issues that should be addressed, including important controls. As most of the data are only supported by fixed cell analysis, key results should be confirmed by live cell imaging (see detailed comments below).

Major comments

1. The whole model is based on the local activation of RhoA/ROCK/LIMK/mDia1 at the bases of chromatin bridges, in the regions of the cell close to the intercellular canal. According to the authors, this is due to the very local recruitment of the PDZ RhoA GEF by the LINC complex, precisely where the nuclear envelope is stretched. It is therefore difficult to understand how activated ROCK delta2 (Fig. 2) and activated LIMK T508EE (Fig. 3) can properly localize to the chromatin bridge bases -and thus rescue the formation of local actin patches and consequently chromatin breakage- when RhoA is inhibited. Activated LIMK and ROCK are cytosolic and should induce actin polymerization throughout the cytoplasm. In contrast, they induce the formation actin patches only where they are expected, while RhoA-GTP which normally directs ROCK/LIMK activity locally is absent. Similarly, the expression of activated RhoA G14V in the absence of the PDZ RhoGEF has no reason to induce the local polymerization of actin patches at the bases of the chromatin bridges (Fig. 4). The point of Figure 7 is to show that the GEF domain must be localized

where the LINC complex is locally enriched. As shown by others, the expression of activated RhoA should induce the formation of actin stress fibers throughout the cell. Similar questions arise in Fig. 4l/m/n (activated Src rescue), Fig. 6e/f (siSUN1/2 rescue) and Fig. 7a-f (siLaminA/C rescue). The authors must solve this conundrum as local activation of the RhoA pathway is at the center of the proposed model.

2. In the same vein, the authors use an elegant fusion strategy to show that local GEF activity at the enriched LINC complex is sufficient to restore the formation of local actin patches and to prevent chromatin breakage (Fig. 7). To conclude that local RhoA activation is key for actin patch formation, an important control is missing. The authors should show that the loss of actin patches and the broken chromatin bridges observed upon Nesprin-2 depletion are not rescued by (over)expression of full-length (unfused) PDZ GEF or DHPH alone, which should be diffusely localized.

3. Almost all the conclusions (especially rescue experiments) are based on quantification of "broken bridges" from fixed cells. Chromatin bridges are extremely stretched, over tens of micrometers in BE cells (e.g. Fig. 1a), something which is not observed in other cell types. It is important to rule out that these thin and long structures are not artefactually broken during the process of fixation, with the hypothesis that these stretched chromatin bridges are more fragile in the absence of actin patches. The authors should therefore confirm their main findings using live cell imaging of BE cells (not HeLa cells, as shown in the movies presented) and quantify the percentage of broken bridges and time of breakage (as in Fig. 1i), in particular upon RhoA inhibition, ROCK inhibition (or mDia1+LIMK1 inhibition) and Nesprin2 depletion.

4. Several important controls are lacking:

- In Fig. 1a/b (F-actin), Fig. 2a/b (ROCK), Fig. 2d/e (pLIMK), Fig. 2g/h/i (F-actin), Fig. 3g/h/i (pCofilin), Fig. 3k/l (mDia1), Fig. 3n and Fig. 6a/b (RhoA), the authors show images and quantifications of staining in situations that cannot be compared (presence vs. absence of chromatin bridges in control vs. perturbed situations, respectively). It is therefore important to also provide localization and quantification of this staining only when chromatin bridges are present.

- Fig. 1c: Actin accumulation is seen very close to the broken chromatin bridge. Why isn't this considered as an actin patch? What is exact the definition of actin patches (see point #5 below)?

- Fig. 1o: The cell on the bottom left shows actin patches, but there is no chromatin bridge. It is therefore important to show that in dividing cells with no chromatin bridges, there are no actin patches close to the intercellular canal. Please provide images and quantification of actin patches in cytokinetic cells (tubulin positive bridge) with no chromatin bridges (LAP2 negative).

- In figure 5, the "no bridge" controls are not appropriate controls as they correspond to interphase cells but not to dividing cells with no chromatin bridges. No doubt that interphase cells have round-shape nuclei. Throughout this figure, please provide quantifications of nuclei/staining of cells undergoing cytokinesis but without chromatin bridges (e.g. ac-tubulin positive and LAP2 negative canals). The authors need to exclude that telophase nuclei do not have asymmetric LINC complexes nor elongated shape in the absence of chromatin bridges.

5. A better description of the structure/organization of the actin patches should be provided. 3D reconstruction of z-stacks with higher resolution microscopy to visualize the nuclear envelope vs. the actin cortex/plasma membrane should be provided. Are the actin patches all around chromatin bridges? Rather, they seem to point outwards and to deform the plasma membrane, making microvilli/microspikes. This is important, because there is no discussion of how actin patches could protect against chromosome breakage.

6. Related to the point above: do nuclei experience more tension in the absence of actin patches? Perhaps actin patches buffer the forces exerted on the nuclei by chromatin bridges, preventing breakage. Quantifying differences in the number of SUN2 lines per nucleus after siRhoA or siPDZ should answer this question.

7. I do not find the conclusion based on the DN VPS4 conclusive, as this mutant not only inhibits abscission but also perturbs nuclear envelope reformation. The live cell imaging mentioned in point #3 should provide better evidence that the chromatin bridges are broken by longitudinal tension, rather than by membrane abscission.

Minor comments:

1. The role of the actin-binding domain of the LINC complex in generating nuclear tension is unclear. It seems to this reviewer that the LINC complex distal to the chromatin bridges couples the nuclei to the cell cortex. As the dividing cell migrates/spreads, cortical forces are transmitted to the nuclei and chromatin bridges are pulled by the opposite nuclei. This

point should at least be discussed.

2. Fig. 6b, 6n and 6o: In contrast to control cells, F-actin is much more abundant along the intercellular canals when the LINC complex is perturbed. Can the authors speculate why?

3. Discussion/Introduction: The present results should be better discussed in the context of the previously published literature. In particular:

- How do the results fit with the regulation of actin levels by actin oxidation/reduction (PMID 32029597)?

- Why broken bridges here and not canal reopening in the absence of actin patches (see PMID 19203582, 26929449, PMID 32029597)?

- What is the link between this pathway and the activated NoCut checkpoint?

- How do these results fit with the Pellman paper showing that actomyosin II is required for chromatin breakage (PMID: 32299917)? According to this published paper, RhoA inhibition should prevent myosin II activation and thus chromatin breakage. Here, the opposite is seen (Fig. 1). This should at least be discussed

4. In several quantifications (Fig. 1d, 2c, 2f, 2j, 2q etc.), I assume that intensities from different independent experiments have been pooled. It would be better to present the results as mean +/- SD for independent experiments, as in the other quantifications. If these quantifications have been done only once, results from independent replicates should be reported.

Referee #2 (Report for Author)

This manuscript explores an interesting potential mechanism linking nuclear tension during cytokinesis to actin patch formation through LINC complex-mediated RhoA signaling. While the biological phenomenon is intriguing and potentially important for understanding genome stability, there are significant technical issues that undermine the authors' central conclusions.

The manuscript contains numerous technical oversights and terminology issues that need addressing. Several key abbreviations require proper definition, including SUN (Sad1/UNC-84), NESPRIN (nuclear envelope spectrin repeat-enriched protein), KASH (Klarsicht/ANC-1/SYNE homology), PDZ (PSD-95/DGL/ZO-1), LIMK (LIN-11/Isl-1/MEC-3 kinase), and ROCK (Rho-associated protein kinase). Technical corrections are needed, such as changing "N-terminal actin-binding calponin homology (CH) domain" to "N-terminal tandem actin-binding calponin homology (CH) domains" (Page 4, Lines 9-10). The authors should also

note that small Rho GTPases regulate microtubule dynamics in addition to actin organization (PMID: 24691223 and 24988197).

Several critical controls are missing throughout the study. The authors rely heavily on RNAi experiments but fail to include rescue controls for most of their knockdowns, making it impossible to rule out off-target effects. While they demonstrate rescue with GFP:RhoAres for RhoA-depletion, similar controls are notably absent for other key proteins. The DN-KASH experiments lack the essential KASH- Δ L control, a construct based off DN-KASH but lacking the luminal SUN protein-binding KASH domain (PMID: 20724637), which is critical for demonstrating specificity of LINC complex disruption versus general nuclear envelope perturbation. Vehicle controls are missing for small molecule experiments, and a wild-type GFP:Vps4 control should be included. The authors should also consider examining FHOD1 or FHOD3 localization at the actin patches, given that nesprin-2G directly interacts with FHOD1 in other contexts (PMID: 24880667, 32460023).

There are several significant concerns regarding the characterization and presentation of nesprin-2-related experiments. The authors propose that PDZ-RhoGEF activates RhoA to promote actin patch formation, but no direct evidence supports this conclusion. They should demonstrate that actin RhoA localizes to actin patches in a PDZ-RhoGEF-dependent manner, perhaps using a FRET-based biosensor. The manuscript fails to consider the different isoforms of nesprin-2 expressed in their experimental cell lines, which is particularly important given that their anti-nesprin-2 antibody (Abcam) recognizes an epitope within amino acids 4200-4350 of the SR-containing cytoplasmic domain. The authors should verify whether full-length nesprin-2G displays similar localization patterns. Additionally, the authors should adopt standard field nomenclature, referring to their "short" nesprin-2 constructs lacking SRs 3-54 as "mini-nesprin-2G" constructs. Key references for these constructs are missing, including their first description (PMID: 19843581) and initial functional demonstration (PMID: 20724637). Similarly, the I128,131A mutation in the actin-binding CH domains needs proper definition and citation (PMID: 20724637). Importantly, this mutation impairs but does not completely abolish actin binding, contrary to what is depicted in Figure 6G.

The reagents used raise significant concerns. The lamin A/C-RFP construct shows an abnormal localization pattern reminiscent of non-farnesylated progerin (PMID: 22895092), suggesting the C-terminal RFP tag may be disrupting proper lamin processing and assembly. An N-terminal tag, which is well-established in the field, should be used instead. The biochemical evidence for interaction between PDZ-RhoGEF and nesprin-2 SRs is compromised by extensive protein degradation products. It would be valuable to

demonstrate whether endogenous nesprin-2 interacts with endogenous PDZ-RhoGEF via co-immunoprecipitation. The authors should also note that the formin inhibitor SMIFH2 is not specific for formins as it can inhibit members of the myosin superfamily (PMID: 33589498), which might alter the interpretation of their findings.

The manuscript also overlooks important literature and contains inaccurate statements. The authors should reference previous work showing that LINC complexes regulate small Rho GTPase signaling (PMID: 28035049) and that nesprin-2G promotes nuclear envelope-associated actin meshwork assembly during prophase (PMID: 31264963). Their description of TAN lines needs correction - these are specifically linear arrays of nesprin-2G/SUN2-containing LINC complex that form along dorsal perinuclear actin cables perpendicular to the direction of cell migration, and do not include SUN1 (PMID: 20724637). Moreover, TAN lines are not differentiated from "actin caps" as being longitudinal or transverse, respectively. Further, the statement that "LINC complex promotes generation of mechanical tension" is inaccurate - LINC complexes transmit, rather than generate, mechanical tension. Maybe they can store tension.

In summary, while this work presents an interesting biological observation, the technical issues significantly impact the reliability of the core mechanistic conclusions. Major revisions addressing these concerns would be needed before this manuscript would be suitable for publication in EMBO Journal. The authors should also consider examining whether increased nuclear stiffness (e.g., through progerin expression) affects nesprin-2 accumulation at chromatin bridges and investigate whether SRs 31-17 directly interact with PDZ RhoGEF through a conserved motif.

Referee #1

In this manuscript, Balafouti et al. address the role of the LINC complex in the formation of actin patches at the base of chromatin bridges that can be abnormally present during cytokinesis. Previous work from the Zachos lab and others indicates that actin polymerization/ actin patches prevent chromatin breakage and presumably promote genome stability. However, the mechanisms that induce the formation of actin patches close to the nucleus, at the base of chromatin bridges are unknown. Here, the authors propose a mechanotransduction-based model. They show that actin patches are generated by local RhoA-ROCK-LIMK-mDia1-pCofilin-1 signaling. They found that chromatin bridges are associated with nuclear envelop stretching, which is associated with the asymmetric localization of the LINC complex proximal to the chromatin bridge. There, the SR domain of Nesprin-2 recruits (through direct interaction) the PDZ RhoA GEF to locally activate RhoA and generate actin patches. Perturbing any of these components leads to defective patch formation and is associated with chromatin breakage. Using elegant fusion protein strategies, it is concluded that local activation of RhoA by this molecular cascade generates actin patches that prevent chromosome breakage.

The paper is very well written, and all the steps are generally well demonstrated. It is arguably a bit overwhelming (> 150 panels...), but overall, the conclusions are well supported by the data. In my opinion, the model is very interesting, and the manuscript deserves to be published in the EMBO Journal. However, there are several issues that should be addressed, including important controls. As most of the data are only supported by fixed cell analysis, key results should be confirmed by live cell imaging (see detailed comments below).

Major comments

1. The whole model is based on the local activation of RhoA/ROCK/LIMK/mDia1 at the bases of chromatin bridges, in the regions of the cell close to the intercellular canal. According to the authors, this is due to the very local recruitment of the PDZ RhoA GEF by the LINC complex, precisely where the nuclear envelope is stretched. It is therefore difficult to understand how activated ROCK delta2 (Fig. 2) and activated LIMK T508EE (Fig. 3) can properly localize to the chromatin bridge bases -and thus rescue the formation of local actin patches and consequently chromatin breakage- when RhoA is inhibited. Activated LIMK and ROCK are cytosolic and should induce actin polymerization throughout the cytoplasm. In contrast, they induce the formation actin patches only where they are expected, while RhoA-GTP

which normally directs ROCK/LIMK activity locally is absent. Similarly, the expression of activated RhoA G14V in the absence of the PDZ RhoGEF has no reason to induce the local polymerization of actin patches at the bases of the chromatin bridges (Fig. 4). The point of Figure 7 is to show that the GEF domain must be localized where the LINC complex is locally enriched. As shown by others, the expression of activated RhoA should induce the formation of actin stress fibers throughout the cell. Similar questions arise in Fig. 4l/m/n (activated Src rescue), Fig. 6e/f (siSUN1/2 rescue) and Fig. 7a-f (siLaminA/C rescue). The authors must solve this conundrum as local activation of the RhoA pathway is at the center of the proposed model.

Depletion of RhoA does not impair nuclear tension in cells with intact chromatin bridges compared with controls (Fig. 5n, o) and Nesprin-2 is expected to accumulate at the base of the intercellular canal in RhoA-depleted cells, similar with controls. One possibility is that the remaining endogenous RhoA at the base of the intercellular canal in cells transfected with siRhoA (Supplementary Fig. 1a), although it is not sufficient to support actin patch formation by locally activating the endogenous ROCK and LIMK, can still target the constitutively active ROCK or LIMK proteins to the base of the intercellular canal to promote actin patches. Also, depletion of Sun1/2 or Lamin A/C reduces, but does not completely abolish, nuclear tension in cytokinesis with intact chromatin bridges (Fig. 6p). Therefore, one possibility is that, in Sun1/2 or Lamin A/C-depleted cells, the transfected constitutively active RhoA associates (through PDZ) with the remaining Nesprin-2 at the base of the intercellular canal to promote actin patches, whereas the endogenous RhoA is not sufficient to support actin patch formation under these conditions, in the absence of a relatively pronounced Nesprin-2 accumulation. We will include this in the Discussion of our paper, as suggested by the reviewer.

2. In the same vein, the authors use an elegant fusion strategy to show that local GEF activity at the enriched LINC complex is sufficient to restore the formation of local actin patches and to prevent chromatin breakage (Fig. 7). To conclude that local RhoA activation is key for actin patch formation, an important control is missing. The authors should show that the loss of actin patches and the broken chromatin bridges observed upon Nesprin-2 depletion are not rescued by (over)expression of full-length (unfused) PDZ GEF or DHPH alone, which should be diffusely localized.

We will investigate actin patches and broken chromatin bridges in cells transfected with DHPH after Nesprin-2 depletion, as requested by the

reviewer. If local RhoA activation at the base of chromatin bridges is required for actin patch formation as predicted by our model, we expect that cells transfected with DPH will exhibit impaired actin patches and increased chromatin bridge-breakage compared with controls, similar with cells expressing the CH* or CH*:DPH proteins that cannot bind to actin (Fig. 7i, j).

3. Almost all the conclusions (especially rescue experiments) are based on quantification of "broken bridges" from fixed cells. Chromatin bridges are extremely stretched, over tens of micrometers in BE cells (e.g. Fig. 1a), something which is not observed in other cell types. It is important to rule out that these thin and long structures are not artefactually broken during the process of fixation, with the hypothesis that these stretched chromatin bridges are more fragile in the absence of actin patches. The authors should therefore confirm their main findings using live cell imaging of BE cells (not HeLa cells, as shown in the movies presented) and quantify the percentage of broken bridges and time of breakage (as in Fig. 1i), in particular upon RhoA inhibition, ROCK inhibition (or mDia1+LIMK1 inhibition) and Nesprin2 depletion.

The reviewer asks whether, in BE cells with relatively long chromatin bridges, loss of actin patches makes stretched bridges more fragile to breaking during fixation, instead of directly causing them to break. We believe this is unlikely because in HeLa cells exhibiting much shorter chromatin bridges, loss of actin patches associated with chromatin bridge-breakage by live-cell microscopy, thus showing that loss of actin patches causes DNA bridges to break. The reviewer proposes us to perform live-cell imaging analysis of BE cells, to complement our live-cell imaging analysis of HeLa cells. We have repeatedly tried to generate BE stable cell lines (e.g., BE cells expressing H2B:GFP, Lap2b:RFP, etc) for time-lapse microscopy purposes, but without success. We will therefore use the BioTracker 488 Green Nuclear Dye (#SCT120, Sigma) for live-cell imaging of BE cells, to quantify the percentage of broken DNA bridges and the time of breakage upon treatment with the RhoA-inhibitor, the ROCK-inhibitor or after Nesprin-2-depletion by siRNA, as suggested by the reviewer.

4. Several important controls are lacking:

- In Fig. 1a/b (F-actin), Fig. 2a/b (ROCK), Fig. 2d/e (pLIMK), Fig. 2g/h/i (F-actin), Fig. 3g/h/i (pCofilin), Fig. 3k/l (mDia1), Fig. 3n and Fig. 6a/b

(RhoA), the authors show images and quantifications of staining in situations that cannot be compared (presence vs. absence of chromatin bridges in control vs. perturbed situations, respectively). It is therefore important to also provide localization and quantification of this staining only when chromatin bridges are present.

In all the above stainings, mutant cells exhibited reduced localization of the corresponding proteins at the base of the intercellular canal compared with controls, regardless of their bridges been intact or broken. We will provide images and quantifications of mutant cells with intact DNA bridges to complement our analysis, as requested by the reviewer.

- Fig. 1c: Actin accumulation is seen very close to the broken chromatin bridge. Why isn't this considered as an actin patch? What is exact the definition of actin patches (see point #5 below)?

Actin patches are structures of polymerized actin at the base of the cytoplasmic canal that connects the two daughter cells, in cytokinesis with chromatin bridges (Steigemann, 2009). The actin accumulation in Fig. 1c appears inside the intercellular canal, and not at the base of the chromatin bridge (please also see minor comment 3.1). We will provide a more representative image of RhoAi cells to avoid confusion.

- Fig. 1o: The cell on the bottom left shows actin patches, but there is no chromatin bridge. It is therefore important to show that in dividing cells with no chromatin bridges, there are no actin patches close to the intercellular canal. Please provide images and quantification of actin patches in cytokinetic cells (tubulin positive bridge) with no chromatin bridges (LAP2 negative).

Control cells without chromatin bridges do not exhibit actin patches close to the intercellular canal. We will quantify actin intensity at the base of the intercellular canal in BE cells in late cytokinesis, i.e., in cells with decondensed chromosomes connected by thin "late" midbodies (PMID:33355621, PMID: 27126587), in the absence of DNA bridges, as requested.

- In figure 5, the "no bridge" controls are not appropriate controls as they correspond to interphase cells but not to dividing cells with no chromatin bridges. No doubt that interphase cells have round-shape nuclei. Throughout this figure, please provide quantifications of nuclei/staining of cells undergoing cytokinesis but without chromatin bridges (e.g. ac-tubulin positive and LAP2 negative canals). The authors need to exclude that telophase nuclei do not have asymmetric LINC complexes nor elongated shape in the absence of chromatin bridges.

We will quantify nuclear shape and Nesprin-2 localization in BE cells in late cytokinesis, i.e., in cells with decondensed chromosomes connected by thin "late" midbodies (PMID:33355621, PMID: 27126587), in the absence of DNA bridges, as requested by the reviewer.

5. A better description of the structure/organization of the actin patches should be provided. 3D reconstruction of z-stacks with higher resolution microscopy to visualize the nuclear envelope vs. the actin cortex/plasma membrane should be provided. Are the actin patches all around chromatin bridges? Rather, they seem to point outwards and to deform the plasma membrane, making microvilli/microspikes. This is important, because there is no discussion of how actin patches could protect against chromosome breakage.

Actin patches are around chromatin bridges. A 3D reconstruction of z-stacks of the DNA bridge and actin patches will be provided, as requested.

The mechanism by which actin patches protect against chromosome breakage is currently unknown. One possibility is that the dense structure of actin patches increases the stiffness and elasticity at the base of chromatin bridges, thus providing local mechanical support to prevent bridges from breaking (PMID: 20110992). This point will be included in the Discussion section of our paper.

6. Related to the point above: do nuclei experience more tension in the absence of actin patches? Perhaps actin patches buffer the forces exerted on the nuclei by chromatin bridges, preventing breakage. Quantifying differences in the number of SUN2 lines per nucleus after

siRhoA or siPDZ should answer this question.

In the presence of intact DNA bridges, cells depleted of RhoA exhibited deformed nuclear shape and Sun2 nuclear lines similar with controls (Fig. 5n,o), suggesting that nuclei do not experience more tension in the absence of actin patches. Instead, actin patches could offer local mechanical support to DNA bridges to prevent them from breaking (see also point #5 above). This point will be included in the Discussion section of our paper (also see previous comment above).

7. I do not find the conclusion based on the DN VPS4 conclusive, as this mutant not only inhibits abscission but also perturbs nuclear envelope reformation. The live cell imaging mentioned in point #3 should provide better evidence that the chromatin bridges are broken by longitudinal tension, rather than by membrane abscission.

The purpose of the dnVps4 experiment was to demonstrate that chromatin breakage in RhoA-deficient cells was not caused by abscission. However, because dnVps4 can potentially interfere with nuclear envelope formation and dynamics, we propose to remove this experiment from our revised manuscript. Because 70% of RhoA-depleted cells with broken DNA bridges exhibited intact intercellular canals as evidenced by membrane staining with the FM1-43FX dye (Fig. 1l,n and Supplementary Fig. 1o,p), our results from fixed cells still support that chromatin breakage in RhoA-deficient cells correlates with impaired actin patches and is not caused by premature abscission.

Minor comments:

1. The role of the actin-binding domain of the LINC complex in generating nuclear tension is unclear. It seems to this reviewer that the LINC complex distal to the chromatin bridges couples the nuclei to the cell cortex. As the dividing cell migrates/spreads, cortical forces are transmitted to the nuclei and chromatin bridges are pulled by the opposite nuclei. This point should at least be discussed.

The exact role of the actin-binding domain of the LINC complex in nuclear tension in cytokinesis with chromatin bridges is unclear. One possibility is that LINC complexes distal to chromatin bridges couple the nuclei to the actin cytoskeleton and the cell cortex and that, as the dividing cell

migrates/spreads, cortical forces are transmitted to the nuclei while chromatin bridges are pulling the daughter nuclei towards the opposite direction, thus generating nuclear tension. This point will be discussed in paragraph 2, page 18, as suggested by the reviewer.

2. Fig. 6b, 6n and 6o: In contrast to control cells, F-actin is much more abundant along the intercellular canals when the LINC complex is perturbed. Can the authors speculate why?

Control cells with chromatin bridges typically exhibit long and thin intercellular canals that appear as narrow as the chromatin bridge. However, LINC-deficient cells with broken chromatin bridges often exhibit wider intercellular canals, perhaps because connections between membrane-bound proteins (e.g., anillin) and cortical microtubules around the midbody that anchor the plasma membrane to the midbody cell cortex (PMID: 15854913, PMID: 16461284) are disrupted; as a result, the actin signal can appear stronger.

3. Discussion/Introduction: The present results should be better discussed in the context of the previously published literature. In particular:

- How do the results fit with the regulation of actin levels by actin oxidation/reduction (PMID 32029597)?
- Why broken bridges here and not canal reopening in the absence of actin patches (see PMID 19203582, 26929449, PMID 32029597)?
- What is the link between this pathway and the activated NoCut checkpoint?
- How do these results fit with the Pellman paper showing that actomyosin II is required for chromatin breakage (PMID: 32299917)? According to this published paper, RhoA inhibition should prevent myosin II activation and thus chromatin breakage. Here, the opposite is seen (Fig. 1). This should at least be discussed

3.1. In the presence of chromatin bridges, cells prevent the depolymerization of actin filaments inside the intercellular canal that links the two daughter cells (PMID 32029597), by recruiting the human methionine sulfoxide reductase B2 (MsrB2) to the midbody. In turn, this pool of polymerized actin delays recruitment of ESCRT-III proteins at the abscission site and stabilizes the intercellular canal to prevent binucleation. This mechanism is distinct from the formation of actin patches that we describe, and will be mentioned in paragraph 3, page 3 of our Introduction, as suggested by the reviewer.

3.2. Why impaired actin polymerization/stabilization leads to chromatin bridge breakage in some cases (PMID 29954829) but furrow regression in others (PMID 19203582, PMID 32029597) remains unclear and could depend on the experimental conditions. This point will be briefly discussed in the Discussion section, page 17 of our manuscript.

3.3 Whether actin patch formation is functionally linked to the activation of the abscission checkpoint is unknown. It may be interesting to examine whether cells with dicentric bridges that do not activate the abscission checkpoint (PMID 37638884) can still generate actin patches through the mechanism described in this paper. This point will be briefly discussed in the Discussion section, page 17 of our manuscript.

3.4 It was previously reported that actomyosin II activity is required for chromatin bridge breakage in cytokinesis (PMID: 32299917). Because RhoA inhibition could prevent myosin II activation, perhaps other Rho protein family members contribute to myosin II activation in RhoA-deficient cells with chromatin bridges (PMID: 36476233). Alternatively, longitudinal tension rather than contractile actomyosin forces may constitute the main cause of chromatin bridge breakage in RhoA-deficient cells. This point will be discussed in page 17 of our manuscript.

4. In several quantifications (Fig. 1d, 2c, 2f, 2j, 2q etc.), I assume that intensities from different independent experiments have been pooled. It would be better to present the results as mean +/- SD for independent experiments, as in the other quantifications. If these quantifications have been done only once, results from independent replicates should be reported.

Because chromatin bridges are relatively hard to find, multiple relatively small samples from at least two independent experiments, typically three or four experiments, were analyzed for each treatment. As a result, we decided to show the pooled experiments, otherwise each independent experiment would only contain a small number of bridges. Because the differences in intensities between control and mutant cells is very clear, we believe it wouldn't make a difference either way and propose to keep the quantifications as they are also for simplicity reasons.

Referee #2 (Report for Author)

This manuscript explores an interesting potential mechanism linking nuclear tension during cytokinesis to actin patch formation through LINC complex-mediated RhoA signaling. While the biological phenomenon is intriguing and potentially important for understanding genome stability, there are significant technical issues that undermine the authors' central conclusions.

The manuscript contains numerous technical oversights and terminology issues that need addressing. Several key abbreviations require proper definition, including SUN (Sad1/UNC-84), NESPRIN (nuclear envelope spectrin repeat-enriched protein), KASH (Klarsicht/ANC-1/SYNE homology), PDZ (PSD-95/DGL/ZO-1), LIMK (LIN-11/Isl-1/MEC-3 kinase), and ROCK (Rho-associated protein kinase). Technical corrections are needed, such as changing "N-terminal actin-binding calponin homology (CH) domain" to "N-terminal tandem actin-binding calponin homology (CH) domains" (Page 4, Lines 9-10). The authors should also note that small Rho GTPases regulate microtubule dynamics in addition to actin organization (PMID: 24691223 and 24988197).

Definitions for Sun, Nesprin, KASH, PDZ, LIMK and ROCK abbreviations will be provided at their first mentioning in the text, as requested. We will change "N-terminal actin-binding calponin homology (CH) domain" to "N-terminal tandem actin-binding calponin homology (CH) domains". We will also mention that, in addition to actin organization, small Rho GTPases regulate microtubule dynamics and will include reference PMID 24691223 in the Introduction, as requested by the reviewer.

Several critical controls are missing throughout the study. The authors rely heavily on RNAi experiments but fail to include rescue controls for most of their knockdowns, making it impossible to rule out off-target effects. While they demonstrate rescue with GFP:RhoARes for RhoA-depletion, similar controls are notably absent for other key proteins. The DN-KASH experiments lack the essential KASH- Δ L control, a construct based off DN-KASH but lacking the luminal SUN protein-binding KASH domain (PMID: 20724637), which is critical for demonstrating specificity of LINC complex disruption versus general nuclear envelope perturbation.

Vehicle controls are missing for small molecule experiments, and a wild-type GFP:Vps4 control should be included.

The authors should also consider examining FHOD1 or FHOD3 localization at the actin patches, given that nesprin-2G directly interacts with FHOD1 in other contexts (PMID: 24880667, 32460023).

We will generate GFP:PDZRes protein resistant to degradation by PDZ siRNA and demonstrate rescue of the phenotype (actin patches and chromatin bridge breakage) with GFP:PDZRes for PDZ-depletion, as requested by the reviewer. Rescue with GFP:RhoARes for RhoA-depletion has already been included. Rescue for Nesprin-2 depletion by transfection of full-length Nesprin-2Res isn't technically possible because of the huge size of the protein (~800 kDa), but CB offers a rescue of sorts. Furthermore, we will generate the KASH- Δ L control construct, as requested. We expect that expression of KASH- Δ L will not reduce actin patches and will not increase chromatin bridge breakage in cytokinesis compared with controls, demonstrating specificity of LINC complex disruption.

In time-lapse movies, control cells were treated with vehicle (i.e., with 10 μ M DMSO) and this will be clarified in the Methods section, relevant Figures and Legends. In fixed cells analysis, vehicle controls will be included for actin patches and broken chromatin bridges alongside the current siRNA controls in Figs 1d, 1e and 3p and also in Figs 2j, 3o and 3q. Also, vehicle controls were shown for ROCK (Fig. 2c), pLIMK-T508 (Fig. 2f), and pCofilin-S3 (Fig. 3j) and this will be clarified in the relevant Figures and Legends. Because expression of dnVps4 can potentially interfere with nuclear envelope formation and dynamics, we will remove the experiment with dnVps4 at reviewers' #1 suggestion.

We will also examine the potential localization of FHOD1, which interacts with Nesprin-2 (PMID:24880667), to actin patches in control cells with chromatin bridges by confocal microscopy analysis of fixed samples, as requested by the reviewer.

There are several significant concerns regarding the characterization and presentation of nesprin-2-related experiments. The authors propose that PDZ-RhoGEF activates RhoA to promote actin patch formation, but no direct evidence supports this conclusion. They should demonstrate that actin RhoA localizes to actin patches in a PDZ-RhoGEF-dependent manner, perhaps using a FRET-based biosensor.

The manuscript fails to consider the different isoforms of nesprin-2 expressed in their experimental cell lines, which is particularly important given that their anti-nesprin-2 antibody (Abcam) recognizes an epitope

within amino acids 4200-4350 of the SR-containing cytoplasmic domain. The authors should verify whether full-length nesprin-2G displays similar localization patterns.

Additionally, the authors should adopt standard field nomenclature, referring to their "short" nesprin-2 constructs lacking SRs 3-54 as "mini-nesprin-2G" constructs. Key references for these constructs are missing, including their first description (PMID: 19843581) and initial functional demonstration (PMID: 20724637). Similarly, the I128,131A mutation in the actin-binding CH domains needs proper definition and citation (PMID: 20724637). Importantly, this mutation impairs but does not completely abolish actin binding, contrary to what is depicted in Figure 6G.

To directly examine Rho activity, we will use the dimeric Tomato-2xrGBD Rho biosensor, which can visualize endogenous Rho activity with high spatial resolution (PMID: 34357388). To demonstrate that active Rho localizes to actin patches in a PDZ-RhoGEF-dependent manner, we will examine the localization of Tomato-2xrGBD Rho to actin patches in control and PDZ-depleted cells, as requested by the reviewer.

There are 13 Nesprin-2 isoforms produced by alternative splicing according to the Uniprot database (<https://www.uniprot.org/uniprotkb/Q8WXH0/entry#sequences>). Because most of them are truncated proteins, the reviewer wants us to verify that it is the full-length Nesprin-2 that is enriched at the base of chromatin bridges. Isoform 1 is the canonical sequence and isoform 2 has an insertion of additional 22 amino acids; therefore, they both represent full-length proteins. Isoforms 3, 4, 5, 6, 10, 11, 12 and 13 are missing the first (i.e., the C-terminal) approximately 6000 amino acids; therefore, they are not detected by the anti-Nesprin-2 antibody (ab204308) we use, which recognizes an epitope within amino acids 4200-4350. Isoforms 8 and 9 are missing amino acids 286-6885; therefore, they also are not detected by our antibody. Isoform 7 is missing amino acids 1-3638 and is the only truncated isoform that can be detected by the anti-Nesprin-2 antibody we use. Therefore, to verify enrichment of full-length Nesprin-2 at the base of chromatin bridges, localization of Nesprin-2 in cells without chromatin bridges, controls with chromatin bridges, or in Sun1/2-depleted cells with intact chromatin bridges will be examined using a second antibody (Abcam, ab233034) that is raised against amino acids 1-300 of Nesprin-2 and is not expected to recognize the truncated isoform #7, by confocal microscopy.

We will replace "short" Nesprin-2 with "mini-Nesprin-2G" throughout the paper and reference the PMID:19843581, PMID:20724637 and PMID:20724637 papers, as requested. We will also note "impaired actin binding" in Fig. 6G, bottom.

The reagents used raise significant concerns. The lamin A/C-RFP construct shows an abnormal localization pattern reminiscent of non-farnesylated progerin (PMID: 22895092), suggesting the C-terminal RFP tag may be disrupting proper lamin processing and assembly. An N-terminal tag, which is well-established in the field, should be used instead.

The biochemical evidence for interaction between PDZ-RhoGEF and nesprin-2 SRs is compromised by extensive protein degradation products. It would be valuable to demonstrate whether endogenous nesprin-2 interacts with endogenous PDZ-RhoGEF via co-immunoprecipitation.

The authors should also note that the formin inhibitor SMIFH2 is not specific for formins as it can inhibit members of the myosin superfamily (PMID: 33589498), which might alter the interpretation of their findings.

Cells expressing relatively low levels of Lamin A:RFP exhibited correct localization of Lamin A:RFP at the periphery of the nucleus (paragraph 3, page 15 and Supplementary Fig. 5p), suggesting that this protein was properly processed at least when expressed at low levels. Regardless of the exact reason behind the mislocalization of Lamin A:RFP (overexpression or improper processing under specific conditions), our purpose was to use it as a tool to engineer the mislocalization of the endogenous Nesprin-2 from the periphery of the nucleus in cells with chromatin bridges (PMID:15843432). By using this approach, we found that Nesprin-2 did not accumulate at the base of chromatin bridges, but co-localized with Lamin A:RFP at nuclear granules, and this correlated with impaired actin patches and with local polymerization of actin inside the Nesprin-2/Lamin A:RFP nuclear granules (Fig. 8). If the N-terminal RFP-tag of Lamin A does not generate Lamin A granules and, as a result, does not cause mislocalization of the endogenous Nesprin-2 in cells with chromatin bridges, it will not be useful for our purposes. Furthermore, we believe that investigating the exact cause of Lamin A:RFP-mislocalization in this construct is beyond the scope of our study. We propose to add the following sentence in paragraph 3, page 15, to note the potential causes of Lamin A:RFP mis-localization: "Mis-localization of Lamin A:RFP could be due to overexpression and/or improper processing of the protein under high levels of expression (PMID:22895092)". We will also investigate whether increased nuclear stiffness in cells expressing progerin:GFP (DOI:10.1039/c5sm00521c) reduces accumulation of Nesprin-2 at the base of intact chromatin bridges compared with controls expressing GFP-only, as another way to interfere with the localization of the endogenous Nesprin-2 (also see reviewer #2, last point). In this case, cells with chromatin bridges expressing progerin:GFP are expected to exhibit impaired actin patches compared with GFP-controls.

Because Nesprin-2 is a giant protein (~800 kDa), co-immunoprecipitations with the endogenous protein will be technically impossible to work. Instead, to further support the interaction of PDZ with Nesprin-2 SRs, we propose to investigate binding of the endogenous (not transfected) PDZ RhoGEF to GST-SRs(21-30) that is largely devoid of degradation products compared with GST-only, by GST-pull downs.

We will acknowledge in the text that SMIFH2 can also inhibit members of the myosin superfamily and cite the PMID: 33589498 paper, as suggested by the reviewer. To further investigate this, we will treat cells with the myosin II inhibitor blebbistatin: If treatment with blebbistatin reduces actin patches without interfering with nuclear shape deformation (x/y ratio) or with Nesprin-2 localization in cells with intact chromatin bridges, at least part of the SMIFH2 effect will be attributed to inhibition of myosin II.

The manuscript also overlooks important literature and contains inaccurate statements. The authors should reference previous work showing that LINC complexes regulate small Rho GTPase signaling (PMID: 28035049) and that nesprin-2G promotes nuclear envelope-associated actin meshwork assembly during prophase (PMID: 31264963). Their description of TAN lines needs correction - these are specifically linear arrays of nesprin-2G/SUN2-containing LINC complex that form along dorsal perinuclear actin cables perpendicular to the direction of cell migration, and do not include SUN1 (PMID: 20724637). Moreover, TAN lines are not differentiated from "actin caps" as being longitudinal or transverse, respectively. Further, the statement that "LINC complex promotes generation of mechanical tension" is inaccurate - LINC complexes transmit, rather than generate, mechanical tension. Maybe they can store tension.

Our data is consistent with previous findings that LINC complexes can promote RhoA activation and the assembly of an actomyosin network around the nucleus through transcription-independent mechanisms in different contexts (PMID: 28035049) (PMID: 31264963). This will be discussed in our paper, as suggested by the reviewer.

We thank the reviewer for these corrections. We will rewrite paragraph 2, page 18 of our manuscript, also taking into account the Minor comment #1 of reviewer 1: The exact role of the actin-binding domain of the LINC complex in nuclear tension in cytokinesis with chromatin bridges is unclear. It was previously shown that linear arrays of LINC complexes containing Nesprin-2G/Sun2 that form along dorsal perinuclear actin cables perpendicular to the direction of cell migration, transmembrane actin-associated nuclear (TAN) lines, allow the forces generated by the actin cytoskeleton to be transmitted

across the nuclear envelope to move the nucleus (PMID: 20724637). However, actin nuclear lines in cells with chromatin bridges were relatively less pronounced and were not organized in linear arrays. One possibility, is that LINC complexes distal to chromatin bridges couple the nuclei to the actin cytoskeleton and the cell cortex and that, as the dividing cell migrates/spreads, cortical forces are transmitted to the nuclei while chromatin bridges are pulling the daughter nuclei towards the opposite direction, thus generating nuclear tension.

Also, we will change: “the nuclear envelope Sun1/2-Nesprin-2 LINC complex promotes generation of mechanical tension on daughter nuclei by...”, to:

“the nuclear envelope Sun1/2-Nesprin-2 LINC complex promotes mechanical tension on daughter nuclei by...”, paragraph 2, page 16, as suggested by the reviewer.

In summary, while this work presents an interesting biological observation, the technical issues significantly impact the reliability of the core mechanistic conclusions. Major revisions addressing these concerns would be needed before this manuscript would be suitable for publication in EMBO Journal. The authors should also consider examining whether increased nuclear stiffness (e.g., through progerin expression) affects nesprin-2 accumulation at chromatin bridges and investigate whether SRs 31-17 directly interact with PDZ RhoGEF through a conserved motif.

We will investigate whether increased nuclear stiffness in cells expressing progerin:GFP (DOI: 10.1039/c5sm00521c) reduces accumulation of Nesprin-2 at the base of intact chromatin bridges compared with GFP-only controls, as suggested by the reviewer. In this case, cells with chromatin bridges expressing progerin:GFP are expected to exhibit impaired actin patches compared with GFP-controls.

How do Nesprin-2 SRs activate PDZ RhoGEF? One possibility is that SRs bind to the RGS-homology domain of PDZ RhoGEF to allosterically activate PDZ. Alternatively, Nesprin-2 SRs may act as scaffold to promote PDZ activation by G-proteins (discussed in paragraph 3, page 18). Mapping the binding regions of Nesprin-2 SRs on PDZ Rho GEF and investigating their significance for PDZ activation is perhaps beyond the scope of the present study and we propose to investigate this in a future study.

Prof. George Zachos
University of Crete
Department of Biology
Vassilika Vouton
Heraklion, Crete 70013
Greece

7th Feb 2025

Re: EMBOJ-2024-119872

Tension-sensitive LINC-RhoA signaling prevents chromatin breakage by generating actin patches in cytokinesis

Dear Dr. Zachos,

Thank you for your tentative point-by-point responses to the referee comments on your recent EMBO Journal submission. I have now had a chance to go through your proposal for how you would address them, and in general found it sufficiently promising to warrant inviting a revision of this work for The EMBO Journal. With regard to referee 1's concerns, I appreciate your plans for addressing them through additional experiments and controls, even though the outcome of these experiments and the referee's assessment of these results can obviously not be predicted at this stage. For the issues raised by referee 2, it will be key to include the missing RNAi rescues, to directly examine Rho activity, sorting our Nesprin-2 functions using additional antibodies and semi-endogenous co-IPs, and testing the effects of increased nuclear stiffness. On the other hand, I understand that Lamin-RFP was mainly intended for engineering Nesprin-2 mislocalization, and would tend to agree that detailed follow-up on the cause of Lamin-RFP mislocalization would be beyond the scope of the present study. Finally, in agreement with the referees, I would encourage you to carefully re-write key sections of the manuscript and having it proof-read by non-specialists to ensure broad accessibility of the study.

In summary, we shall be interested in further pursuing a new version of this study, revised along the lines suggested in your response letter, for EMBO Journal publication. Please keep in mind that it is our policy to allow only a single round of major revision, and please update me in case there should be any unexpected problems with the revisions, or if you should require an extension beyond the default 3-months deadline. As always, competing manuscript published during the course of this revision will not affect our final decision on your study. Finally, please note the detailed information and guidelines on how to prepare a revision below (and in our online Guide to Authors) - closely adhering to them shall greatly facilitate the editorial process at the time of resubmission.

Thank you again for the opportunity to consider this work, and I look forward to receiving your revision in due time.

With kind regards,

Hartmut Vodermaier

- size of the scale bars that are mandatory for all micrograph panels
- the statistical test used to generate error bars and P-values
- the type error bars (e.g., S.E.M., S.D.)
- the number (n) and nature (biological or technical replicate) of independent experiments underlying each data point
- Figures may not include error bars for experiments with $n < 3$; scatter plots showing individual data points should be used

instead.

- 3) Revised manuscript text (including main tables, and figure legends for main and EV figures) has to be submitted as editable text file (e.g., .docx format). We encourage highlighting of changes (e.g., via text color) for the referees' reference.
- 4) Each main and each Expanded View (EV) figure should be uploaded as individual production-quality files (preferably in .eps, .tif, .jpg formats). For suggestions on figure preparation/layout, please refer to our Figure Preparation Guidelines: <http://bit.ly/EMBOPressFigurePreparationGuideline>
- 5) Point-by-point response letters should include the original referee comments in full together with your detailed responses to them (and to specific editor requests if applicable), and also be uploaded as editable (e.g., .docx) text files.
- 6) Please complete our Author Checklist, and make sure that information entered into the checklist is also reflected in the manuscript; the checklist will be available to readers as part of the Review Process File. A download link is found at the top of our Guide to Authors: embopress.org/page/journal/14602075/authorguide
- 7) All authors listed as (co-)corresponding need to deposit, in their respective author profiles in our submission system, a unique ORCID identifier linked to their name. Please see our Guide to Authors for detailed instructions.
- 8) Please note that supplementary information at EMBO Press has been superseded by the 'Expanded View' for inclusion of additional figures, tables, movies or datasets; with up to five EV Figures being typeset and directly accessible in the HTML version of the article. For details and guidance, please refer to: embopress.org/page/journal/14602075/authorguide#expandedview
- 9) To facilitate reproducibility and cross-laboratory adoption of methodologies, please structure the Materials & Methods section as outlined in our guide to authors, including a completed Reagents and Tools Table that can be downloaded from our author guidelines as well (<https://www.embopress.org/page/journal/14602075/authorguide#structuredmethods>).
- 10) Digital image enhancement is acceptable practice, as long as it accurately represents the original data and conforms to community standards. If a figure has been subjected to significant electronic manipulation, this must be clearly noted in the figure legend and/or the 'Materials and Methods' section. The editors reserve the right to request original versions of figures and the original images that were used to assemble the figure. Finally, we generally encourage uploading of numerical as well as gel/blot image source data; for details see: embopress.org/page/journal/14602075/authorguide#sourcedata

At EMBO Press, we ask authors to provide source data for the main manuscript figures. Our source data coordinator will contact you to discuss which figure panels we would need source data for and will also provide you with helpful tips on how to upload and organize the files.

In the interest of ensuring the conceptual advance provided by the work, we recommend submitting a revision within 3 months (8th May 2025). Please discuss the revision progress ahead of this time with the editor if you require more time to complete the revisions. Use the link below to submit your revision:

EMBOJ-2024-119872

Referee #1

In this manuscript, Balafouti et al. address the role of the LINC complex in the formation of actin patches at the base of chromatin bridges that can be abnormally present during cytokinesis. Previous work from the Zachos lab and others indicates that actin polymerization/ actin patches prevent chromatin breakage and presumably promote genome stability. However, the mechanisms that induce the formation of actin patches close to the nucleus, at the base of chromatin bridges are unknown. Here, the authors propose a mechanotransduction-based model. They show that actin patches are generated by local RhoA-ROCK-LIMK-mDia1-pCofilin-1 signaling. They found that chromatin bridges are associated with nuclear envelop stretching, which is associated with the asymmetric localization of the LINC complex proximal to the chromatin bridge. There, the SR domain of Nesprin-2 recruits (through direct interaction) the PDZ RhoA GEF to locally activate RhoA and generate actin patches. Perturbing any of these components leads to defective patch formation and is associated with chromatin breakage. Using elegant fusion protein strategies, it is concluded that local activation of RhoA by this molecular cascade generates actin patches that prevent chromosome breakage.

The paper is very well written, and all the steps are generally well demonstrated. It is arguably a bit overwhelming (> 150 panels...), but overall, the conclusions are well supported by the data. In my opinion, the model is very interesting, and the manuscript deserves to be published in the EMBO Journal. However, there are several issues that should be addressed, including important controls. As most of the data are only supported by fixed cell analysis, key results should be confirmed by live cell imaging (see detailed comments below).

Major comments

1. The whole model is based on the local activation of RhoA/ROCK/LIMK/mDia1 at the bases of chromatin bridges, in the regions of the cell close to the intercellular canal. According to the authors, this is due to the very local recruitment of the PDZ RhoA GEF by the LINC complex, precisely where the nuclear envelope is stretched. It is therefore difficult to understand how activated ROCK delta2 (Fig. 2) and activated LIMK T508EE (Fig. 3) can properly localize to the chromatin bridge bases -and thus rescue the formation of local actin patches and consequently chromatin breakage- when RhoA is inhibited. Activated LIMK and ROCK are cytosolic and should induce actin polymerization throughout the cytoplasm. In contrast, they induce the

formation actin patches only where they are expected, while RhoA-GTP which normally directs ROCK/LIMK activity locally is absent. Similarly, the expression of activated RhoA G14V in the absence of the PDZ RhoGEF has no reason to induce the local polymerization of actin patches at the bases of the chromatin bridges (Fig. 4). The point of Figure 7 is to show that the GEF domain must be localized where the LINC complex is locally enriched. As shown by others, the expression of activated RhoA should induce the formation of actin stress fibers throughout the cell. Similar questions arise in Fig. 4l/m/n (activated Src rescue), Fig. 6e/f (siSUN1/2 rescue) and Fig. 7a-f (siLaminA/C rescue). The authors must solve this conundrum as local activation of the RhoA pathway is at the center of the proposed model.

Because depletion of RhoA does not impair nuclear tension in cells with intact chromatin bridges (e.g., Fig. 5O), one possibility is that, in RhoA-depleted cells, the endogenous RhoA that remains, although it is not sufficient to support actin patch formation, can still target the constitutively active forms of ROCK and LIMK to the base of the intercellular canal to promote actin patches. Also, because depletion of Sun1/2 or Lamin A/C reduces, but does not completely abolish, nuclear tension (Fig. 6P), the transfected constitutively active RhoA could associate, through PDZ RhoGEF, with the remaining Nesprin-2 at the base of the intercellular canal in Nesprin-2-depleted cells to promote actin patches, whereas the endogenous RhoA may not be sufficient to support actin patch formation in the absence of a relatively pronounced Nesprin-2 accumulation. This is now discussed in paragraph 1 page 20 of the revised manuscript.

2. In the same vein, the authors use an elegant fusion strategy to show that local GEF activity at the enriched LINC complex is sufficient to restore the formation of local actin patches and to prevent chromatin breakage (Fig. 7). To conclude that local RhoA activation is key for actin patch formation, an important control is missing. The authors should show that the loss of actin patches and the broken chromatin bridges observed upon Nesprin-2 depletion are not rescued by (over)expression of full-length (unfused) PDZ GEF or DHPH alone, which should be diffusely localized.

After Nesprin-2 depletion, expression of mCherry-tagged DHPH alone, did not rescue actin patches and did not prevent chromatin bridge breakage in cytokinesis compared with GFP controls (Figure 7I,J and Appendix Figure S8N; paragraph 2, page 16 of the revised manuscript).

3. Almost all the conclusions (especially rescue experiments) are based on quantification of "broken bridges" from fixed cells. Chromatin bridges are extremely stretched, over tens of micrometers in BE cells (e.g. Fig. 1a), something which is not observed in other cell types. It is important to rule out that these thin and long structures are not artefactually broken during the process of fixation, with the hypothesis that these stretched chromatin bridges are more fragile in the absence of actin patches. The authors should therefore confirm their main findings using live cell imaging of BE cells (not HeLa cells, as shown in the movies presented) and quantify the percentage of broken bridges and time of breakage (as in Fig. 1i), in particular upon RhoA inhibition, ROCK inhibition (or mDia1+LIMK1 inhibition) and Nesprin2 depletion.

We confirmed our main findings using live-cell imaging of BE cells, as suggested by the Reviewer. More specifically, DNA bridge breakage after RhoA inhibition was examined in BE cells stained with the DNA dye Biotracker, by live-cell imaging. We found that 22/22 vehicle control (DMSO) BE cells with DNA bridges sustained the DNA bridges and intercellular canals for the duration of the experiment, whereas 17/37 (46%) cells treated with RhoAi exhibited breakage of the DNA bridges and intercellular canals after 59 ± 25 min (paragraph 1, page 7; Appendix Figure S1L-N and Movies EV7-10). Also, 14/30 (47%) BE cells stained with Biotracker and treated with the ROCKi exhibited breakage of DNA bridges and intercellular canals after 62 ± 23 min (paragraph 2, page 8; Appendix Figure S3A,B and Movies EV17,18). Finally, 19/48 (40%) BE cells depleted of Nesprin-2 exhibited breakage of DNA bridges and intercellular canals after 44 ± 26 min by live-cell microscopy (paragraph 2, page 13; Appendix Figure S7A,B and Movie EV20).

4. Several important controls are lacking:

- In Fig. 1a/b (F-actin), Fig. 2a/b (ROCK), Fig. 2d/e (pLIMK), Fig. 2g/h/i (F-actin), Fig. 3g/h/i (pCofilin), Fig. 3k/l (mDia1), Fig. 3n and Fig. 6a/b (RhoA), the authors show images and quantifications of staining in situations that cannot be compared (presence vs. absence of chromatin bridges in control vs. perturbed situations, respectively). It is therefore important to also provide localization and quantification of this staining only when chromatin bridges are present.

Perturbed situations are a mixture of intact and broken (typically around 60% intact and 40% broken) chromatin bridges. We took into consideration the

Reviewers' comment and supplemented our analysis by providing localization and quantification of the relevant staining in perturbed situations with only intact chromatin bridges, in Figures 1D, S1C (actin-siRhoA intact bridges), Figures 2C, S2C (ROCK-RhoAi intact bridges), Figure 2F, S2D (pLIMK-RhoAi intact bridges), Figures 2J, S2F, S2G (actin-ROCKi intact bridges and LIMKi intact bridges), Figures 3J, S3E, S3F (pCofilin-S3-RhoAi intact bridges and LIMKi intact bridges), Figures 3M, S3G (mDia1-siRhoA intact bridges), Figures 3O, S3H (actin-mDia1i intact bridges) and Figures 6C, S7C (RhoA-siSun1/2 intact bridges).

- Fig. 1c: Actin accumulation is seen very close to the broken chromatin bridge. Why isn't this considered as an actin patch? What is exact the definition of actin patches (see point #5 below)?

Actin patches are structures of polymerized actin at the base of the cytoplasmic canal that connects the two daughter cells, in cytokinesis with chromatin bridges (Steigemann, 2009). The actin accumulation in the previous Fig. 1c was inside the intercellular canal, and not at the base of the chromatin bridge. We provided a more representative image of RhoAi cells without actin patches in our revised Figure 1C, to avoid confusion.

- Fig. 1o: The cell on the bottom left shows actin patches, but there is no chromatin bridge. It is therefore important to show that in dividing cells with no chromatin bridges, there are no actin patches close to the intercellular canal. Please provide images and quantification of actin patches in cytokinetic cells (tubulin positive bridge) with no chromatin bridges (LAP2 negative).

BE cells in late cytokinesis (i.e., connected by midbodies exhibiting midbody thickness of 400–700 nm; Petsalaki & Zachos, 2016, Petsalaki & Zachos, 2021b) and without chromatin bridges as evidenced by DAPI staining, were devoid of actin patches (paragraph 2, page 6; Figure 1D and Appendix Figure S1E).

- In figure 5, the "no bridge" controls are not appropriate controls as

they correspond to interphase cells but not to dividing cells with no chromatin bridges. No doubt that interphase cells have round-shape nuclei. Throughout this figure, please provide quantifications of nuclei/staining of cells undergoing cytokinesis but without chromatin bridges (e.g. ac-tubulin positive and LAP2 negative canals). The authors need to exclude that telophase nuclei do not have asymmetric LINC complexes nor elongated shape in the absence of chromatin bridges.

We took into consideration the Reviewers' comment and quantified the nuclear shape and Nesprin-2 localization in cells in late cytokinesis, i.e., in telophase BE cells connected by relatively thin (400-700 nm thickness) "late" midbodies (PMID:33355621, PMID: 27126587), in the absence of DNA bridges. We found that control cells with intact chromatin bridges exhibited deformed (oval-shaped) nuclei with a longer (x) axis extending parallel to the DNA bridge and a shorter (y) axis perpendicular to the bridge ($x/y > 1$) compared with telophase cells in late cytokinesis without DNA bridges that were not elongated in any specific direction (paragraph 2, page 11; Appendix Figure S5J,L). We also found that control cells with chromatin bridges exhibited accumulation of Nesprin-2 at the base of chromatin bridges (nuclear "front"), compared with late cytokinesis cells without chromatin bridges showing even distribution of Nesprin-2 at the periphery of the nucleus (paragraph 1, page 12; Appendix Figure S5J,M).

5. A better description of the structure/organization of the actin patches should be provided. 3D reconstruction of z-stacks with higher resolution microscopy to visualize the nuclear envelope vs. the actin cortex/plasma membrane should be provided. Are the actin patches all around chromatin bridges? Rather, they seem to point outwards and to deform the plasma membrane, making microvilli/microspikes. This is important, because there is no discussion of how actin patches could protect against chromosome breakage.

A 3D reconstruction of z-stacks of actin patches is provided, as requested. Actin patches are all around the base of the chromatin bridge (Movie EV19; paragraph 1, page 11 of the revised manuscript).

The mechanism by which actin patches protect against chromosome breakage is unclear. One possibility is that actin patches buffer the forces exerted on the nuclei by chromatin bridges thus preventing breakage. However, RhoA-deficient cells with intact chromatin bridges (but without actin patches) exhibit nuclear shape deformations and Sun2 nuclear lines similar with controls (Figure 5N,O), suggesting that nuclei with chromatin bridges do not exhibit additional tension in the absence of actin patches. Alternatively, the

dense structure of actin patches could increase cell stiffness and elasticity near the base of chromatin bridges, providing local mechanical support and preventing bridges from breaking (Fletcher & Mullins, 2010). This is now discussed in paragraph 3, page 20 and paragraph 1, page 21 of the revised manuscript.

6. Related to the point above: do nuclei experience more tension in the absence of actin patches? Perhaps actin patches buffer the forces exerted on the nuclei by chromatin bridges, preventing breakage. Quantifying differences in the number of SUN2 lines per nucleus after siRhoA or siPDZ should answer this question.

Please see our response to Reviewers' comment #5 above.

7. I do not find the conclusion based on the DN VPS4 conclusive, as this mutant not only inhibits abscission but also perturbs nuclear envelope reformation. The live cell imaging mentioned in point #3 should provide better evidence that the chromatin bridges are broken by longitudinal tension, rather than by membrane abscission.

We removed the dnVps4 experiment from our revised manuscript, as suggested by the Reviewer. The purpose of this experiment was to demonstrate that chromatin bridge breakage in RhoA-deficient cells was not caused by abscission. Because approximately 70% of RhoA-depleted cells with broken DNA bridges exhibited intact intercellular canals as evidenced by membrane staining with the FM1-43FX dye (Figure 1M), our results from fixed cells are consistent with chromatin bridge breakage in RhoA-deficient cells correlating with impaired actin patches and not with premature abscission. Live-cell imaging of BE cells also showed breakage of DNA bridges after RhoA-, ROCK or Nesprin-2-inhibition, as requested by the Reviewer (see our response to Comment #3, above).

Minor comments:

1. The role of the actin-binding domain of the LINC complex in generating nuclear tension is unclear. It seems to this reviewer that the LINC complex distal to the chromatin bridges couples the nuclei to the cell cortex. As the dividing cell migrates/spreads, cortical forces are

transmitted to the nuclei and chromatin bridges are pulled by the opposite nuclei. This point should at least be discussed.

How does the LINC complex promote nuclear tension in cytokinesis with chromatin bridges? One possibility is that LINC complexes distal to chromatin bridges couple the nuclei to the actomyosin network and the cell cortex through their actin binding domains and that, as the dividing cell migrates or spreads, cortical forces are transmitted to the nuclei while chromatin bridges are pulling the daughter nuclei towards the opposite direction thus generating nuclear tension. This is now discussed in paragraph 2, page 21 of the revised manuscript.

2. Fig. 6b, 6n and 6o: In contrast to control cells, F-actin is much more abundant along the intercellular canals when the LINC complex is perturbed. Can the authors speculate why?

Control cells with chromatin bridges typically exhibit long and thin intercellular canals that appear as narrow as the chromatin bridge. In contrast, LINC-deficient cells often exhibit relatively short and wide intercellular canals, perhaps because connections between membrane-bound proteins (e.g., anillin) and cortical microtubules that anchor the plasma membrane to the midbody cell cortex (PMID: 15854913, PMID: 16461284) are disrupted; as a result, the actin signal could appear stronger. Alternatively, impaired actin patches in LINC-deficient cells with chromatin bridges may lead to reorganization of actin along the length of the intercellular canal.

3. Discussion/Introduction: The present results should be better discussed in the context of the previously published literature. In particular:

- How do the results fit with the regulation of actin levels by actin oxidation/reduction (PMID 32029597)?**
- Why broken bridges here and not canal reopening in the absence of actin patches (see PMID 19203582, 26929449, PMID 32029597)?**
- What is the link between this pathway and the activated NoCut checkpoint?**
- How do these results fit with the Pellman paper showing that actomyosin II is required for chromatin breakage (PMID: 32299917)? According to this published paper, RhoA inhibition should prevent myosin II activation and thus chromatin breakage. Here, the opposite is**

seen (Fig. 1). This should at least be discussed

3.1. Control cells with chromatin bridges prevent the depolymerization of actin filaments inside the intercellular canal by recruiting the human methionine sulfoxide reductase B2 (MsrB2) to the midbody, to delay recruitment of ESCRT-III proteins to the abscission site and to delay abscission (Bai, Wioland et al., 2020). This mechanism prevents chromatin bridge breakage in cytokinesis, but is different from the actin patches mechanism we describe: Actin patches are structures of polymerized actin at the base of the cytoplasmic canal that connects the two daughter cells; furthermore, impaired actin patches correlate with chromatin bridge breakage that is not caused by premature abscission (Dandoulaki et al., 2018, Steigemann et al., 2009). This is now discussed in paragraph 2, page 3 and in paragraph 1, page 4 of the revised manuscript.

3.2. Why impaired actin polymerization/stabilization leads to chromatin bridge breakage in some cases (Dandoulaki et al., 2018) but furrow regression in others (Steigemann et al., 2009) remains unclear and could depend on the experimental conditions (Petsalaki & Zachos, 2021a). This is noted in paragraph 3, page 19 of the revised manuscript.

3.3. Whether actin patch formation is somehow linked to the activation of the abscission checkpoint is a matter of active investigation. For example, it will be interesting to examine whether cells with dicentric bridges that do not activate the abscission checkpoint can generate actin patches through the present mechanism (Petsalaki et al., 2023). This is now discussed in paragraph 3, page 19 of the revised manuscript.

3.4. It was previously reported that inhibition of myosin light chain kinase, which prevents myosin II activation but did not apparently block chromatin bridge extension, delayed chromatin bridge breakage in cytokinesis (Umbreit et al., 2020). Because RhoA inhibition can prevent myosin II activation, other members of the Rho protein family could activate myosin II in RhoA-deficient cells with chromatin bridges (Hirano & Hirano, 2022). Alternatively, longitudinal tension rather than contractile actomyosin forces may constitute the main cause of chromatin bridge breakage in RhoA-deficient cells in our experimental system. This is now discussed in paragraph 2, page 19 of the revised manuscript.

4. In several quantifications (Fig. 1d, 2c, 2f, 2j, 2q etc.), I assume that intensities from different independent experiments have been pooled. It would be better to present the results as mean +/- SD for independent experiments, as in the other quantifications. If these quantifications have been done only once, results from independent replicates should be reported.

Bar graphs in Figures 1D, 2C, 2F, 2J, etc, typically indicate mean fluorescence intensities \pm SD from n cells from two independent experiments, as mentioned in the relevant Figure legends. By plotting the individual values for each cell, these graphs provide additional clarity and are consistent with the Journals' recommendation that, when the number of independent repeats for practical reasons is limited, the actual individual data from each experiment should be plotted. Chromatin bridges are relatively hard to find because they are infrequent and we have presented our data this way in several recent papers. Because the differences in intensities between control and mutant cells is very clear, we believe it wouldn't make a difference either way and propose to keep the quantifications as they are, also for clarity reasons.

Referee #2 (Report for Author)

This manuscript explores an interesting potential mechanism linking nuclear tension during cytokinesis to actin patch formation through LINC complex-mediated RhoA signaling. While the biological phenomenon is intriguing and potentially important for understanding genome stability, there are significant technical issues that undermine the authors' central conclusions.

The manuscript contains numerous technical oversights and terminology issues that need addressing. Several key abbreviations require proper definition, including SUN (Sad1/UNC-84), NESPRIN (nuclear envelope spectrin repeat-enriched protein), KASH (Klarsicht/ANC-1/SYNE homology), PDZ (PSD-95/DGL/ZO-1), LIMK (LIN-11/IsI-1/MEC-3 kinase), and ROCK (Rho-associated protein kinase). Technical corrections are needed, such as changing "N-terminal actin-binding calponin homology (CH) domain" to "N-terminal tandem actin-binding calponin homology (CH) domains" (Page 4, Lines 9-10). The authors should also note that small Rho GTPases regulate microtubule dynamics in addition to actin organization (PMID: 24691223 and 24988197).

Definitions for Sun (paragraph 2, page 4), Nesprin (paragraph 2, page 4), KASH (paragraph 2, page 4), PDZ (paragraph 2, page 5), LIMK (paragraph 1, page 5) and ROCK (paragraph 1, page 5) abbreviations were provided at their first mentioning in the text, as requested. We also changed "N-terminal actin-binding calponin homology (CH) domain" to "N-terminal tandem actin-binding calponin homology (CH) domains" (paragraph 2, page 4). We also noted that Rho GTPases regulate a diverse range of cellular functions including cell polarity, cell movement and cell division through their abilities to modulate the actin cytoskeleton and regulate microtubule dynamics, and included reference PMID 24691223 (paragraph 1, page 5 of the revised manuscript), as suggested by the Reviewer.

Several critical controls are missing throughout the study. The authors rely heavily on RNAi experiments but fail to include rescue controls for most of their knockdowns, making it impossible to rule out off-target effects. While they demonstrate rescue with GFP:RhoA_{res} for RhoA-depletion, similar controls are notably absent for other key proteins. The DN-KASH experiments lack the essential KASH- Δ L control, a construct based off DN-KASH but lacking the luminal SUN protein-binding KASH domain (PMID: 20724637), which is critical for demonstrating specificity

of LINC complex disruption versus general nuclear envelope perturbation.

Vehicle controls are missing for small molecule experiments, and a wild-type GFP:Vps4 control should be included.

The authors should also consider examining FHOD1 or FHOD3 localization at the actin patches, given that nesprin-2G directly interacts with FHOD1 in other contexts (PMID: 24880667, 32460023).

We took into consideration the Reviewers' comment and showed that expression of Citrine:PDZ^{Res} resistant to degradation by PDZ siRNA-2 (siPDZ-2), but not of GFP-only, rescued actin patches and prevented chromatin bridge breakage in cytokinesis after PDZ-depletion compared with GFP controls (paragraph 3, page 9 and paragraph 1, page 10; Appendix Figure S3L-R).

Rescue for Nesprin-2 depletion by transfection of a siRNA-resistant full-length Nesprin-2 is not technically possible because of the huge size of the protein (~800 kDa).

We also generated the KASH- Δ L construct lacking the luminal SUN protein-binding KASH domain (Luxton et al., 2010), as requested. We showed that cells transfected with dnKASH, but not cells expressing KASH- Δ L, exhibited reduced actin patches at the base of the intercellular canal and increased frequency of broken chromatin bridges in cytokinesis compared with mCherry controls (paragraph 2, page 13; Appendix Figure S6J, N-Q).

For live-cell imaging, control cells were treated with vehicle (DMSO) and this was clarified in the relevant Methods section (page 26), Results (pages 6-8), Figures and Figure Legends. In fixed cells analysis, vehicle controls (DMSO) were included for actin patches and broken DNA bridges in Figures 1D,E, 2J,K and 3O-Q. Furthermore, vehicle controls were shown for ROCK (Figure 2C), pLIMK-T508 (Figure 2F), and pCofilin-S3 (Figure 3J) as requested.

Because expression of dnVps4 could potentially interfere with nuclear envelope formation and dynamics, this experiment was removed as requested by the Reviewer #1.

Perhaps interestingly, the mammalian formin FHOD1 (Formin Homology 2 Domain Containing 1), which is a target of ROCK and can interact with Nesprin-2 under specific conditions (Luxton et al., 2010, Takeya, Taniguchi et al., 2008), is also found at actin patches in control cells (paragraph 2, page 15 and paragraph 1, page 16; Appendix Figure S8K).

There are several significant concerns regarding the characterization and presentation of nesprin-2-related experiments. The authors propose that PDZ-RhoGEF activates RhoA to promote actin patch formation, but no direct evidence supports this conclusion. They should demonstrate that actin RhoA localizes to actin patches in a PDZ-RhoGEF-dependent manner, perhaps using a FRET-based biosensor.

The manuscript fails to consider the different isoforms of nesprin-2 expressed in their experimental cell lines, which is particularly important given that their anti-nesprin-2 antibody (Abcam) recognizes an epitope within amino acids 4200-4350 of the SR-containing cytoplasmic domain. The authors should verify whether full-length nesprin-2G displays similar localization patterns.

Additionally, the authors should adopt standard field nomenclature, referring to their "short" nesprin-2 constructs lacking SRs 3-54 as "mini-nesprin-2G" constructs. Key references for these constructs are missing, including their first description (PMID: 19843581) and initial functional demonstration (PMID: 20724637). Similarly, the I128,131A mutation in the actin-binding CH domains needs proper definition and citation (PMID: 20724637). Importantly, this mutation impairs but does not completely abolish actin binding, contrary to what is depicted in Figure 6G.

To examine Rho activity, we used the dimericTomato-2xrGBD (dT-2xrGBD) Rho biosensor consisting of a dimericTomato fluorescence protein and a double rhotekin G protein-binding domain (Mahlandt, Arts et al., 2021). dT-2xrGBD localized to actin patches at the base of the intercellular canal in control cells with chromatin bridges and this localization was impaired in PDZ-deficient cells, indicating that localization of active Rho to actin patches is PDZ-dependent (paragraph 1, page 10; Appendix Figure S4A-C).

There are 13 Nesprin-2 isoforms produced by alternative splicing according to the Uniprot database (<https://www.uniprot.org/uniprotkb/Q8WXH0/entry#sequences>). Because most of them are truncated proteins, the Reviewer asks to verify that it is the full-length Nesprin-2 that is enriched at the base of chromatin bridges. Isoform 1 is the canonical sequence and isoform 2 has an insertion of additional 22 amino acids; therefore, they both represent full-length proteins. Isoforms 3, 4, 5, 6, 10, 11, 12 and 13 are missing the first (i.e., the C-terminal) approximately 6000 amino acids; therefore, they are not detected by the anti-Nesprin-2 antibody (ab204308) we use, which recognizes an epitope within amino acids 4200-4350. Isoforms 8 and 9 are missing amino acids 286-6885; therefore,

they also are not detected by the anti-Nesprin-2 ab204308 antibody. Isoform 7 is missing amino acids 1-3638 and is the only truncated isoform that can be detected by the anti-Nesprin-2 ab204308 antibody we use. Therefore, to verify enrichment of full-length Nesprin-2 at the base of chromatin bridges, we also used a second antibody (Ab2; ab233034) that is raised against amino acids 1-300 of Nesprin-2 and is not expected to recognize the truncated isoform #7. By using Ab2, we showed that Nesprin-2 accumulates at the nuclear front in control cells with intact chromatin bridges and that this accumulation is diminished in Sun1/2-depleted cells, by confocal microscopy (paragraph 1, page 12; Appendix Figure S6F-I).

We replaced “short” Nesprin-2 with “mini-Nesprin-2G” throughout the manuscript (e.g., in pages 13, 14, 16 and elsewhere). We also included references PMID:19843581 and PMID:20724637 (paragraph 3, page 13) and explained that the CH* mutant exhibits the Isoleucines 128, 131 to alanine point mutations that impair Nesprin-2 binding to F-actin (Luxton et al., 2010). (paragraph 1, page 23). Also, we noted that the CH* mutant exhibits “impaired actin binding” in Figure 6G, bottom, as suggested by the Reviewer.

The reagents used raise significant concerns. The lamin A/C-RFP construct shows an abnormal localization pattern reminiscent of non-farnesylated progerin (PMID: 22895092), suggesting the C-terminal RFP tag may be disrupting proper lamin processing and assembly. An N-terminal tag, which is well-established in the field, should be used instead.

The biochemical evidence for interaction between PDZ-RhoGEF and nesprin-2 SRs is compromised by extensive protein degradation products. It would be valuable to demonstrate whether endogenous nesprin-2 interacts with endogenous PDZ-RhoGEF via co-immunoprecipitation.

The authors should also note that the formin inhibitor SMIFH2 is not specific for formins as it can inhibit members of the myosin superfamily (PMID: 33589498), which might alter the interpretation of their findings.

Cells expressing relatively low levels of Lamin A:RFP exhibited correct localization of Lamin A:RFP at the periphery of the nucleus (Appendix Figure S9F), suggesting that this protein was properly processed at least when expressed at low levels. Regardless of the exact reason behind the mislocalization of Lamin A:RFP (i.e., whether it is caused by overexpression or by improper processing under specific conditions), our purpose was to use this protein as a tool to engineer the mislocalization of the endogenous

Nesprin-2 from the periphery of the nucleus in cells with chromatin bridges (PMID:15843432). By using this approach, we found that Nesprin-2 did not accumulate at the base of chromatin bridges, but co-localized with Lamin A:RFP at nuclear granules, and this correlated with impaired actin patches and with local polymerization of actin inside the Nesprin-2/Lamin A:RFP nuclear granules (Figure 8). If the N-terminal RFP-tag of Lamin A does not generate Lamin A granules and, as a result, does not cause mislocalization of the endogenous Nesprin-2 in cells with chromatin bridges, it will not be useful for our purposes. Furthermore, we believe that investigating the exact cause of Lamin A:RFP-mislocalization in this construct is beyond the scope of our study. We noted that “The mislocalization of Lamin A:RFP could be due to overexpression and/or improper processing of the protein under high levels of expression (Wang, Ostlund et al., 2012)”, in paragraph 2, page 17 of the revised manuscript.

Because Nesprin-2 is a giant protein (~800 kDa), co-immunoprecipitations with the endogenous protein are technically impossible to work. To further support the interaction of PDZ RhoGEF with Nesprin-2 SRs, we showed that GST-SRs(21-30), but not GST-only, associated with the endogenous PDZ RhoGEF from cell extracts, by GST-pull downs (paragraph 2, page 15; Appendix Figure S8G).

We noted that SMIFH2 can also inhibit members of the myosin superfamily and cited the PMID:33589498 paper, as suggested by the Reviewer (paragraph 1, page 15). To differentiate between a potential role for mDia1 and/or myosin II in actin patch formation, we used the myosin II inhibitor blebbistatin. Treatment of cells with blebbistatin impaired nuclear shape deformation, Nesprin-2 accumulation and actin patches in cytokinesis compared with controls, suggesting that actomyosin forces are required for nuclear tension and actin patch formation (Appendix Figure S8C-F) (Kovacs, Toth et al., 2004, Ramamurthy, Yengo et al., 2004). Because treatment with the mDia1i SMIFH2 did not attenuate nuclear shape deformation (Appendix Figure S8F), these results also argue against mDia1i reducing actin patches by inhibiting myosin II (Nishimura, Shi et al., 2021) and further support a role for mDia1 in actin patch formation (paragraph 1, page 15 of the revised manuscript).

The manuscript also overlooks important literature and contains inaccurate statements. The authors should reference previous work showing that LINC complexes regulate small Rho GTPase signaling (PMID: 28035049) and that nesprin-2G promotes nuclear envelope-associated actin meshwork assembly during prophase (PMID:

31264963). Their description of TAN lines needs correction - these are specifically linear arrays of nesprin-2G/SUN2-containing LINC complex that form along dorsal perinuclear actin cables perpendicular to the direction of cell migration, and do not include SUN1 (PMID: 20724637). Moreover, TAN lines are not differentiated from "actin caps" as being longitudinal or transverse, respectively. Further, the statement that "LINC complex promotes generation of mechanical tension" is inaccurate - LINC complexes transmit, rather than generate, mechanical tension. Maybe they can store tension.

Our data is consistent with previous findings that LINC complexes can promote RhoA activation and the assembly of an actomyosin network around the nucleus through transcription-independent mechanisms under specific conditions (Booth, Yue et al., 2019, Thakar, May et al., 2017). This is now discussed in paragraph 3, page 18 and in paragraph 1, page 19, as suggested by the Reviewer.

We thank the Reviewer for these corrections. We rewrote paragraph 2, page 21, also taking into consideration the minor Comment #1 of Reviewer #1. Furthermore, we changed: "the nuclear envelope Sun1/2-Nesprin-2 LINC complex promotes generation of mechanical tension on daughter nuclei...", to: "the nuclear envelope Sun1/2-Nesprin-2 LINC complex transmits mechanical tension on daughter nuclei..." (paragraph 2, page 18), as suggested by the Reviewer.

In summary, while this work presents an interesting biological observation, the technical issues significantly impact the reliability of the core mechanistic conclusions. Major revisions addressing these concerns would be needed before this manuscript would be suitable for publication in EMBO Journal. The authors should also consider examining whether increased nuclear stiffness (e.g., through progerin expression) affects nesprin-2 accumulation at chromatin bridges and investigate whether SRs 31-17 directly interact with PDZ RhoGEF through a conserved motif.

Cells were transfected with progerin fused to GFP (progerin:GFP), which increases nuclear stiffness (Booth, Spagnol et al., 2015, Verstraeten, Ji et al., 2008). In cells with intact chromatin bridges, expression of progerin:GFP correlated with reduced nuclear chromatin shape-deformation, impaired Nesprin-2 accumulation at the base of chromatin bridges and impaired actin patches compared with mCherry- or GFP-only controls (paragraph 1, page 18; Appendix Figure S9G-K).

To map the region(s) of PDZ RhoGEF that interact with Nesprin-2 Spectrin Repeats (SRs), we generated constructs expressing the N-terminal PDZ domain (PDZdom), the RH domain, the DH/PH domain, or the C-terminal region of PDZ RhoGEF fused to GFP. We found that GFP-tagged proteins containing the RGS-homology (RH) or the Dbl homology/pleckstrin homology (DH/PH) domains of PDZ RhoGEF, but not the PDZ domain (PDZdom) or the C-terminal (C-term) region, associated with GST-SRs 31-37 by GST pull-downs (Appendix Figure S8H-J). These results suggest that the Nesprin-2 SRs bind to PDZ RhoGEF at the RH and DH/PH domains (paragraph 2, page 15; also discussed in paragraph 3, page 21 of the revised manuscript).

Prof. George Zachos
University of Crete
Department of Biology
Vassilika Vouton
Heraklion, Crete 70013
Greece

29th Jul 2025

Re: EMBOJ-2024-119872R
Tension-sensitive LINC-RhoA signaling prevents chromatin bridge breakage in cytokinesis

Dear Dr. Zachos,

Thank you for submitting your revised manuscript to The EMBO Journal. It has now been re-assessed by the two original reviewers, whose comments are copied below. Both referees acknowledge the revisions and the satisfactory addressing of the majority of concerns, but while referee 1 only retains a few minor presentational issues, referee 2 still feels that two key points have not been adequately addressed during revision. In particular, it remains essential to validate the absence of off-target effects in Nesprin-2G depletion experiments, for which the referee suggests one straightforward possibility. The other concern regards the lack of at least semi-endogenous co-IPs of PDZ-RhoGEF with Nesprin-2G (or parts of it) from cells, instead of just showing in-vitro pulldown interactions.

As I mentioned, we normally allow only a single round of major revision - to avoid repeated rounds of partial improvements and to give authors a decisive commitment from our side. However, given that you have already made significant efforts and addressed a majority of issues during the first revision, I would in this case allow an exceptional second round of experimental revision, to allow you to clarify the remaining open points. With such additional strengthening of the study, we would be happy to eventually accept it for EMBO Journal publication. As always, I would be open to considering possible options for revising based on a preliminary response letter, which I might in this case discuss with the critical referee to get their views on how promising they would seem.

When preparing a re-revised manuscript addressing the persistent scientific issues, please also take care of the following editorial points that we had noted during our routine revision checks:

- Please adjust the order of the manuscript sections: Title page with complete author information, Abstract, Keywords, Introduction, Results, Discussion, Methods, Data Availability, Acknowledgements, Disclosure and Competing Interests Statement, References, Main Figure Legends, Tables, Expanded Figure Legends.
- Please rename the Conflict of Interest section into "Disclosure and Competing Interests Statement", in accordance with our updated Guide to Authors (<https://www.embopress.org/competing-interests>)
- As we are switching from a free-text author contribution statement towards a more formal statement based on Contributor Role Taxonomy (CRediT) terms, please remove the present Author Contribution section and instead specify each author's contribution(s) directly in the Author Information page of our submission system during upload of the final manuscript. See <https://casrai.org/credit/> for more information. Please, also remove the section title Materials and Correspondence.
- In the Data Availability section, please include a direct URL linking to the GenBank database in which manuscript-related data has been deposited.
- Please cut the EV movie legends from the main text, instead placing each one into one separate legend text file per EV movie; then move each legend file together with the respective movie file into a separate ZIP archive before re-uploading as "Movie EV1/2/3..."
- For the Appendix, our routine pre-publication image checks indicate that the certain panels appear in both Appendix Figures S1H and S4G, without this being stated in the respective figure legends. This needs to be clarified, and if justifiable, explicitly mentioned in the figure legends.
- Furthermore, the resolution for the Appendix Figures is currently too low. Please supply a reformatted Appendix file at a higher resolution. Currently, blots and microscopy are too low-resolution. Such a reduction in resolution is commonly caused by converting original 16-bit TIFF files to RGB format for publication. While this is not inherently problematic, it can raise concerns about image integrity for critical readers.

- On the title page of the Appendix, please state "Appendix for [manuscript title]", and the table of contents should refer to "Figure titles", not "Figure legends" as is currently written.

- Finally, during routine pre-acceptance checks, our data editors have raised the following queries regarding figures, data, and legends; I would appreciate if you briefly answered to them in the cover letter of your final submission, and made the requested text modifications with changes/additions highlighted via the "Track changes" option, to facilitate our final checking.

1. Please note that the legends for figures 6 and 7 are not provided in an alphabetical (A, B, C...) manner. This needs to be rectified. Maybe consider mentioning details applying to several panels in a separate paragraph "Data information: xxx" at the end of the legend of Figure 2. Also keep in mind that all references to figures and individual panels in the main text should appear in a sequential manner.

2. Please note that information related to N is missing in the legend of figure 2F

3. Please note that the scale bar needs to be defined for figures 4B, D, F, G, H; 6A, B; 7A, B, D, E; 8B, C, D, E.

4. Please note that the intact and broken white arrows are not defined in the legend of figures 6A, B; 7A, B, D, E. This needs to be rectified.

I am therefore returning the manuscript to you for a second, final round of experimental revision, with the link below for eventual resubmission. Should you have any questions regarding the referee comments or this decision, please do not hesitate to contact me directly.

Yours sincerely,

Hartmut Vodermaier

- size of the scale bars that are mandatory for all micrograph panels

- the statistical test used to generate error bars and P-values

- the type error bars (e.g., S.E.M., S.D.)

- the number (n) and nature (biological or technical replicate) of independent experiments underlying each data point

- Figures may not include error bars for experiments with $n < 3$; scatter plots showing individual data points should be used instead.

9) To facilitate reproducibility and cross-laboratory adoption of methodologies, please structure the Materials & Methods section as outlined in our guide to authors, including a completed Reagents and Tools Table that can be downloaded from our author guidelines as well (<https://www.embopress.org/page/journal/14602075/authorguide#structuredmethods>).

10) Digital image enhancement is acceptable practice, as long as it accurately represents the original data and conforms to community standards. If a figure has been subjected to significant electronic manipulation, this must be clearly noted in the figure legend and/or the 'Materials and Methods' section. The editors reserve the right to request original versions of figures and the original images that were used to assemble the figure. Finally, we generally encourage uploading of numerical as well as gel/blot image source data; for details see: embopress.org/page/journal/14602075/authorguide#sourcedata

In the interest of ensuring the conceptual advance provided by the work, we recommend submitting a revision within 3 months (27th Oct 2025). Please discuss the revision progress ahead of this time with the editor if you require more time to complete the revisions. Use the link below to submit your revision:

Link Not Available

Referee #1:

The authors responded convincingly to my comments. I therefore recommend publication in the EMBO J. They are to be congratulated on this very interesting paper.

However, a few errors remain to be corrected:

- Fig. S5L legend: the sentence '(L) Nuclear chromatin shape in control cells with DNA bridges and in interphase cells in late cytokinesis without DNA bridges.' should be corrected.

- Discussion p.19 'Why impaired actin polymerization/stabilization leads to chromatin bridge breakage in some cases (Dandoulaki et al., 2018) but furrow regression in others (Steigemann et al., 2009) remains unclear and could depend on the experimental conditions (Petsalaki & Zachos, 2021a).' has a reference issue. To my knowledge, furrow regression upon actin perturbation has been reported in (Bai, Wioland et al. 2020) and not in (Steigemann et al., 2009).

The sentence should thus read: 'Why impaired actin polymerization/stabilization leads to chromatin bridge breakage in some cases (Dandoulaki et al., 2018) but furrow regression in others (Bai, Wioland et al., 2020) remains unclear and could depend on the experimental conditions (Petsalaki & Zachos, 2021a).'

Referee #2:

Overall, the authors have successfully addressed most of my previously raised concerns. However, two points remain.

1) The authors claim that nesprin-2G rescue is "not technically possible because of the huge size of the protein (~800 kDa)." This reasoning is unconvincing given their own data. They extensively use mini-nesprin-2G constructs throughout Figures 6-8, demonstrating these truncated proteins are functional and localize correctly to chromatin bridges. If mini-nesprin-2G constructs can recapitulate endogenous nesprin-2G function in their assays, which the authors clearly demonstrate, then rescue experiments are absolutely feasible. The authors need only design siRNAs targeting the 3' UTR or spectrin repeats absent from their constructs, then show mini-nesprin-2G rescues the phenotype. This is standard practice. Without these controls, off-target effects cannot be ruled out, which is problematic given how central nesprin-2 depletion is to their conclusions.

2) The dismissal of endogenous co-IP of nesprin-2G as "technically impossible" is overstated. Multiple groups have successfully co-IP'ed nesprin-2G despite its size (e.g., PMIDs: 20801886, 15671068). Yes, it is challenging, but hardly impossible. The GST pulldowns only show in vitro binding. For a model this mechanistically detailed, demonstrating the nesprin-2/PDZ-RhoGEF interaction occurs in cells at physiological levels is essential. Even attempting the experiment and reporting negative results would be more convincing than refusing to try.

Referee #1:

The authors responded convincingly to my comments. I therefore recommend publication in the EMBO J. They are to be congratulated on this very interesting paper.

However, a few errors remain to be corrected:

- Fig. S5L legend: the sentence '(L) Nuclear chromatin shape in control cells with DNA bridges and in interphase cells in late cytokinesis without DNA bridges.' should be corrected.

This sentence is now changed to: “Nuclear chromatin shape in control cells with intact DNA bridges and in cells in late cytokinesis without DNA bridges”, in Fig. S5 legend.

- Discussion p.19 'Why impaired actin polymerization/stabilization leads to chromatin bridge breakage in some cases (Dandoulaki et al., 2018) but furrow regression in others (Steigemann et al., 2009) remains unclear and could depend on the experimental conditions (Petsalaki & Zachos, 2021a).' has a reference issue. To my knowledge, furrow regression upon actin perturbation has been reported in (Bai, Wioland et al. 2020) and not in (Steigemann et al., 2009).

The sentence should thus read: 'Why impaired actin polymerization/stabilization leads to chromatin bridge breakage in some cases (Dandoulaki et al., 2018) but furrow regression in others (Bai, Wioland et al., 2020) remains unclear and could depend on the experimental conditions (Petsalaki & Zachos, 2021a).'

The sentence was changed and the Steigemann et al reference was replaced by the Bai et al reference, as requested (paragraph 3, page 19).

Referee #2:

Overall, the authors have successfully addressed most of my previously raised concerns. However, two points remain.

1) The authors claim that nesprin-2G rescue is "not technically possible because of the huge size of the protein (~800 kDa)." This reasoning is unconvincing given their own data. They extensively use mini-nesprin-2G constructs throughout Figures 6-8, demonstrating these truncated proteins are functional and localize correctly to chromatin bridges. If mini-nesprin-2G constructs can recapitulate endogenous nesprin-2G function in their assays, which the authors clearly demonstrate, then rescue experiments are absolutely feasible. The authors need only design siRNAs targeting the 3' UTR or spectrin repeats absent from their constructs, then show mini-nesprin-2G rescues the phenotype. This is standard practice. Without these controls, off-target effects cannot be ruled out, which is problematic given how central nesprin-2 depletion is to their conclusions.

The mouse mini-Nesprin-2G (CB) we use (and its' derivative proteins) is resistant to degradation by the pool of three Nesprin-2 siRNAs against the human sequence we use (siNesprin-2). More specifically, the human Nesprin-2 siRNAs A and B are inside a coding region that exhibits some differences between the mouse and human sequences (see below), whereas Nesprin-2 siRNA C is outside the coding sequence.

```
human      CTGGAAGAGGAGCTGGTAGAACGTCAACCTCAAAGTGGACATGTTACAGGAGATTTCAAAC
           || ||||| ||||| ||||| ||||| ||||| ||||| ||||| ||||| ||||| ||||| |||||
mouse      CTAGAAAAGGAGTTGGTAGCCCGTCAGCCTCAAGTCAGCTCCTTGCGGGAGATTTCTCTCC

human      AGCCTTCTCATTAAAGGACATGGAGAAGACTGTATTGAAGCTGAAGAAAAGGTGCATGTT
           ||||| ||||| ||||| ||||| ||||| ||||| ||||| ||||| ||||| |||||
mouse      AGCCTTCTGTTGAAGGGCAAGGAGAAGACTACATTGAGGCCGAAGAGAAGGTCCACGTG
```

siRNA sc-61630A is against human 20,139 - 20,157 of Nesprin-2
(GGTAGAACGTCAACCTCAA)
siRNA sc-61630B is against human 20,180 - 20,198 of Nesprin-2
(CAAACAGCCTTCTCATTAA)

This was also shown experimentally by western blotting: we showed that mini-Nesprin-2G CB or mutant CH* protein levels were not reduced by siNesprin-2 (Appendix Figure S8N, O). Importantly, after depletion of the endogenous Nesprin-2 in human cells, expression of siRNA-resistant mini-Nesprin-2G (CB) containing the spectrin repeats 31-37 (CB:SRs 31-37) rescued actin patches and prevented chromatin bridge breakage in cytokinesis compared with GFP controls (Figure 7I, J). Furthermore, cells with intact chromatin bridges expressing CB:SRs31-37 in the presence of siNesprin-2, exhibited nuclear chromatin-shape deformation and relatively low frequency of loose chromatin bridges similar to GFP controls (Figure 7K and Appendix Figure S9A). This is described in paragraph 2, page 16 of the revised manuscript. We also clarified that mouse mini-Nesprin-2 is resistant to human Nesprin-2 siRNA because the mouse sequence is sufficiently diverse from the human sequence, in paragraph 3, page 13.

2) The dismissal of endogenous co-IP of nesprin-2G as "technically impossible" is overstated. Multiple groups have successfully co-IP'ed nesprin-2G despite its size (e.g., PMIDs: 20801886, 15671068). Yes, it is challenging, but hardly impossible. The GST pulldowns only show in vitro binding. For a model this mechanistically detailed, demonstrating the nesprin-2/PDZ-RhoGEF interaction occurs in cells at physiological levels is essential. Even attempting the experiment and reporting negative results would be more convincing than refusing to try.

We showed that the endogenous Nesprin-2 associated with PDZ RhoGEF by coimmunoprecipitation experiments in cell extracts (Appendix Figure S8H; paragraph 2, page 15 of the revised manuscript), as requested. This experiment was done twice.

Prof. George Zachos
University of Crete
Department of Biology
Vassilika Vouton
Heraklion, Crete 70013
Greece

29th Aug 2025

Re: EMBOJ-2024-119872R1
Tension-sensitive LINC-RhoA signaling prevents chromatin bridge breakage in cytokinesis

Dear George,

Thank you for submitting your final revised manuscript for our consideration, and please excuse the holiday-related delays in its editorial assessment. I have now had the chance to carefully go through your final responses, and found them to satisfactorily address the concerns that had remained in previous round of reviews. We are therefore pleased to inform you that we are herewith accepting the study for publication in The EMBO Journal.

With kind regards,

Hartmut
